



**Reconstructing past variations in environmental conditions and paleoproductivity**
**over the last ~8000 years off Central Chile (30° S)**
Práxedes Muñoz[1,2], Lorena Rebolledo[3,4], Laurent Dezileau[5], Antonio Maldonado[2],
Christoph Mayr[6,7], Paola Cárdenas[4,8], Carina B. Lange[4,9,10], Katherine Lalangui[9], Gloria
Sanchez[11], Marco Salamanca[9], Karen Araya[1,5], Ignacio Jara[2], Gabriel Vargas[12], Marcel
Ramos[1,2].
[1]Departamento de Biología Marina, Universidad Católica del Norte, Larrondo 1281,
Coquimbo, Chile.
[2]Centro de Estudios Avanzados en Zonas Áridas (CEAZA), Coquimbo-La Serena,
Chile.
[3]Departamento Científico, Instituto Antártico Chileno, Punta Arenas, Chile.
[4]Centro FONDAP de Investigación Dinámica de Ecosistemas Marinos de Altas
Latitudes (IDEAL), Universidad Austral de Chile, Campus Isla Teja, Valdivia, Chile.
[5]Laboratoire Géosciences Montpellier (GM), Université de Montpellier, 34095
Montpellier Cedex 05, France.
[6]Institut für Geographie, FAU Erlangen-Nürnberg, 91058 Erlangen, Germany.
[7]Department of Earth and Environmental Sciences & GeoBio-Center, LMU Munich,
80333 Munich.
[8]Programa Magister en Oceanografía, Universidad de Concepción, casilla 160C,
Concepción, Chile.
[9]Departamento de Oceanografía, Facultad de Ciencias Naturales y Oceanográficas,
Universidad de Concepción, Casilla 160C, Concepción, Chile.
[10]Centro de Investigación Oceanográfica COPAS Sur-Austral, Universidad de
Concepción, Casilla 160C, Concepción, Chile.
[11]Universidad de Magallanes, Punta Arenas, Chile.
[12]Departamento de Geología, Universidad de Chile, Santiago, Chile.
*Correspondence*: Práxedes Muñoz (praxedes@ucn.cl)





**Abstract**
This study aims at establishing past variations of the main oceanographic and climatic
features in the Central Chilean coast, using recent sedimentary records of a transitional
semi-arid ecosystem susceptible to environmental forcing conditions. Coquimbo (30°S)
region is characterized by dry summers and short rainfall periods during winter. The
relatively wet-winter climate results from the interactions between the southern westerly
winds and the South Pacific Anticyclone (SPA); in summer, the SPA moves southwards
while in winter it returns to the north, allowing the passage of storm fronts. This semi-
arid zone is strongly affected by variations associated with El Niño-Southern Oscillation
(ENSO), caused by seasonal latitudinal changes in the SPA that produce high variability
and precipitation in Chilean mid-latitudes. Sediment cores were retrieved in two bays,
Guanaqueros and Tongoy, for geochemical analyses including: sensitive redox trace
elements, biogenic opal, total organic carbon (TOC), diatoms, stable isotopes of organic
carbon and nitrogen. The results suggest a main dry phase of high primary productivity
concomitant with high fluxes of organic compounds to the bottom and suboxic-anoxic
conditions in the sediments. This period reached a maximum at cal BC ~4500, followed
by a continuous increase in wet conditions, low primary productivity and a more
oxygenated environment towards the present, being remarkably stronger in the last 2000
years. We suggest that this might be associated with greater El Niño frequencies or
similar conditions that increase precipitation, concomitantly with the introduction of
oxygenated waters to coastal zones by the propagation of equatorial origin waves.

Keywords: paleoproductivity, paleoredox, trace metals, diatoms, opal, organic carbon,
ENSO, Coquimbo, SE-Pacific





## 1. Introduction

The northern-central Chilean continental margin (18−30°S) has distinct characteristics in terms of topography, climate and oceanographic conditions, which modulate primary productivity and the chemical composition of the water column. This zone is characterized by several foci of high primary production (0.5-9.3 g C m$^{-2}$ d$^{-1}$; González et al., 1998; Daneri et al., 2000, Thomas et al., 2001) off Iquique (21°S), Antofagasta (23°S) and Coquimbo (30°S), resulting from the influx of nutrient-rich waters due to a semi-permanent upwelling forced by local winds. This productivity takes place close to the coast above the narrow continental shelf, allowing for the development of relevant fisheries and accounting for up to 40% of total annual catches in the Chilean margin (Escribano et al., 2004 and references therein).

Upwelling areas are recognized as special zones where the considerable amount of organic material produced in the photic zone causes large particle depositions on the bottom. The remineralization of this organic carbon promotes anoxic/suboxic environments where chemosynthetic and authigenic reactions occur. The distribution of chemical compounds present in the sediments and water column are driven by diagenetic reactions, involving metals and nutrients, related to the destruction of organic matter and thus are linked to the diagenetic sequence (oxic or anoxic). In the case of trace metals, these reactions result in soluble or insoluble phases, which are particular for each metal, depending on the oxidative state of the sediments and bottom waters (Chester, 1990; Guieu et al., 1998; Wells et al., 1998).

The high productivity observed in these boundary current ecosystems develops and maintains zones of low dissolved oxygen content known as oxygen minimum zones (OMZs). These naturally occurring regions are found in the upwelling areas of the Eastern Boundary Currents, developed along the North and South Pacific Ocean where their intensity, thickness, and temporal stability vary as a function of latitude (Helly and Levin, 2004, Ulloa et al., 2012). To the north (e.g. at 21°S) and off Peru, the OMZ is permanently present, can extend into the euphotic zone and, in the case of northern Chile and southern Peru, it shows no significant interface with the benthic environment due to the presence of a narrow continental shelf (Helly and Levin, 2004).

Although ecosystems in Eastern Boundary Currents represent less than 5% of the ocean's surface, they support regions of unique microbial processes that influence global element cycles. The nature of the chemical processes that are taking place in surface sediments and bottom waters establishes the enrichment or depletion of trace





elements within sediments. This could respond to different causes and is not always
directly related to the amount of organic carbon that has settled on the bottom.
Diagenetic reactions could modify the original extension of metal enrichment that
occurs in the upper sediment layer, reflecting the redox conditions during early
diagenesis in the sediments (Nameroff et al., 2002) and the environmental conditions of
the overlying water, i.e. oxygen content (Zheng et al., 2000; McManus et al., 2002;
Siebert et al., 2003), both intimately related to the biological pump.
Climatic trends in the last century point to the beginning of a global warming period,
mainly observed by a $CO_2$ increase in the atmosphere (Bolin, 1992; Sarmiento and
Gruber, 2002), and corroborated by studies on ice and coral growth records (Mosley-
Thompson et al., 2006; Schneider and Steig, 2008; Nurhati et al., 2009). Higher mean
temperatures produce the stratification of the oceans suppressing vertical nutrient
transport that sustains primary production, as observed after 1999 (Behrenfeld et al.,
2006). The El Nino/Southern Oscillation cycles (ENSO) are the most important
phenomena with a major impact on sea life – from phytoplankton to marine mammals–
and are closely related to changes in atmospheric variables. Their frequency seems to
have intensified since cal  BP 3200 (Marchant et al., 1999), while extreme variations
have been observed in the last 200 yrs. Heavy rainfall episodes normally occur during
strong El Niño conditions (Montecinos and Aceituno, 2003), which can also be
observed in sedimentary records in northern Chile (23°S; Garreaud and Rutllant, 1996;
Vargas et al., 2007). In addition, the deepening of nutrient-rich Equatorial Subsurface
Water (ESW) reduces primary productivity in this zone, considered the most productive
area in the global ocean promoting biogeochemical changes validated through organic
and inorganic sedimentary records (Muñoz et al., 2012 and references therein).
Our work focusses on past variations of environmental conditions and marine
productivity in recent sedimentary records from a transitional semi-arid ecosystem of
Central Chilean coast, susceptible to oceanographic and climatic forcing. The study area
(Fig. 1) provides an adequate platform to observe environmental variability within
different time scales. We suggest that the low continental inputs allow us to clearly
identify wet/dry as well as high/low export primary production periods because
sediments in this area receive the organic flux mainly from the upwelling foci in the
area. Moreover, the organic flux is restricted to the narrow shelf where low oxygen
content has been detected promoting the preservation of inorganic (trace metals) and
organic proxies.





## 2. Study area

The Coquimbo region constitutes a border area between the most arid zones of northern Chile (Atacama Desert) and the more mesic Mediterranean climate of central Chile (Montecinos et al., 2015). Here, the shelf is narrow and several small bays trace the coast line. Favorable winds throughout the year promote an important upwelling center in the southern part of the region (Lengua de Vaca Point), developing high biomass along a narrow coastal area (Moraga-Opazo et al., 2011), and reaching maximum concentrations of ~20 mg m$^{-3}$ (Torres et al., 2009). High variability in pigment concentrations have been attributed to high variability in physical processes, which modulate Chl-*a* distribution along the coast. The bays of Tongoy and Coquimbo register a bipolar circulation consisting of counter rotating gyres: clockwise to the south of the bays and counterclockwise to the north (Moraga-Opazo et al., 2006; 2011). Additionally, large squirts of photosynthetic pigments are observed from the coast to the open ocean with large eddies acting as retention zones counteracting the transport of Chl-*a* offshore (Rutllant and Montecino, 2002; Marín et al., 2003).

The Tongoy and Guanaqueros bays are located at the southern edge of a broad embayment between small islands in the north (29°S; Choros, Damas and Chañaral) and Lengua de Vaca Point in the south (30°S) (Fig. 1), protected from predominant southerly winds. Tongoy Bay is a narrow marine basin (10 km at its maximum width) with a maximum depth of ~100 m. To the northeast lies Guanaqueros Bay, a smaller and shallower basin. Recent oceanographic studies in this zone suggest the relevance of low dissolved oxygen water intrusions from the shelf to shallow waters (Fig. 2), which seem related to sea level decreases resulting from local wind annual cycles at a regional meso-scale (Gallardo et al., 2017)). Furthermore, in the shallow waters of Tongoy Bay, the high primary productivity results in high TOC in the water column that allows for the deposition of fine material on the bottom; TOC increases concurrently with the periods of low oxygen conditions (Fig. 3; Muñoz et al., unpublished data). The spatial and temporal variability of this process is still under study.

Sedimentological studies are scarce in the northern-central Chilean shelf. A few technical reports indicate that sediments between 27°S and 30°S are composed of very fine sand and silt with relatively low organic carbon content (<3 and ~5%), except at very limited coastal areas where organic material accounts for around ~16% (Muñoz, unpublished data; FIP2005-61 Report, www.fip.cl). Coastal weathering is the main source of continental input due to scarce river flows and little rainfall in the zone (0.5 to



~20 mm yr$^{-1}$; https://es.climate-data.org/location/940/, Fig.1). Freshwater discharges are
represented by creeks, which receive the drainage of the coastal range forming wetland
areas in the coast and even small estuaries, as observed in Tongoy and Pachingo. These
basins cover ~300 and 487 km$^2$, respectively. The water volume in the estuaries is
maintained by the influx of seawater mixed with groundwater supply. No surface flux to
the sea is observed. Therefore, freshwater discharge occurs only during high rainfall
periods in the coastal zone (DGA, 2011), which normally takes place during El Niño
years when higher runoff has been recorded in the area (Valle-Levinson et al., 2000)
normally during the June-July-August quarter (Garreaud et al., 2009). In this scenario,
marine sediments are often highly influenced by primary production in the water
column. This organic input is relevant for the sediment composition and, thus,
sedimentary records could clearly reveal the variability in primary production and
hence, the oceanographic conditions over the shelf, which ultimately respond to major
atmospheric patterns.

**3. Materials and methods**
**3.1. Sampling**
Sediment cores were retrieved from two bays in the Coquimbo region: Bahía
Guanaqueros (core BGGC5; 30°09' S, 71°26' W; 89 m water depth) and Bahía Tongoy
(core BTGC8; 30°14' S, 71°36' W; 85 m water depth) (Fig. 1.), using a gravity corer
(KC-Denmark) in May 2015, on board the L/C Stella Maris II owned by the
Universidad Católica del Norte. The length of the cores was 126 cm for BGGC5 and 98
cm for BTGC8. Both cores were cut along the main axis and a general visual
characterization was done. Different textures and color layers were identified using the
Munsell color chart.
Subsequently, the cores were sliced into 1-cm sections and subsamples were separated
for grain size measurements, magnetic susceptibility, trace elements, biogenic opal, C
and N stable isotope signatures ($\delta^{13}$C, $\delta^{15}$N), and TOC analyses. The samples were first
kept frozen (–20° C) and then freeze-dried before laboratory analyses.
The magnetic signal indicates the concentrations and compositions of magnetic
minerals and is usually used combined with others detrital proxies -as grain size- to
establish changes in sedimentary processes closely controlled by climatic conditions.
We considered redox trace elements measurements that respond to local hypoxia (U,
Mo and Re) as well as nutrient-type elements, which follow the organic fluxes to the



sediments (Ba, Ni Cu, P). Additionally, we measured Fe and Mn relevant in adsorption-
desorption and scavenging processes of dissolved elements in the bottom water and
sediments. We also measured Ca, K and Pb that have relevant continental sources by
coastal erosion, weathering and eolian transport, which is also true for Fe and Mn.
Besides, Ca accumulation within the sediments depends on the carbonate productivity
and dissolution, being used as paleoproductivity proxy (Paytan, 2008; Govin et al.,
2012). We use Al as a normalizing parameter for enrichment/depletion of elements due
to its conservative behavior. The crustal contribution and the elements are presented as
metal/Al ratios. The authigenic enrichment factor of elements was estimated according
to: $EF = (Me/Al)_{sample} / (Me/Al)_{detrital}$; where $(Me/Al)_{sample}$ is the bulk sample metal (Me)
concentration normalized to Al content and the denomination "detrital" indicates a
lithogenic background (Böhning et al., 2009). Detrital concentrations ($[Me]_{detrital}$ and
$[Al]_{detrital}$) were established considering the local TM abundance, which is more accurate
than using mean Earth crust values (Van der Weijden, 2002). We used the average of or
element concentrations at the surface sediments (0−3 cm) of Pachingo wetland (Table

208 1).

Diatoms and siliceous microfossils were identified and counted; jointly with biogenic
opal content, they constitute our proxies of siliceous export production. Pollen grains
were also identified and counted, and used to identify wet and dry environmental
conditions. TOC and stable isotopes of organic matter were used to identify the
variability of organic fluxes to the bottom and establish biogeocheemical changes in the
organic matter remineralization.
$^{14}C$ and $^{210}Pb$ were analyzed at selected sediment levels and foraminifera species were
selected for radiocarbon analyses (Table 2).

### 218 3.2, Geochronology ($^{210}Pb$ and $^{14}C$)

$^{210}Pb$ activities were quantified through alpha spectrometry of its daughter $^{210}Po$ in
secular equilibrium with $^{210}Pb$, using $^{209}Po$ as a yield tracer (Flynn, 1968). The chemical
procedure considered a total digestion of the sediment samples and then autoplated onto
silver disks at ~75°C for 3 three hours in the presence of ascorbic acid. The $^{210}Po$
activity was counted in a CANBERRA QUAD alpha spectrometer, model 7404, until
the desired counting statistics was achieved (4−10% $1\sigma$ errors) in the Chemical
Oceanography Laboratory of Universidad de Concepción. $^{210}Po$ activity –assumed to be
in secular equilibrium with $^{210}Pb$– was calculated using the ratio between natural





radionuclide and the tracer, which is multiplied by the activity of the tracer at the time
of plating. The period elapsed between plating and counting produces $^{210}$Po decay (half-
life: 138 days) and between sampling and plating $^{210}$Pb decay (half-life: 22.3 yr);
counting was corrected to these elapsed times even when there was a short time period
between the collection date and the time of sample analysis (less than one year). Ages
were estimated using the inventories of the activities in excess ($^{210}$Pb$_{xs}$, unsupported),
based on the Constant Rate of Supply Model (CRS, Appleby and Oldfield, 1978).
Unsupported activities were determined as the difference between $^{210}$Pb and $^{226}$Ra
activities measured in some sediment column intervals. $^{226}$Ra was measured with a
gamma spectrometry at the Laboratoire Géosciences of the Université de Montpellier
(France). Standard deviations (SD) of the $^{210}$Pb inventories were estimated propagating
counting uncertainties (Bevington and Robinson, 1992) (Table S1, supplementary data).
Radiocarbon measurements were performed on a mix of planktonic foraminifera species
in core BGGC5 whereas the benthic foraminifera species *Bolivina plicata* was selected
for core BTGC8 (Table 2). Freeze-dried sediment was washed over a 63 µm mesh-size
sieve and dried after washing at 50°C. At least 2 mg of mixed planktonic foraminifera
were picked from the 125−250 µm fraction. The samples were submitted to the National
Ocean Sciences AMS Facility (NOSAMS) of the Woods Hole Oceanographic
Institution (WHOI). The Fraction Modern (Fm) was corrected by the $\delta^{13}$C value, and
ages were calculated using 5568 (yrs) as the half-life of radiocarbon. The ages were
converted to calendar years before present using the calibration curve
Calpal2007_HULU (Jöris and Weninger, 1998), and considering an age reservoir effect
that compares the $^{14}$C value in the foraminifera core with ages estimated from the CRS
model. The deviation from the global mean reservoir age (DR) was estimated
subtracting the value corresponding to the age obtained with the CRS model in the
marine 13.14c curve (Reimer et al., 2013) from the apparent $^{14}$C age measured at depths
of 5 and 10 cm for cores BTGC8 and BGGC5, respectively (Sabatier et al., 2010; Table
3). The time scale was obtained according to the best fit of $^{210}$Pb$_{xs}$ curves and $^{14}$C points,
using the CLAM 2.2 software and Marine curve 13C (Reimer et al, 2013), and
considering a global reservoir of 441 years (Fig. 4).

**3.3. Geophysical characterization**





Magnetic susceptibility (SIx10$^{-8}$) was measured with a Bartington Susceptibility Meter
MS2E in the Sedimentology Laboratory at Centro Eula, Universidad de Concepción.
Mean values from three measurements were calculated for each sample.
Grain size was determined using a Mastersizer 2000 laser particle analyzer, coupled to a
Hydro 2000−G Malvern in the Sedimentology Laboratory of Universidad de Chile.
Skewness, sorting and kurtosis were evaluated using the GRADISTAT statistical
software (Blott and Pye, 2001), which includes all particle size spectra.

**3.4. Trace elements analysis**
Trace element analyses were performed by ICP-MS (Inductively Coupled Plasma-Mass
Spectrometry) and carried out at Université de Montpellier 2, France (OSU
OREME/AETE regional facilities), using an Agilent 7700x. About 50 mg of samples
and geochemical reference materials (UBN, BEN and MAG1) were dissolved twice
through the conventional digestion method using a concentrated HF-HNO$_3$-HClO$_4$ mix
(1:1:0.1) in Savillex screw-top Teflon beakers at 120°C, on a hot plate during 48h.
Following digestion, the samples were subjected to three evaporation steps in order to
remove fluorine. Shortly before analysis, samples were dissolved in 2 ml of
concentrated HNO$_3$ and transferred to 20 ml polypropylene bottles. Final sample
preparation was undertaken by dilution with ultrapure water to a sample-solution weight
ratio of 1: 4000-5000 and the addition of a known weight of internal standard solution
consisting of 1 ppb of In and Bi. Internal standardization used ultra-pure solution
enriched in In and Bi, both elements whose natural abundances in geological samples
do not contribute significantly to the added internal standard. This is used to deconvolve
mass-dependent sensitivity variations of both matrix and instrumental origin, occurring
during the course of an analytical session.
Sample introduction uses a peristaltic pump, a micro-nebulizer and a cooled double-
pass Scott type spray chamber. The uptake time (typically 45 s) is set to facilitate stable
analyte signals prior to a 120 seconds analysis for each sample. Elements with an
atomic mass lower than 80 were analyzed in collision mode using He; heavier elements
were analyzed in no-gas mode. A wash out procedure consisting of 60 seconds with
HNO$_3$ 10% and 120 seconds with 2% HNO$_3$ has been found appropriate to achieve
instrument blank level. The total time for analysis of a single sample solution is *c.* 3
minutes. Mean concentrations for the analyzed samples were determined by external
calibrations prepared daily from multi- and mono-elemental solutions, with



concentrations in the range of 0.05−10 ppb for trace elements and of 1−10 ppm for
major elements (Ca, K). Polyatomic interferences were controlled by running the
machine at an oxide production level <1%. Typical analytical precisions attained by this
technique are generally between 1% and 3%, relative standard deviation. Accuracy has
been assessed with an analysis of international reference materials and results show
agreement generally better than ±5% with reference values.

**3.5. TOC and stable isotopes**
TOC and stable isotope ($\delta^{15}$N and $\delta^{13}$C) analyses were performed at the Institut für
Geographie, Friedrich Alexander Universität (FAU) Erlangen-Nürnberg, Germany. Dry
material was placed into tin and silver capsules for N and C analyses respectively, and
combusted at 1060° C in a continuous helium flow in an elemental analyzer (NC2500,
Carlo Erba), in the presence of chromium oxide and silvered cobalt oxide. The resulting
gases, were passed over copper wires at 650° C to reduce nitrogen and excess oxygen.
Thereafter, water vapor was trapped with $Mg(ClO_4)_2$ and the remaining gases ($N_2$ and
$CO_2$) were separated in a gas chromatography column at 45° C. $N_2$ and $CO_2$ were
passed successively via a ConFloII interface into the isotope-ratio-mass spectrometer
(Delta Plus, Thermo-Finnigan) and isotopically analyzed. Carbon and nitrogen contents
were determined from the peak-area-versus-sample-weight ratio of each individual
sample and calibrated with the elemental standards cyclohexanone-2,4-
dinitrophenylhydrazone ($C_{12}H_{14}N_4O_4$) and atropine ($C_{17}H_{23}NO_3$) (Thermo Quest). A
laboratory-internal organic standard (Peptone) with known isotopic composition was
used for final isotopic calibrations. Stable isotope ratios are reported in the δ notation as
the deviation relative to international standards (Vienna Pee Dee Belemnite for $\delta^{13}$C and
atmospheric $N_2$ for $\delta^{15}$N), so $\delta^{13}$C or $\delta^{15}$N = [(R sample/R standard) − 1] x $10^3$, where R
is $^{13}C/^{12}C$ or $^{15}N/^{14}N$, respectively. Typical precision of the analyses was ±0.1‰ for
$\delta^{15}$N and $\delta^{13}$C.

**3.6. Biogenic opal**
Biogenic opal was estimated following the procedure described by Mortlock and
Froelich (1989) with a slight modification, which consists in extracting 50 mg of
sediment with 1 M NaOH (instead of 2 M $Na_2CO_3$) at 85°C for 5 hours. Extraction and
analysis by molybdate-blue spectrophotometry (Hansen and Koroleff, 1999) were



conducted at the laboratories of Marine Organic Geochemistry and Paleoceanography,
University of Concepción, Chile. Values are expressed as biogenic opal by multiplying
the Si (%) by 2.4 (Mortlock and Froelich, 1989). Analytical precision was ± 0.5%.
Accumulation rates were determined based on sediment mass accumulation rates and
amount of opal at each core section in %.

**3.7. Diatoms and siliceous microfossils**

Smear slides for qualitative abundances of siliceous microfossils were carried out every
centimeter following the Ocean Drilling Program (ODP) protocol described by
Mazzullo et al. (1988.) To determine the quantitative abundance of siliceous
microfossils (diatoms, silicoflagellates, sponge spicules, crysophyts and phytoliths), ~
0.5 g of freeze-dried sediment was treated according to Schrader and Gersonde (1978).
Samples were chosen every ~4, 8 and 12 cm for BGGC5 and at an average of 6 cm for
BTGC8. Permanent slides were prepared by placing a defined sample volume (0.2 ml)
onto microscope slides that were then air-dried and mounted with Naphrax mounting
medium (refraction index =1.3). Siliceous microfossils were identified and counted
under an Olympus CX31 microscope with phase contrast. 1/5 of the slides were counted
at 400X for siliceous microfossils and one transect at 1000x was counted for
*Chaetoceros* resting spores. Two slides per sample were counted; the estimated
counting error was 15%. Total diatom abundances are given in valves $g^{-1}$ of dry
sediments.

**3.8. Pollen**

Sample preparation for pollen analysis was conducted following the standard
methodology for sediment samples (Faegri and Iversen, 1989), which includes
deflocculating with 10% KOH, carbonate dissolution with a 5% HCl treatment, silica
dissolution with 30% HF, and cellulose removal *via* acetolysis reactions. Samples were
mounted with liquid glycerol and sealed permanently with paraffin wax. Pollen
identification was conducted under a stereomicroscope 400 fold magnification with the
assistance of the Heusser (1971) pollen catalogue. A total of 100-250 terrestrial pollen
grains were counted on each sample depending on their abundance. Pollen percentage
for each taxon was calculated from the total sum of terrestrial pollen. The percentage of
aquatic pollen and fern spores was calculated based on the total terrestrial sum plus the
respective group. Pollen percentage diagrams were generated using the Tilia software



(E. Grimm, Illinois State Museum, Springfield, IL. USA). The diagram was divided into
"zones" based on the identification of the most important changes in pollen percentage
and assisted by a cluster ordination (CONISS) performed by the same software.
We further summarize pollen-based precipitation trends by calculating a Pollen
Moisture Index (PMI), which is defined as the normalized ratio between Euphorbiaceae
(wet coastal scrubland) and Chenopodiaceae (arid scrubland). Thus, positive (negative)
values of this index indicate the relative expansion (reduction) of coastal scrubland
under relatively wetter (drier) conditions.

**4. Results**
**4.1. Geochronology**
$^{210}Pb_{xs}$ (unsupported activity) was obtained from the surface down to 8 cm depth in the
two cores, with an age of ~cal AD 1860 at 7 cm in both of them (Table S1). Greater
surface activities were obtained for core BGGC5 ($13.48 \pm 0.41$ dpm g$^{-1}$) compared to
core BTGC8 ($5.80 \pm 0.19$ dpm g$^{-1}$), showing an exponential decay with depth (Fig. 4).
A recent sedimentation rate of $0.11 \pm 0.01$ cm yr$^{-1}$ was estimated.
The age reservoir estimation was similar for both cores, 441 and 442 years for BGGC5
and BTGC8, respectively (Table 3). The age model provided a maximum age of cal BC
6300 for core BGGC5, and cal BC 5760 for core BTGC8 (Fig. 4). A mean
sedimentation rate of 0.02 cm yr$^{-1}$ was estimated for core BGGC5, with a period of
relative low values (0.01 cm yr$^{-1}$) between cal BC 4000 and 2000. For BTGC8,
sedimentation rates were less variable and around 0.013 cm yr$^{-1}$ in the entire core.

**4.2. Geophysical characterization**
The sediments retrieved from the bays showed fine grains in the range of very fine sand
and silt in the southern areas. There, the grain size distribution was mainly unimodal,
very leptokurtic, better sorted and skewed to fine grain when compared to sediments
from the northern areas. Sediment cores obtained from the northern areas were sandy
(coarse sand and gravel), with abundant calcareous debris. Longer cores of soft
sediment were retrieved at the southern areas (BGGC5 and BTGC8), where the silty
component varied between 40 % and 60 % (Fig. 1 and 5a,b). The clay component was
very low at both cores (<2%). The sediment's color ranged from very dark grayish
brown to dark olive brown (2.5Y 3/3−3/2) at Guanaqueros Bay (BGGC5) and from dark
olive gray to olive gray (5Y 3/2−4/2) at Tongoy Bay (BTGC8). Visible macro-remains





(snails and fish vertebrae) were found and weak laminations were identified at both
cores. The magnetic susceptibility showed higher values close to the surface, up to 127
SI x$10^{-8}$ at BGGC5 and relative lower values (85 SI x$10^{-8}$) at BTGC8. At greater depths,
however, the values were very constant, around 5−8 x$10^{-8}$ SI at BGGC5 core and
around 12−20 x$10^{-8}$ SI at BTGC8 core. In both cores, susceptibility increases
substantially after cal AD 1800 (Figs. 5a, 5b). Lower bulk densities were estimated at
the core BGGC5 (0.7−0.9 g cm$^{-3}$) compared to the core BTGC8 (>1 g cm$^{-3}$) (Fig. 5a,
5b). In accordance with this, the mean grain size was 60−80 µm at Guanaqueros Bay
(BTGC8), compared to 50−60 µm at Tongoy Bay (BGGC5). Both cores were
negatively skewed, with values of -1 to -1.2 at BGGC5, and -1 to -2.5 at BTGC8. Minor
increases towards coarser grain size were observed in the last 2000 years, especially at
Tongoy Bay (BTGC8). In both cases, grain size distributions were strongly leptokurtic.
Ca/Fe ratio also diminished in time, except at core BTGC8 where it was only observed
during the last ~3000 years. The diminishing of the Ca/Fe ratio is due to a decrease in
Ca content mainly but also because of a slight increase in Fe within the sediments (Figs.
6a, 6b).

**4.3. Biogenic components**
**4.3.1. Siliceous microfossils and biogenic opal**
Total diatom abundance fluctuated between 5.52 x$10^5$ and 4.48 x$10^7$ valves g$^{-1}$ in core
BGGC5. Total diatom abundance showed a good correlation with biogenic opal content
at BGGC5 ($R^2$ =0.52, P<0.5), with the highest values from 72−74 cm to the bottom of
the core, corresponding to cal BC 3390−3790. In contrast, diatom abundance and
biogenic opal were much lower in core BTGC8 (< 2 ×$10^5$ valves g$^{-1}$ and <3%,
respectively). Here, the siliceous assemblage was almost completely conformed by
*Chaetoceros* resting spores (RS) (Fig. 7).
A total of 135 and 8 diatom taxa were identified in cores BGGC5 and BTGC8
respectively, where the core BTGC8 registered very low abundances of diatoms. In
general, diatoms were the most important assemblage of siliceous microfossils (96%),
followed by sponge spicules (3%). The contribution of phytoliths and chrysophyte cysts
was less than 2% in core BGGC5. *Chaetoceros* (RS) dominated the diatom assemblage
(~90%; Fig. 7), and included the species *C. radicans*, *C. cinctus*, *C. constrictus*, *C.*
*vanheurckii*, *C. coronatus*, *C. diadema*, and *C. debilis*. Other species recorded of
upwelling group (mainly in core BGGC5) were: *Skeletonema japonicum,* and



*Thalassionema nitzschioides* var. *nitzschioides* (Table S2). Freshwater diatoms (*Diploneis papula, Cymbella tumida, Fragilaria capucina, Diatoma elongatum)* and non-planktonic diatoms (*Cocconeis scutellum*, *C. costata* and *Gramatophora angulosa*) accounted for ~0.1−5%; while the group of coastal planktonic diatoms accounted for ~0.3− 6% of the total assemblage. The main planktonic diatoms were (*Rhizosolenia imbricata*, and *Thalassiosira eccentrica*). Oceanic-warm diatoms (*Roperia tesselata*, *Th. nitzschioides var inflatula)* and the tycoplanktonic diatom group were rare with less than 1%.

### 4.3.2. TOC and stable isotopes distribution

Consistent with opal and diatoms, core BGGC5 showed higher values of TOC (between 2 % and 5 %) compared with less than ~1.5 % in core BTGC8 (Fig. 5a,b). Furthermore, $\delta^{13}$C was slightly higher at core BTGC8 (-20 ‰ to -21 ‰) compared with core BGGC5 (-21 ‰ to -22 ‰), the former also showing slightly increased values of $\delta^{15}$N from the deeper sections to the surface of the core (<7 ‰ to >10 ‰). This increase was less evident at core BGGC5, with values of ~9 ‰ at depth to >10 ‰ on the surface (Fig. 5a,b). Diminishing TOC contents was related to slightly higher $\delta^{13}$C values (~ -20 ‰) in both cores.

### 4.3.3. Pollen record

Initial surveys on core BTGC8 (Tongoy Bay) revealed extremely low pollen abundances which impeded further palynology work. A complete pollen analysis was only conducted for core BGGC5 (Guanaqueros Bay). The pollen record of core BGGC5 consisted of 29 samples shown in Figure 8. The record was divided in five general zones following visual observation of changes in the main pollen types and also assisted by the cluster analysis CONISS.

Zone BG-1 (cal BC 6250 – cal BC 5650): This zone is dominated by the herbaceous taxa Chenopodiaceae, Ast. *Leucheria-type*, Ast. Asteroideae, Apiaceae with relatively high values of the wetland genus *Typha spp*.

Zone BG-2 (cal BC 5650 – cal BC 4600): This zone is also dominated by Chenopodiaceae, Ast. *Leucheria-type* and Ast. Asteroideae. In addition, other non-arboreal elements such as Ast. *Ambrosia-type*, Poaceae, Brassicaceae and *Chorizanthe spp.* expand notably.





Zone BG-3 (cal BC 4600 – cal BC 1450): This zone is marked by a steady decline in
Chenopodiaceae and Ast. *Leucheria-type*, and by the expansion of several other
herbaceous elements, such as Euphorbiaceae, Ast. *Baccharis-type* and Brassicaceae.
Zone BG-4 (cal BC 1450 – cal AD 1800): This zone is mostly dominated by Ast.
Asteroideae, and marked by the decline of Chenopodiaceae and Ast. *Leucheria-type*.
Other coastal taxa such as Euphorbiaceae, Ast. *Baccharis-type*, Ast. Chichorioideae,
*Quillaja saponaria*, Brassicaceae and *Salix spp*. also expand in this zone.
Zone BG-5 (cal AD 1800 – cal AD 2014): The upper portion of the record is
dominated by Ast. Asteroideae and Poaceae, and marked by increments of
Geraniaceae, Ast. Mutisieae, Myrtaceae and *Q. saponaria*. Additionally, this zone
includes introduced pollen types such as *Rumex* and *Pinus*. The latter is not shown in
the diagram of Figure 8 because its abundance was minimal.
The most distinctive change revealed in core BGGC-5 is a long-term decrease in
Chenopodiaceae and increments in Euphorbiaceae and Ast. Asteroideae. Along with
this trend, a later expansion of several other representatives of the coastal shrub land
starts at around cal BC 4600.

**4.4. Trace element distributions**
Trace element distributions are shown in figures 6a and 6b for Guanaqueros (BGGC5)
and Tongoy Bays (BTGC8), respectively. Trace metals sensitive to the presence of
oxygen (U, Re, Mo) showed increasing metal/Al ratios from the base of the core (cal
BC ~6300) until cal BC 4500 in core BGGC5. After this maximum, ratios presented a
slight increase towards the beginning of the recent era (cal AD 1) followed by a sharp
decrease until present. Similarly, the metal ratios in the core BTGC8 increase over
time, yet the maximum was observed at cal AD 1000. The exception of this trend was
Mo which exhibited maximum values until cal BC 4000 and then a steady decrease
towards the present. Additionally, metal/Al values were higher at core BGGC5. Iron
displayed a clear increase around cal BC 2000 at core BGGC5, and at core BTGC8
was observed a slight increase after cal BC ~3000. Manganese did not show a clear
trend.
A second element group (metal/Al ratios), including Cd, Ni and P (related to primary
productivity and organic fluxes), showed a similar pattern than Mo/Al towards the
bottom of core BGGC5, i.e. the highest values around cal BC 4500 and a constant
reduction towards the present. A third group, consisting of Ba, P and Ca, exhibited a



less clear pattern. The Cd/Al and Ni/Al ratios in core BTGC8 showed only slightly
decreasing values, and the maximum values were very low compared to the BGGC5
core. The same pattern is observed for other elements. Metal/Al ratios for Ba, Ca and
P were lower and presented a long-term decreasing pattern towards the present.
An exception to the previously described patterns was Cu/Al, which peaked at cal BC
2000 and showed a conspicuous increase in the past ~150 years. This was also
observed at core BTGC8, but with lower concentrations than at core BGGC5.

**5. Discussion**
**5.1. Sedimentary composition of the cores: terrestrial *versus* biogenic inputs**
The sediments in the southern zones of the bays constitute a sink of fine particles
transported from northern areas and the shelf (Fig. 5a, 5b), responding to the water
circulation in the Guanaqueros and Coquimbo Bays described as bipolar, i.e. two
counter-rotating gyres which are counterclockwise to the north and clockwise to the
south (Valle-Levinson and Moraga, 2006). This is the result of the wind and a
coastline shape delimited by two prominent points to the north and south. In the case
of Tongoy Bay (the southernmost bay of the system), circulation shows a different
pattern due to its northern direction compared to Guanaqueros Bay, which opens to the
west. The cyclonic recirculation in Tongoy Bay seems to be part of a gyre larger than
the Bay's circulation (Moraga et al., 2011). This could explain differences in sediment
particle distribution and composition between the bays. At Tongoy Bay, there are less
organic carbon accumulation (< 2 %), siliceous microfossils and pollen (Figs. 5, 7 and
8). Similarly, in Guanaqueros Bay TOC contents are only slightly higher (> 2 %),
especially before cal BC 1000 (~ 4 %). However, sediments there contain enough
microfossils to establish differences in primary productivity periods and also provide a
pollen record evidencing the prevailing environmental conditions.
The stable isotopes measured in the study area were in the range of marine
sedimentary particles for southern oceans at low and mid latitudes ($\delta^{13}C$; -20 ‰ – -24
‰; Williams 1970; Rau et al., 1989; Ogrinc et. al. 2005), and slightly lower than the
TOC composition at the water column (-18 ‰, Fig. 3). This suggests that the organic
particles that settle on the bottom are a more refractory material (C/N: 9−11),
remineralized during particle transportation and sedimentation. This results in lighter
isotopic compositions, especially at core BTGC8. Besides, the $\delta^{15}N$ and $\delta^{13}C$ of settled
particles are more negative at surface sediments due to a preferential degradation of





molecules rich in $^{13}$C and $^{15}$N, resulting in more negative values and higher C/N ratios
at sediments than in suspended particles (Fig. 3, 5a, 5b). However, this is also due to
the stronger diagenetic reactions observed near the bottom layer (Nakanishi and
Minagawa, 2003), indicating that the particles' diagenesis and time of transportation
over the shelf should be a relevant factor in this differentiation. Thus, these sediments
are composed by winnowed particles transported by water circulating over the shelf,
and the isotopic variations should not establish clearly the contribution of terrestrial
inputs.
Magnetic susceptibility (MS) measurements revealed lower values throughout both
cores (BGGC5: 5 – 8 x10$^{-8}$ SI; BTGC8: 12 – 20 x10$^{-8}$ SI), except at dates after ~ cal
AD 1800, when the susceptibility increases substantially to values similar to those
observed in the Pachingo wetland (40 – ~200 x10$^{-8}$ SI; unpublished data) on the
southern side of Tongoy Bay. Magnetite has strong response to magnetic fields and its
concentration is considered proportional to magnetic susceptibility (Dearing, 1999).
Besides, mineral post-depositional transformations (alteration of magnetite minerals)
and dilution by biogenic components (carbonates, silicates) should also be relevant in
the MS intensity in zones with high organic accumulation rates (Hatfield and Stoner,
2013). However, this is not expected to be the case for our cores and the MS should be
mainly accounting for the source of the particles. High MS measurements in the bay
sediments would be associated with relevant terrestrial input. The area is surrounded
by several creeks that are only active during major flooding events, with greater
impacts in Tongoy Bay compared to Guanaqueros Bay. An important increment in the
contribution of terrestrial material has occurred in Tongoy Bay in recent times (Ortega
et al., in review), which is diluting organic proxy records and increasing the grain size.
Our records indicate a slight increase in mean grain size at both bays, supported also
by a slight decrease in Ca/Fe ratio indicating more Fe input from continental erosion
(Fig. 5a, 5b).
Recent information indicates that during the intensification of southern winds the
upwelling develops a nutrient-rich and low-oxygen flow within the bay's southern
areas (Gallardo et al., 2017), which promotes phytoplankton blooms and low oxygen
events. Decreasing concentrations of Ca from the deepest part of both cores to the
surface was interpreted as decreasing primary productivity (Keshav and Achyuthan,
2015; Sun et al., 2016), but higher concentrations were measured in core BGGC5
compared with core BTGC8, where more terrestrial influence is being suggested. The





slight increase of K/Ca ratio in time, from bottom to the surface, should also be
interpreted as a slight increase in continental input, since K is related to siliciclastic
material from coastal erosion, fluvial and groundwater inputs. However, the variation
of Ca was larger (Fig.6a, 6b), resulting in higher K/Ca ratios at the surface. This
indicating that the continental input has not changed much in time but rather the
primary productivity has decreased (Fig. 5a, 5b).

**5.2. Temporal variability of proxies for primary productivity**
Several elements participating in phytoplankton growth are useful to interpret
variations in primary productivity in time, as they are preserved within the sediments
under suboxic-anoxic conditions. This produces enrichment over crustal abundance,
which distinguishes them from continental inputs. The presence of free dissolved
sulfides produced by sulfate reduction reactions in the diagenesis of organic matter is
relevant for metal precipitation in pore water (Calvert and Pedersen, 1993; Morse and
Luther, 1999). At the same time, organic matter remineralization releases ions to the
pore water where they could form organic complexes and insoluble metal sulfides.
Conversely, they could be incorporated into pyrite as Cd, Ni and Cu, showing
different degrees of trace metal pyritization (Huerta-Diaz and Morse, 1992). Ca, Sr,
Cd and Ni profiles suggest a lower proportion of organic deposition in time (Fig. 6a,
6b), consistent with the slight reduction of TOC content in time observed in the
sediments (Figs. 5a, 5b), and concomitantly with other elements related to organic
fluxes to the bottom and primary productivity. In the case of Ba, it is actively
incorporated into phytoplankton biomass or adsorbed onto Fe oxyhydroxides,
increasing the Ba flux towards the sediments, where it is also released during organic
matter diagenesis. Ba is precipitating in microenvironments where Ba-sulfate reaches
supersaturation (Tribovillard et al., 2006 and references therein), but it is dissolved in
suboxic-anoxic environments or where sulfate is significantly depleted (Torres et al.,
1996; Dymond et al., 1992). Therefore, it is better preserved in less anoxic
environments with moderate productivity, expected to be the case of our study site
(Gross Primary Productivity =0.35 to 2.9 g C m$^{-1}$d$^{-1}$; Daneri et al., 2000). Hence, the
slight increase of Ba from cal BC 4000 (Fig. 6a) to the present should rather be the
response to a less anoxic environment than to an increase in primary productivity. This
is consistent with the reduction in TOC and other nutrient-type elements (Ni, Sr, Ca,
Cd), and results in a low negative correlation with TOC (-0.59; Table 4) due to the Ba





remobilization in anoxic conditions after high organic material deposition during cal
BC 4000−4500. On the other hand, P distribution showed a trend similar to that of
TOC and other elements related to organic fluxes to the bottom, although with a lower
correlation. The accumulation of P depends on the deposition rate of organic P (dead
plankton, bones and fish scales) to the bottom, and is actively remineralized during
aerobic or anaerobic bacterial activity. Dissolved P diffuses towards the water column
where part of it could be adsorbed onto Fe oxides that maintain this element within the
sediments. P is buried during a continued sedimentation process and could be released
to the pore water under anoxic conditions, when oxides are reduced, creating the
environmental conditions for phosphorite and carbonate-fluorapatite precipitation.
Normally, this takes place in sites with high sedimentation rates and high organic
matter fluxes to the bottom (Filippelli, 1997; Cha et al., 2005), which was not the case
for our study area ($<0.02$ cm $yr^{-1}$). In spite of this difference, P and TOC showed a
decreasing trend towards the present, suggesting reducing flux of organic matter over
time, which was also observed for Ni and Cd distributions. Alternatively, it could be
explained by the increased remineralization of the organic material settled on the
bottom (Figs. 6a, 6b).
Productivity reconstructions were based on diatom relative abundances and biogenic
opal content only in core BGGC5, since core BTGC8 registered valve counts that
were too low (<1% in relative diatom abundance). However, at both cores diatom
assemblages were represented mainly by *Chaetoceros resting spores*, which are used
as upwelling indicators, showing increased concentrations during periods of high
productivity and upwelling (Abrantes 1988, Vargas et al., 2004). In addition,
*Chaetoceros* resting spores are highly silicified and well preserved in coastal
sediments (Blasco et al., 1981). The downcore siliceous productivity based on opal
distribution (Fig. 7) distinguished three main periods of increased productivity: (1)
recent time cal AD 2014 – cal AD ~1800, (2) cal AD 1 – cal BC 2300 and (3) > cal
BC 4300. The mean opal accumulation rate in the second period was $11.8 \pm 4.8$ g $m^2$
$yr^{-1}$, when spicules and minerals (quarz, framboid pyrite) where abundant in smear
slides. During the third period, accumulation increased highly to ~$30.1 \pm 14.5$ g $m^2$ $yr^{-}$
$^1$, when the *Chaetoceros* spores were predominant, indicating upwelling intensification
and low spicules and minerals were observed in the slides. This is partially consistent
with the nutrient–type element distributions. Although the first period was too short,
high opal accumulation and high Cd/U ratios could also be observed, which increased



toward the present (mean opal value of 32.3 ± 22.4 g m$^2$ yr$^{-1}$). Similarly, Cu and Fe
also increased in recent times (Fig. 6a), contributing to fertilize the environment and
promoting primary productivity. The second period was not clearly identified in terms
of metals, except for Fe which shows a conspicuous increment in this period (Fig. 6a).
During the third period, all metal proxies showed primary productivity increases
before cal BC 4500, as indicated by opal accumulation within the sediments. In
anoxic-suboxic environments Cd/U ratios could vary between 0.2 and 2 (Nameroff et
al., 2002), the high concentrations of both elements reflect anoxic conditions but their
different behavior could result in variable Cd/U ratios in suboxic environments. Here,
the Cd and U accumulation on sediments resulted in high Cd/U ratios (>2; Fig. 7)
during periods with high opal accumulation in the cores, especially in the third period,
and even in core BTGC8; and lower ratios (< 1; Fig. 7) when opal was low, indicating
higher variations in the primary productivity in time with moderated changes in
oxygen conditions at the bottoms. Opal showed good correlations with Ni and Cd
(˜0.70; Table 4; Fig. 6a), all suggesting the relevance of bottom organic fluxes for
element accumulation within the sediments, and establishing a clear period of higher
primary productivity around cal BC 4500, when lowest oxygen conditions prevailed
(Fig. 7).

**5.3. Temporal variability of proxies for bottom water oxygenation**
U, Re and Mo distributions in core BGGC5 indicate that anoxic or suboxic conditions
were developed from cal BC 6000 to ~ cal AD 1−1000. After this period and towards
the present, however, a remarkable reduction in their concentration suggests a more
oxygenated bottom environment, concurrent with lower organic fluxes to the
sediments. The Re profile shows the influence of suboxic waters not necessarily
associated with increased organic matter fluxes to the bottom. Since this element is not
scavenged by organic particles, its variability is directly related to oxygen changes
(Calvert and Pedersen, 2007, and references therein). Additionally, it is strongly
enriched above crustal abundance in suboxic conditions (Colodner et al., 1993;
Crusius et al 1996), being >10 times in core BGGC5 (Fig. 9) before cal AD 1. In the
same way, U exhibits a similar pattern, and although organic deposition has an impact
on its distribution (Zheng et al., 2002), it also relates to changes in bottom oxygen
conditions. This is because its shift from a soluble conservative behavior to non-
conservative and insoluble solely depends on the redox potential change that occurs



near the Fe(III) reduction zone (Klinkhammer and Palmer, 1991). Molybdenum,
which showed high increases at cal BC 4500, also indicates the presence of sulfidic
conditions, as shown by the Re distribution highly enriched in anoxic environments
(Colodner et al., 1993), and by the reduction of Re(VII) to Re(IV) forming $ReO_2$ or
ReS (Calvert and Pedersen, 2007). Rhenium, U and Mo enrichment is relevant to
decipher the redox condition within the sediments, even in places with high lithogenic
input that could obscure the authigenic enrichment of other elements under similar
conditions (Crusius et al., 1996). In both places, the concentrations of these elements
showed values above the crustal abundance, especially in core BGGC5 (Fig. 9), with
Re and Mo becoming more enriched than U. This suggests that the presence of anoxic
conditions were stronger around cal BC 4500−5000. The most relevant enrichment
was observed for Cd (> 30, Fig 9), which could similarly indicate the sulfidic
condition within the sediments that allows Cd precipitation. It is also supported by Mo
enrichment, since its accumulation within the sediments is highly controlled by sulfide
concentrations (Chaillou et al., 2002; Nameroff et al., 2002; Sundby et al., 2004).
Something similar occurs in Tongoy Bay (core BTGC8), but trace metal
concentrations are lower for all elements and also for TOC, suggesting that it has
limited influence on metal accumulation within the sediments. Thus, these elements
suggest anoxic conditions within the sediments in both places around cal BC
4500−5000 (Fig. 6a, 6b). After this period, a second maximum but less intense anoxia
is observed at the beginning of the recent era (cal AD 1−1000), continuing with a
conspicuous oxygenation until present times. This interpretation based on the
distribution of U, Re and Mo complements the observations of nutrient-type elements
pointing both to oxygenation changes in time and to changes in organic fluxes to the
sediments. A less prominent accumulation of nutrient-type elements (Ni, Cd, Ba, Ca
and P) would indicate lower organic material deposition to the sediments but
promoting anoxic conditions within the sediments and lower sulfide content with time,
which are nevertheless high enough to sustain Mo accumulation until cal AD 1. After
that, a notorious decrease in Re, U and Mo accumulation was observed, suggesting
that the oxygenation of the bottom becomes relevant. This could also explain the
conspicuous increase of Cu/Al and Fe/Al in recent times due to the presence of oxides
(Fig. 6a, 6b). Apparently, a low level of dissolved Cu is maintained by the
complexation with organic compounds produced by phytoplankton and Cu adsorption
on Fe oxides (Peacock and Sherman, 2004; Vance et al., 2008; Little et al., 2014), with





both processes increasing Cu in the particulate phase over surface sediments. At our
study sites, Fe and Cu concentrations were higher in surface sediments, probably
related to an increase in Fe and Cu availability in the environment (Fig 6a, 6b). This
could be in turn associated with mining activities carried out in the area since the
beginning of cal AD 1900's. At present, the suboxic conditions within the bays result
from the influence of adjacent water masses with low oxygen contents, related to the
oxygen minimum zone (OMZ) (Fig. 2) centered at ~250 m. Upwelling promotes the
intrusion of these waters towards the bays, with strong seasonality. Transition times
develop in short periods by changes in wind directions and intensities along the coast.
Additionally, oceanic variability along the western coast of South America is influenced by
equatorial Kelvin waves on a variety of timescales, from intraseasonal (Shaffer et al.,1997)
and seasonal (Pizarro et al., 2002; Ramos et al., 2006) to interannual (Pizarro et al., 2002;
Ramos et al., 2008). Coastal-trapped Kelvin waves originating from the equator can propagate
along the coast, modify the stability of the regional current system and the pycnocline, and
trigger extratropical Rossby waves (Pizarro et al., 2002; Ramos et al., 2006; 2008). This
oceanographic feature will generate changes in oxygen content within the bays with
major impacts on redox sensitive elements in surface sediments; thus, the increased
frequency and intensity of this variability would result in a mean effect which is
observed as a gradual change in metal contents in time.

### 5.6. Climatic interpretations

Past environmental changes are analogue with the present seasonal meridional
displacement of the ITCZ (Intertropical Convergence Zone) and SPSH (South Pacific
Subtropical High) expansion/contraction, establishing cold-dry/warm-humid climate
conditions ('winter like/summer like') (Hebbeln et al., 2002; Lamy et al., 2001, 2002,
2010). In this regard, the Holocene climate of the semi-arid zone of Chile has alternated
between wet and dry phases, associated with the intensity and latitudinal position of the
Southern Westerly Wind belt (SWW). Studies based on pollen records in southern
coastal areas of Coquimbo (30°S) indicate that wet conditions were predominant before
cal BC 6750, which brought the expansion of swamp forests areas along the coast. This
wet period was followed by a long-lasting arid phase that culminated in peak dry
conditions at cal BC 5550–3750 (Maldonado and Villagrán, 2006; Maldonado and
Rozas, 2008). This timing matches the relative dry conditions detected in the first
portion of our pollen reconstruction, indicated by relative low values of the Pollen





Moisture Index (Fig. 10). Our results further suggest that these dry conditions were
followed by trends towards increasing precipitation from the mid-Holocene onwards
(from cal BC ~4500 to ~ cal AD 1800), as indicated by the increasing values in the
Pollen Moisture Index and continental runoff (Fig. 10). Al and Pb are considered
indicators of continental particle inputs to marine waters transported mainly by river
runoff and dust (Saito et al 1992; Calvert and Pedersen, 2007; Govin et al., 2012; Xu et
al. 2015; Ohnemus and Lam, 2015). In our cores, these elements showed a trend similar
to pollen, i.e., a gradual rise in time, suggesting increased humid conditions during
recent periods. Therefore, this pattern of humidity and continental influence variability
in Guanaqueros Bay sediments suggests an increase in winter rainfall or an increased
frequency of episodic rainfall events since ~250 Cal BC (Ortega et al., submitted).
South of 35°S, an overall strengthening of SWW and a poleward shift driven by orbital
forcings seem to be operating from the mid-Holocene (7 kyr BP) to pre-industrial
modern times (250 yr BP) (Varma et al., 2012), promoting more humid conditions in
this period. Our records indicates increases in grain size, K/Ca ratios, and Fe and Al
contents over time, all of which point to higher continental inputs, most probably by
increasing rainfall events, which should be an important source of Fe in northern
Chilean margin, related to primary productivity increases in sedimentary records
(Dezileau et al., 2004). This study shows slight increases in Fe concentrations at cal BC
~1500−2000 and at cal BC ~4000; both periods showing clear increases in primary
productivity. However, maximum productivity observed at cal BC ~4000 –interpreted
from the accumulation of nutrient type elements, and the distribution of opal and
diatoms– was consistent with overall dry environmental conditions (Fig. 6, 8), also
connected to an anoxic sulfidic environment. This could be related to an increased
intensity of the anticyclone and a consequently higher upwelling during this period
(Frugone-Álvarez et al., 2017).
Peak drying with higher productivity and lower oxygen conditions was followed by a
gradually decreasing primary productivity and organic fluxes. The maximum
productivity occurred during the driest period studied (mid-Holocene), suggesting
stronger upwelling that was weakening in time concomitantly with the increase of
humid conditions. In this case, weakened upwelling could be interpreted as an intrusion
of less nutrient-enriched upwelled waters over the shelf, influenced by remote equatorial
waves as observed today. Normally, the wet/dry phases associated with ENSO-like
conditions also influence bottom oxygenation which has been observed from the central



Peruvian coast to the southern Chilean regions (12 °S − 36 °S) (Escribano et al., 2004; Gutiérrez et al., 2008); thus, OMZ is expected to be less/more intense during warm/cold phases. Besides, the Walker circulation strength and the expansion/contraction of the SPSH also control marine productivity (Salvatteci et al., 2014), a key variable closely related to the supply of nutrient rich and poor oxygen waters belonging to the OMZ. Thus, warm periods tend to be associated with low productivity and weak OMZ (Salvatteci et al., 2014). In this sense, an increase in the frequency of ENSO-like warm periods influences bottom oxygenation, resulting in a less reduced environment concomitantly with less organic fluxes from primary productivity and less oxygen consumption during organic matter diagenesis.

## 6. Conclusions

The circulation seems to affect both places differently, leaving more variable grain compositions and higher TOC contents in the Guanaqueros Bay (core BGGC5) than in the Tongoy Bay (core BTGC8). This difference should be interpreted as an increase in the time of particle transportation resulting in grain size selection (more leptokurtic at core BTGC8), especially after cal AD 1. Furthermore, in both bays, constantly decreasing TOC contents were observed after cal BC ~2000 to the present, probably due to higher oxygenation of the bay bottom in time.

In the investigated sediments, differences in redox conditions could be reconstructed showing a clear decreasing trend in oxygen bottoms before the beginning of recent time (cal AD 1), followed by a rapid change to a more oxygenated environment in the last 2000 years. The environmental conditions at bottom waters were relevant in the metal enrichment factor above crustal abundance within the sediments (highest EFs), since low organic carbon accumulation and low sedimentation rates have been estimated, indicating that the accumulation of these elements (U, Mo and Re) depends mainly on oxygen content instead of on organic carbon burial rates. Apparently, a maximum suboxia-anoxia occurred at cal BC ~4500 year BC, when peak U and Re where recorded, probably due to the presence of a sulfidic environment.

The nutrient-type elements follow a similar trend, reduced at present and showing higher accumulation rates around cal BC 4500 (Ca, Ni, P and Cd). Their distribution is consistent with the diatom and opal distributions, showing their dependence on primary productivity and organic carbon burial rates. If the kinetics reaction is working at low



rates for these elements, they should be highly influenced during oxygenation periods,
something that seems to have been operating with higher frequencies.
The record of continental proxies establishes a continuous increment in wet
conditions, consistent with previous reconstructions in central Chile. The most
distinctive changes were observed after cal BC 4500, when an overall expansion of the
coastal vegetation occurred as a result of a progressive increase in precipitation and
river runoffs, expanding the grain size of the sediments and the increase
concentrations of elements that has relevant continental source (Al, Fe, K and Pb).
Increased regional precipitations amounts have been commonly interpreted by a
northward movement of the Southern Westerly Winds belts, but the increased
frequency of El Niño events have also introduced a high variability of humidity in the
late Holocene. Thus, the apparent increase of oxygen conditions at bottoms would be
the result of this oceanographic feature, which introduced a more oxygenated water
mass to the shelf and bays, temporarily changing the redox conditions in surface
sediments and affecting the sensitive elements to redox potential change in the
environment. Additionally, this also affected the accumulation of organic matter due
to an intensification of its remineralization, showing a decreasing trend in nutrient type
element accumulation and organic carbon burial rates towards the present.

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

**Acknowledgments**

We would like to thank the R/V Stella Maris II crew of Universidad Católica del Norte for their help and support during field work. We extend our acknowledgements to the laboratory assistants of the Paleoceanography Lab at Universidad de Concepción, for their aid in sample analyses. We also wish to thank Dr. Olivier Bruguier of CNRS and his lab personnel for their assistance during ICPMs analyses. We also express our gratitude to INNOVA 07CN13 IXM-150. This manuscript was funded by FONDECYT Project No. 1140851. Partial support from the COPAS Sur-Austral (CONICYT PIA PFB31) and FONDAP-IDEAL centers (No. 15150003) is also acknowledged.



**Tables**

Table 1. Concentration of elements in Pachingo wetland sediments, considered as lithogenic background for the study area. The values correspond to mean concentrations in surface sediments (0−3 cm).

| Element | Metal/Al x $10^3$ | s |
|:---:|:---:|:---:|
| Ca | 686.5 | 139.3 |
| Fe | 591.3 | 84.5 |
| P | 8.6 | 0.7 |
| Sr | 5.7 | 0.6 |
| Ba | 5.6 | 0.1 |
| Cu | 0.258 | 0.019 |
| Ni | 0.174 | 0.005 |
| U | 0.020 | 0.003 |
| Mo | 0.020 | 0.003 |
| Cd | 0.0021 | 0.0003 |
| Re | 0.00004 | 0.00001 |



Table 2. Radiocarbon dates for BGGC5 and BTGC8 sediment cores collected from mixed planktonic foraminifera and monospecific benthic foraminifera (*Bolivina plicata*), respectively. The [14]C-AMS was performed at NOSAM-WHOI. The lab code and conventional ages collected from each core section is indicated. For error calculations see http://www.whoi.edu/nosams/radiocarbon-data-calculations.

| Core identification | material | mass (mg) | Lab. Code NOSAM | modern fraction pMC | error | Conventional BP | Age error |
|---|---|---|---|---|---|---|---|
| **BGGC5** | | | | | | | |
| **10-11** | Planktonic foraminifera (mix) | 1.8 | OS-122160 | 0.8895 | 0.0027 | 940 | 25 |
| **18-19** | Planktonic foraminifera (mix) | 1.1 | OS-122141 | 0.7217 | 0.0024 | 2,620 | 25 |
| **31-32** | Planktonic foraminifera (mix) | 2.7 | OS-122161 | 0.6590 | 0.0021 | 3,350 | 25 |
| **45-46** | Planktonic foraminifera (mix) | 2 | OS-122162 | 0.6102 | 0.0017 | 3,970 | 25 |
| **55-56** | Planktonic foraminifera (mix) | 1.6 | OS-122138 | 0.5864 | 0.0025 | 4,290 | 35 |
| **66-67** | Planktonic foraminifera (mix) | 2.8 | OS-122304 | 0.5597 | 0.0018 | 4,660 | 25 |
| **76-77** | Planktonic foraminifera (mix) | 2.6 | OS-122163 | 0.4520 | 0.0016 | 6,380 | 30 |
| **96-97** | Planktonic foraminifera (mix) | 1.1 | OS-122139 | 0.4333 | 0.0033 | 6,720 | 60 |
| **115-116** | Planktonic foraminifera (mix) | 4.7 | OS-122164 | 0.3843 | 0.0016 | 7,680 | 35 |
| **125-126** | Planktonic foraminifera (mix) | 1.4 | OS-122140 | 0.3964 | 0.0025 | 7,430 | 50 |
| **BTGC8** | | | | | | | |
| **5-6** | Benthic foraminifera (*Bolivina plicata*) | 4.2 | OS-130657 | 0.8953 | 0.0017 | 890 | 15 |
| **20-21** | Benthic foraminifera (*Bolivina plicata*) | 7.7 | OS-123670 | 0.7337 | 0.0021 | 2,490 | 25 |
| **30-31** | Benthic foraminifera (*Bolivina plicata*) | 13 | OS-123671 | 0.6771 | 0.0016 | 3,130 | 20 |
| **40-41** | Benthic foraminifera (*Bolivina plicata*) | 11 | OS-123672 | 0.6507 | 0.0019 | 3,450 | 25 |
| **50-51** | Benthic foraminifera (*Bolivina plicata*) | 8.7 | OS-123673 | 0.5877 | 0.0014 | 4,270 | 20 |
| **60-61** | Benthic foraminifera (*Bolivina plicata*) | 13 | OS-123674 | 0.5560 | 0.0018 | 4,720 | 25 |
| **71-72** | Benthic foraminifera (*Bolivina plicata*) | 10 | OS-123675 | 0.4930 | 0.0013 | 5,680 | 20 |
| **80-81** | Benthic foraminifera (*Bolivina plicata*) | 7.3 | OS-123676 | 0.4542 | 0.0012 | 6,340 | 20 |
| **90-91** | Benthic foraminifera (*Bolivina plicata*) | 6.8 | OS-123677 | 0.4259 | 0.0015 | 6,860 | 30 |
| **96-97** | Benthic foraminifera (*Bolivina plicata*) | 6.8 | OS-123678 | 0.3903 | 0.0013 | 7,560 | 25 |





Table 3. Reservoir age (DR) estimation considering the [210]Pb age determined with the CRS model (McCaffrey and Thomson, 1980) at a selected depth sections of the core, compared with [14]C ages (yr BP) from marine13.14 curve (Reimer et al., 2013), according to Sabatier et al. (2010).

| Core | cm | Age from CRS model | Age years BP[a] | [14]C marine13 curve | [14]C age BP from foram. | DR | s |
|---|---|---|---|---|---|---|---|
| BGGC5 | 10.5 | 1828 | 122 | 499 | 940 | 441 | 15 |
| BTCG8 | 5.5 | 1908 | 42 | 448 | 890 | 442 | 17 |

a. Before present=1950





Table 4. Spearman rank order correlations for geochemical data. Significant values >0.8 are indicated in bold.

**BGGC5**

| | Al | P | K | Ca | Mn | Fe | Ni | Cu | Mo | Cd | Re | Sr | U | Ba | Opal | TOC |
|---|---|---|---|---|---|---|---|---|---|---|---|---|---|---|---|---|
| **Al** | 1.00 | -0.62 | 0.49 | -0.48 | 0.64 | 0.60 | -0.75 | 0.56 | -0.10 | -0.73 | -0.08 | -0.33 | 0.08 | 0.49 | -0.52 | -0.44 |
| **P** | | 1.00 | -0.31 | 0.37 | -0.45 | -0.56 | 0.56 | -0.57 | 0.01 | 0.61 | -0.11 | 0.39 | -0.12 | -0.20 | 0.49 | 0.24 |
| **K** | | | 1.00 | -0.24 | **0.90** | **0.83** | -0.29 | 0.47 | 0.28 | -0.42 | 0.33 | -0.12 | 0.50 | 0.26 | -0.25 | -0.19 |
| **Ca** | | | | 1.00 | -0.47 | -0.50 | 0.44 | -0.64 | 0.23 | 0.59 | 0.39 | **0.92** | 0.30 | -0.60 | 0.18 | 0.32 |
| **Mn** | | | | | 1.00 | **0.94** | -0.51 | 0.68 | -0.01 | -0.68 | 0.07 | -0.32 | 0.24 | 0.43 | -0.39 | -0.31 |
| **Fe** | | | | | | 1.00 | -0.49 | **0.81** | 0.03 | -0.70 | 0.11 | -0.40 | 0.23 | 0.36 | -0.37 | -0.21 |
| **Ni** | | | | | | | 1.00 | -0.51 | 0.49 | **0.91** | 0.35 | 0.25 | 0.26 | -0.70 | 0.72 | 0.64 |
| **Cu** | | | | | | | | 1.00 | -0.12 | -0.71 | -0.06 | -0.61 | 0.00 | 0.31 | -0.39 | -0.07 |
| **Mo** | | | | | | | | | 1.00 | 0.50 | **0.88** | 0.10 | 0.91 | -0.48 | 0.33 | 0.36 |
| **Cd** | | | | | | | | | | 1.00 | 0.36 | 0.42 | 0.27 | -0.67 | 0.70 | 0.54 |
| **Re** | | | | | | | | | | | 1.00 | 0.27 | **0.92** | -0.50 | 0.16 | 0.38 |
| **Sr** | | | | | | | | | | | | 1.00 | 0.24 | -0.36 | 0.05 | 0.17 |
| **U** | | | | | | | | | | | | | 1.00 | -0.39 | 0.10 | 0.29 |
| **Ba** | | | | | | | | | | | | | | 1.00 | -0.30 | -0.59 |
| **Opal** | | | | | | | | | | | | | | | 1.00 | 0.35 |
| **TOC** | | | | | | | | | | | | | | | | 1.00 |

**BTGC8**

| | Al | P | K | Ca | Mn | Fe | Ni | Cu | Mo | Cd | Re | Sr | U | Ba | Opal | TOC |
|---|---|---|---|---|---|---|---|---|---|---|---|---|---|---|---|---|
| **Al** | 1.00 | -0.19 | -0.17 | -0.37 | -0.02 | -0.03 | -0.39 | -0.04 | -0.39 | 0.02 | -0.13 | -0.58 | -0.19 | 0.07 | -0.41 | -0.29 |
| **P** | | 1.00 | 0.23 | 0.00 | 0.43 | 0.28 | 0.58 | 0.23 | 0.37 | 0.13 | -0.04 | 0.30 | 0.14 | -0.14 | 0.56 | 0.13 |
| **K** | | | 1.00 | -0.02 | 0.54 | 0.41 | 0.43 | 0.22 | -0.11 | 0.05 | -0.04 | 0.19 | -0.28 | 0.28 | 0.26 | 0.20 |
| **Ca** | | | | 1.00 | -0.33 | -0.27 | 0.00 | -0.23 | 0.39 | 0.01 | 0.33 | 0.50 | 0.47 | -0.34 | 0.20 | 0.34 |
| **Mn** | | | | | 1.00 | 0.21 | 0.64 | 0.01 | 0.05 | 0.33 | 0.15 | 0.32 | -0.02 | 0.24 | 0.32 | 0.00 |
| **Fe** | | | | | | 1.00 | 0.13 | 0.71 | -0.40 | -0.48 | -0.67 | -0.37 | -0.62 | 0.13 | 0.14 | 0.10 |
| **Ni** | | | | | | | 1.00 | 0.24 | 0.56 | 0.20 | 0.25 | 0.64 | 0.19 | -0.16 | **0.80** | 0.45 |
| **Cu** | | | | | | | | 1.00 | -0.25 | -0.68 | -0.56 | -0.22 | -0.61 | -0.10 | 0.21 | 0.37 |
| **Mo** | | | | | | | | | 1.00 | 0.45 | 0.59 | 0.66 | 0.69 | -0.41 | 0.58 | 0.30 |
| **Cd** | | | | | | | | | | 1.00 | 0.56 | 0.39 | 0.52 | 0.11 | 0.10 | -0.12 |
| **Re** | | | | | | | | | | | 1.00 | 0.53 | **0.83** | -0.16 | 0.13 | 0.17 |
| **Sr** | | | | | | | | | | | | 1.00 | 0.58 | -0.13 | 0.52 | 0.23 |
| **U** | | | | | | | | | | | | | 1.00 | -0.19 | 0.21 | 0.00 |
| **Ba** | | | | | | | | | | | | | | 1.00 | -0.28 | -0.42 |
| **Opal** | | | | | | | | | | | | | | | 1.00 | 0.39 |
| **TOC** | | | | | | | | | | | | | | | | 1.00 |





**Figures**

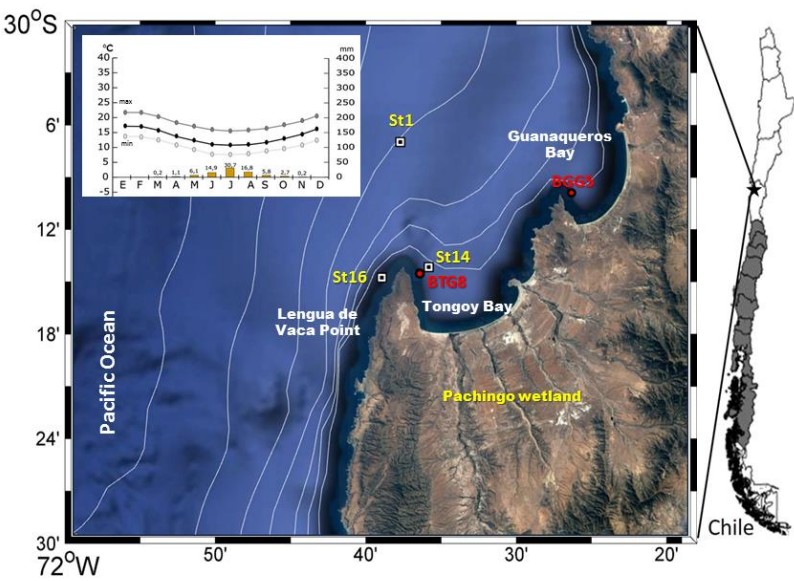

Figure 1. Study area showing the position of sampling stations. Sediment cores were retrieved from Guanaqueros Bay (BGGC5) and from Tongoy Bay (BTGC8) at water depths of 89 and 85 m, respectively. Information of dissolved oxygen (DO) in the water column and of suspended organic particles collected at ST1, ST14 and ST16 sampling sites was gathered in a previous project (INNOVA 07CN13 IXM-150). Climograph of the region is showing the average precipitation in mm (bars) and temperatures in °C (min, max and average) over 12-month period.





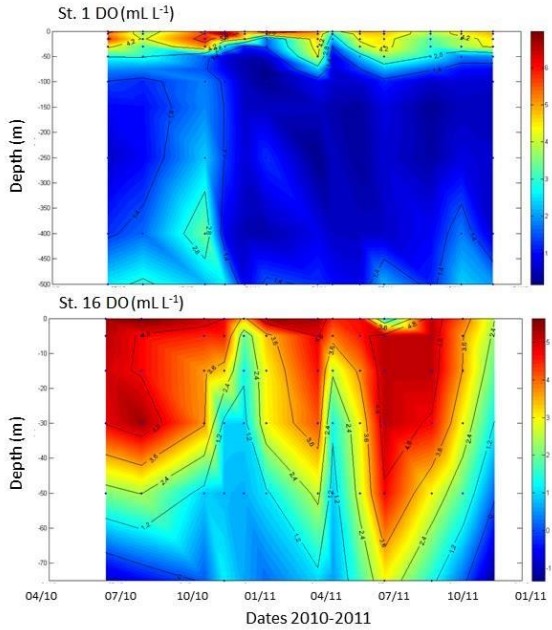

Figure 2. Dissolved Oxygen (DO) time series in the water column measured between
October 2010 and January 2011, at stations St1 and St16 off Tongoy Bay, Coquimbo
(30°S).



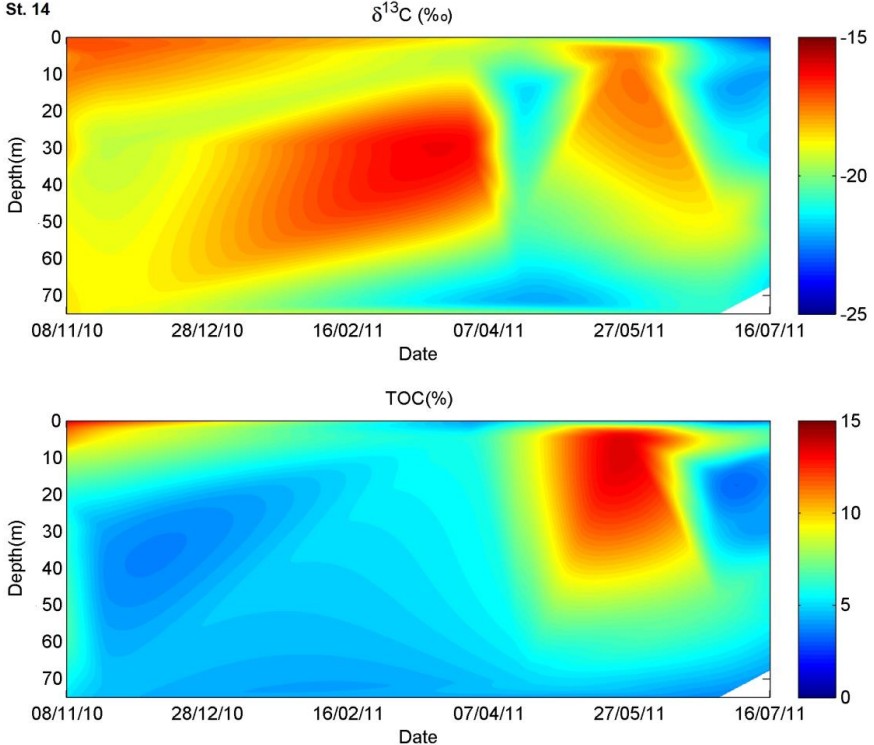

Figure 3. Suspended particulate matter composition (TOC % and $\delta^{13}$Corg) measured in the water column between October 2010 and October 2011, at station St14, Tongoy Bay, Coquimbo (30°S).



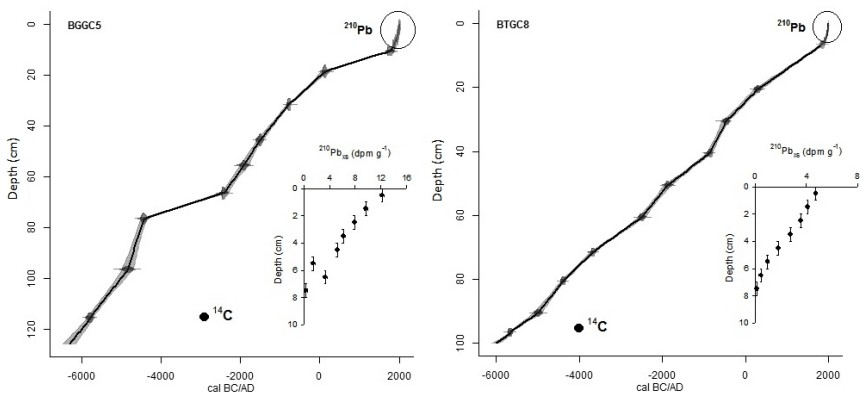

Figure 4. Age model based on $^{14}$CAMS and $^{210}$Pb measurements. The time scale was obtained according to the best fit of curves of $^{210}$Pb$_{xs}$ and $^{14}$C points using CLAM 2.2 software and Marine curve $^{13}$C (Reimer et al., 2013).

10.5194/bg-2018-396
Biogeosciences
2018-09-11

a)

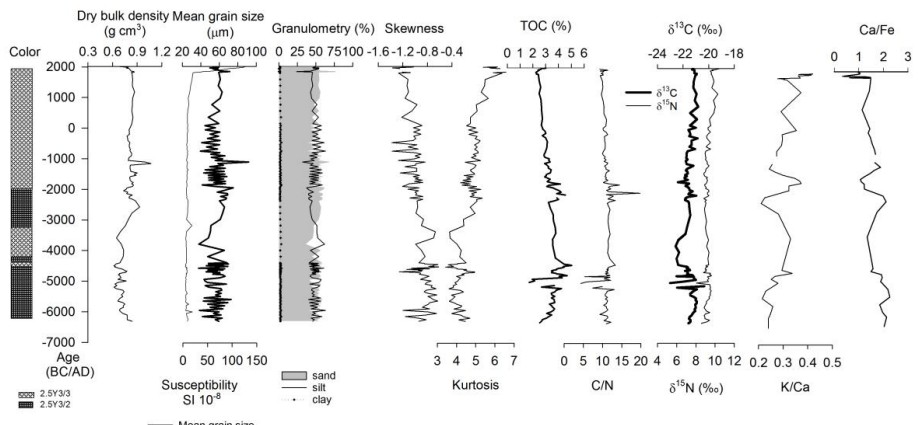

b)

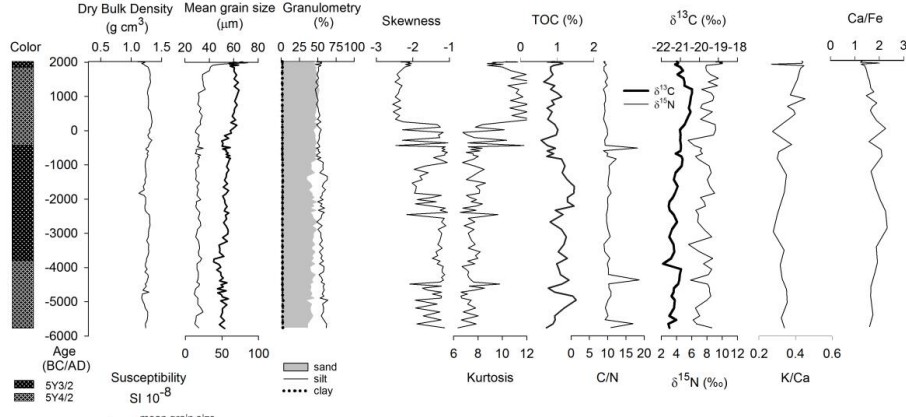

Figure 5. Sediment characterization of sediment cores retrieved from (a) Guanaqueros Bay (BGGC5) and (b) Tongoy Bay (BTGC8). Distribution in depth core of color, dry bulk density, statistical parameters (skewness, mean grain size, kurtosis), organic components (TOC, stable isotopes) and chemical composition (K/Ca, Ca/Fe).





a)



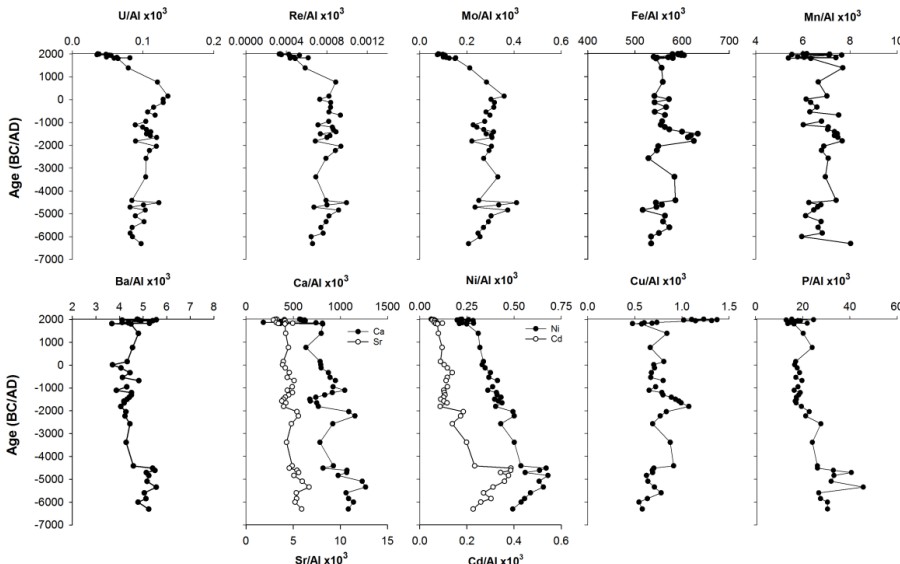

b)

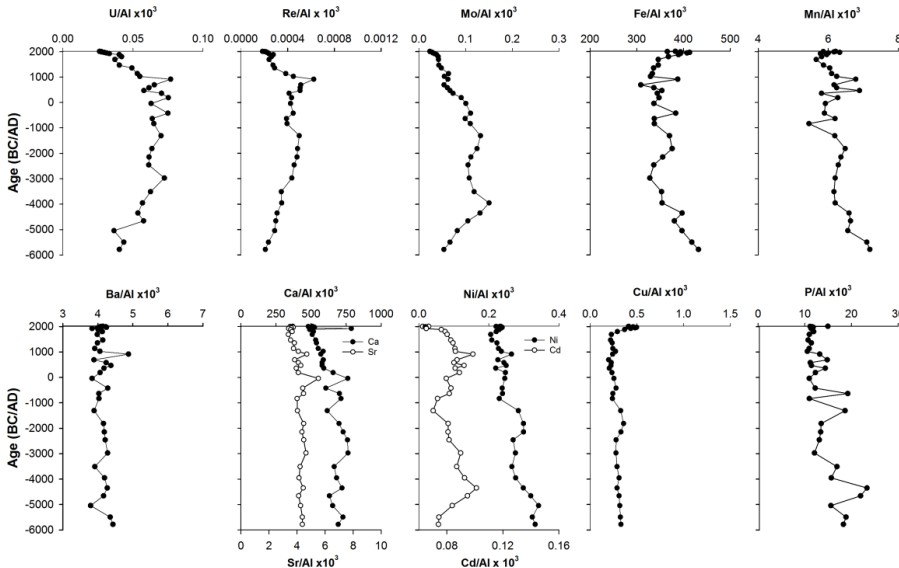

Figure 6. Trace element distribution in sediment cores retrieved from (a) Guanaqueros
Bay (BGGC5) and (b) Tongoy Bay (BTGC8), off Coquimbo (30°S).

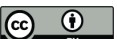



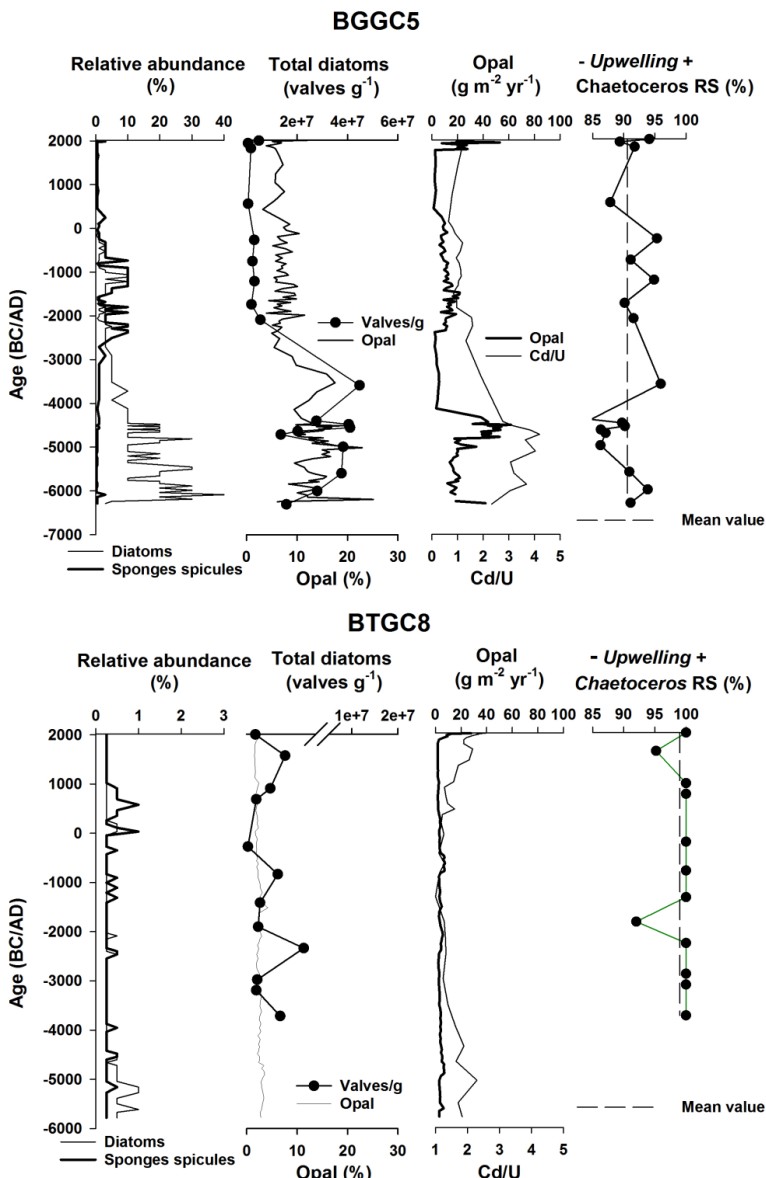

Figure 7. Diatom abundance, opal accumulation and temporal variations in the relative abundance of *Chaetoceros* resting spores in BGGC5 and BTGC8 cores (Guanaqueros and Tongoy Bay, respectively). Cd/U distribution was included as a proxy for redox condition.



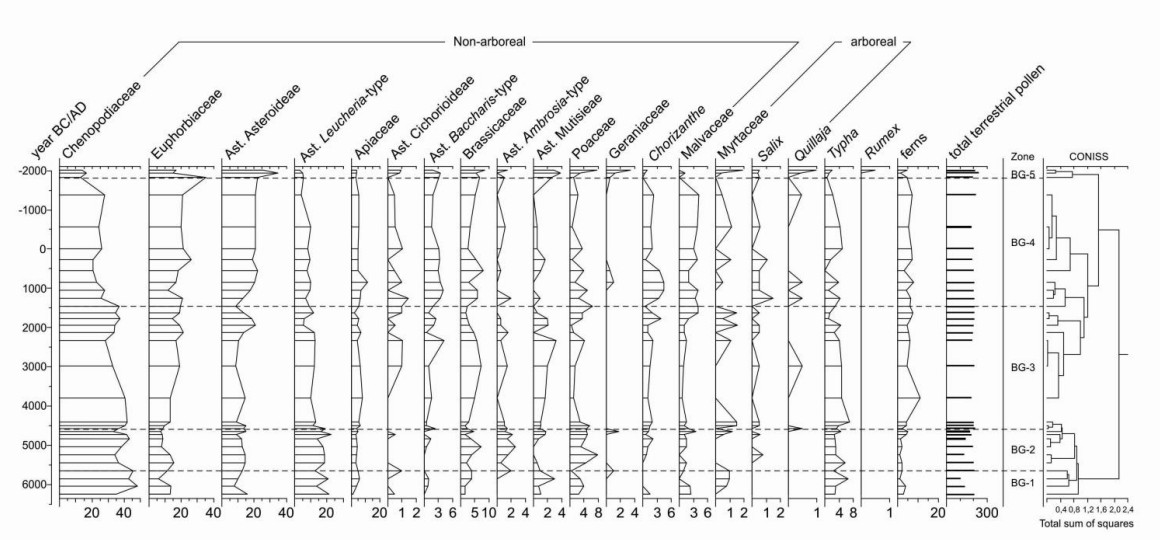

Figure 8. Pollen record in BGGC5 core.





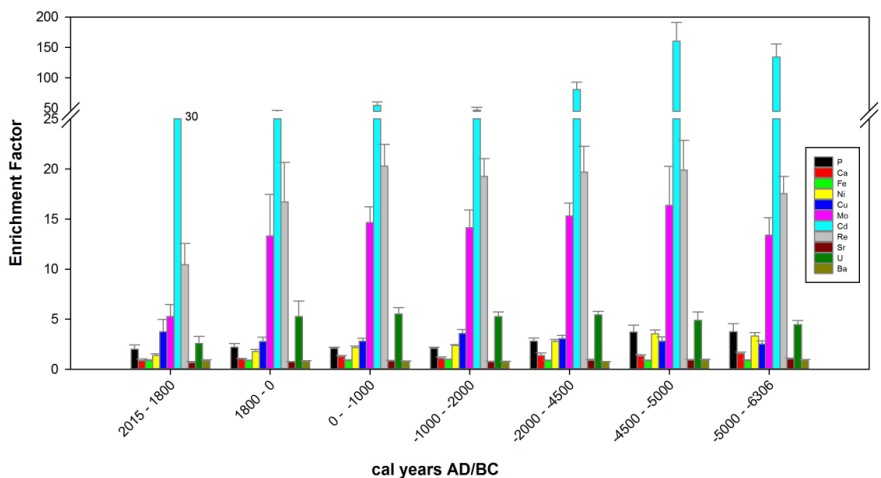

Figure 9. Authigenic enrichment factor (EF) of trace elements in BGGC5 core. Lithogenic background as estimated from surface sediments of Pachingo wetland cores (see text).





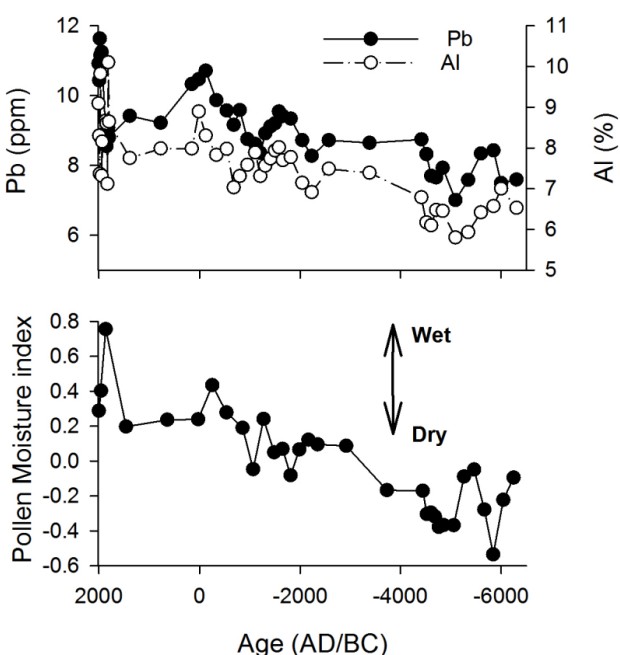

Figure 10. Pollen Moisture Index defined as the normalized ratio between Euphorbiaceae (wet coastal shrub land) and Chenopodiaceae (arid scrubland). Positive (negative) values for this index indicate the relative expansion (reduction) of coastal vegetation under wetter (drier) conditions. Pb and Al distribution at BGGC5 core, representatives of terrigenous input to the bay.