# Peer review of "Reconstructing past variations in environmental conditions and paleoproductivity"

_Biogeosciences, 2018_

## Referee Comment (RC1) · Anonymous Referee #1 · 12 Oct 2018

The paper presents a multiproxy analysis of two short coastal sediments cores collected off Coquimbo, Chile, with the aim to document paleoclimate and paleoceanographic variability during the Holocene. The data presented is original and valuable to understand the millennial dynamics of the South East Pacific coastal upwelling. Authors analyzed a broad range of geochemical and microfossil indicators which should lead to a robust interpretation. However, substantial work is still needed on the manuscript before being published. I have a few methodological concerns with the chronology and with the way metal concentrations are used, that need to be addressed. The text also requires a lot of work. Except for method sections, the text in general lacks clarity, partly because of inappropriate word choices, and partly because of a lack of

focus. The introduction needs to be rewritten since it does not present the context, the research motivation, or the objectives of the work. A proper paleoclimate discussion is missing. Almost no comparison with published results was made and none of the relevant literature on the regional paleoceanography or paleo ENSO is cited.

This study deserves to be published but the manuscript requires substantial revision. So far the article is essentially focused on sediment chemistry but lacks depth in the paleoceanographic interpretation and discussion which is the objective. I recommend a more active contribution of co-authors in writing the introduction, discussion and conclusions.

Detailed comments:

- The presentation of results in the abstract is unclear

- The introduction is a lengthy, disorganized list of unfocused information about upwellings in general and sediment proxies. It needs to be entirely rewritten to present the context, the motivation of the research, the scientific questions, the objectives and the scientific strategies chosen to achieve them.

- L132-139: this paragraph on pigments seems unnecessary

- L145: the words "relevance" and "relevant" are repeatedly used in an inappropriate way throughout the manuscript.

- L167-172: unprecise

- L176-178: the fact that two sediments cores were analyzed and their location should be mentioned in the introduction

- Trace metal concentrations:

The normalization of Me concentrations using Al does not seem justified to me. The analytic technique used here (ICPMS analyses of dissolved samples) yields quantitative and absolute concentration values thanks to the standards used. As far as I know,

uncertainties related to machine variability and matrix effects are not an issue with this technique as it would be with laser ablation technique. In addition, Al does not have a conservative behavior as mentioned: figure 10 shows on the contrary a substantial increase of Al concentration through the Holocene. Normalizing systematically with this element may actually produce biased interpretations.

I recommend to use the accumulation rate from the age model and absolute Me concentration to calculate metal fluxes to the sediment.

Since Al has mainly a continental origin, ratios with Al is informative for elements whose flux is related to productivity to discuss relative contribution of marine vs terrestrial contributions in the sediment.

Finally, the usefulness of the enrichment factors is not obvious. Figure 9 is barely discussed. In addition, I wonder if wetland sediments are really representative of crustal metal concentrations since they also contain organic matter.

- Geochronology

L248: Calpal2007_HULU calibration curve is an odd choice for radiocarbon calibration. It is also inconsistent with L255 in which Marine13 is mentioned (which is the correct calibration curve to use). There is a couple of issues with the regional radiocarbon reservoir age used for calibration. First, the method to calculate it is not correct. 14C reservoir age should be calculated in the 14C age scale, not in the calendar scale as it was done here. dR is the difference between the marine sample 14C age and the 14C age that corresponds to the absolute age (here obtained from the 210Pb model) using the Marine13 curve. See Southon et al. (1995) for details on the technique. The dR value obtained here is larger than any dR values obtained previously on the Chilean coast Authors should read and use Ortlieb et al., 2011; Carré et al., 2016; and Merino-Campos et al., 2018. The latter reference presents 37 prebomb dR values all along the Chilean coast measured with a reliable technique. Using a value from this publication would be more reliable. The first 2 references show changes in dR values through the

Holocene that should also be discussed. Finally, instead of BC/AD, ages should all be presented in the BP scale as it is usual in paleoceanography for Holocene studies.

- Discussion:

L505-L514: unclear

L521-L536: the discussion about d13C values is unclear, in part because there seem to be a confusion between Total organic carbon(TOC) in the water column and suspended particulate organic Matter (SPM). Is it possible that the difference between d13C values in the water and in the sediment are due to the difference between TOC and SPM? A preferential degradation of 13C enriched particles is mentioned (L528-529): could you support this with a reference?

L563-L568: the discussion about K is not very convincing. A reference about the detritic origin of K is needed. Ca could also have a detritic origin so close to the shoreline. Al, Fe are also clear terrestrial input indicators. Why not discuss them together?

L602-L606: references needed

Section 5.3 should be shortened. It is somewhat redundant with other discussion sections and the result section.

- Climatic interpretations

This section lacks in-depth discussion. The results here should be compared to published results to understand how they contribute, support or contradict existing hypothesis about millenial oceanographic variability in Chile.

L720-L723: "past changes are analogue with the present meridional displacement of the ITCZ and the SPCH". This should not be taken as a fact. It is only a hypothesis used as an interpretation model.

L744-L747: this part is unclear and sounds contradictory (a poleward shift of SWW

should not promote humid conditions in central Chile). In addition, this is a model result. Why not compare with existing paleoenvironmental and paleoceanographic data?

There is a series of sediment cores that document past oceanographic conditions in the Peru-Chile upwelling system during the Holocene. This includes Lamy et al (1999, 2001, 2002, 2010), Kim et al. (2002), Hebbeln et al. (2002), Rein et al. (2005), Salvatecci et al. (2014, 2016). On a regional scale, the data presented here confirm a La Niña-like situation in the early to mid-Holocene, which is in agreement with previous datasets including Koutavas et al. (2002), Fontugne et al. (2004), Conroy et al. (2008); Carré et al. (2012), and model experiments such as Brown et al. (2008); Braconnot et al. (2012), Luan et al. (2015). This list is clearly not exhaustive.

The influence of ENSO variability needs obviously to be discussed. It is here briefly mentioned in the text, appears in the key words, but there is no discussion. Data on past ENSO activity do exist (Koutavas et al., 2006; Cobb et al., 2013; Carré et al., 2014) and they need to be included in the discussion if the role of ENSO in the presented data is to be evaluated.

Figure 2: what about st14? Font on Y scale too small Figure 3: SPM is not the same as TOC Figure 5: it is not clear which curve is grain size and which is susceptibility Figure 6: Al and Fe are both related to terrestrial input. What information ndoes Fe/Al provide? Figure 9: This figure is not commented in the text. EF calculation does not seem useful.

References: Braconnot, P., Y. Luan, S. Brewer, and W. Zheng (2012), Impact of Earth's orbit and freshwater fluxes on Holocene climate mean seasonal cycle and ENSO characteristics, Climate Dynamics, 38, 1081-1092.

Brown, J., M. Collins, A. Tudhope, and T. Toniazzo (2008), Modelling mid-Holocene tropical climate and ENSO variability: towards constraining predictions of future change with palaeo-data, Climate Dynamics, 30, 19-36.

Carré, M., M. Azzoug, I. Bentaleb, B. M. Chase, M. Fontugne, D. Jackson, M.-P. Ledru, A. Maldonado, J. P. Sachs, and A. J. Schauer (2012), Mid-Holocene mean climate in the south-eastern Pacific and its influence on South America, Quaternary International, 253, 55-66, doi:10.1016/j.quaint.2011.02.004.

Carré, M., D. Jackson, A. Maldonado, B. M. Chase, and J. P. Sachs (2016), Variability of 14C reservoir age and air-sea flux of CO2 in the Peru-Chile upwelling region during the past 12,000 years, Quaternary Research, 85, 87-93, doi:10.1016/j.yqres.2015.12.002.

Carré, M., J. P. Sachs, S. Purca, A. J. Schauer, P. Braconnot, R. Angeles Falcón, M. Julien, and D. Lavallée (2014), Holocene history of ENSO variance and asymmetry in the eastern tropical Pacific, Science, 345, 1045-1048, 10.1126/science.1252220.

Cobb, K. M., N. Westphal, H. R. Sayani, J. T. Watson, E. Di Lorenzo, H. Cheng, R. L. Edwards, and C. D. Charles (2013), Highly Variable El Niño-Southern Oscillation Throughout the Holocene, Science, 339, 67-70, 10.1126/science.1228246.

Conroy, J. L., J. T. Overpeck, J. E. Cole, T. M. Shanahan, and M. Steinitz-Kannan (2008), Holocene changes in eastern Pacific climate inferred from a Galápagos lake sediment record, Quaternary Science Reviews, 27, 1166-1180.

Fontugne, M., M. Carré, I. Bentaleb, M. Julien, and D. Lavallée (2004), Radiocarbon reservoir age variations in the south Peruvian upwelling during the Holocene., Radiocarbon, 46, 531-537.

Hebbeln, D., M. Marchant, and G. Wefer (2002), Paleoproductivity in the southern Peru-Chile current through the last 33 000 yr, Marine Geology, 186, 487-504.

Kim, J.-H., R. R. Schneider, D. Hebbeln, P. J. Müller, and G. Wefer (2002), Last deglacial sea-surface temperature evolution in the Southeast Pacific the South American continent, Quaternary Science Reviews, 21, 2085-2097.

Koutavas, A., P. B. deMenocal, G. C. Olive, and J. Lynch-Stieglitz (2006), Mid-Holocene El Niño-Southern Oscillation (ENSO) attenuation revealed by individual foraminifera in

eastern tropical Pacific sediments, Geology, 34, 993-996.

Koutavas, A., J. Lynch-Stieglitz, T. M. Marchitto Jr., and J. P. Sachs (2002), El Niño-like pattern in ice age tropical sea surface temperature, Science, 297, 226-230.

Lamy, F., D. Hebbeln, U. Röhl, and G. Wefer (2001), Holocene rainfall variability in southern Chile: a marine record of latitudinal shifts of the southern westerlies, Earth and Planetary Science Letters, 185, 369-382.

Lamy, F., D. Hebbeln, and G. Wefer (1999), High-resolution marine record of climatic change in mid-latitude Chile during the last 28,000 years based on terrigenous sediment parameters, Quaternary Research, 51, 83-93.

Lamy, F., R. Kilian, H. W. Arz, J.-P. Francois, J. Kaiser, M. Prange, and T. Steinke (2010), Holocene changes in the position and intensity of the southern westerly wind belt, Nature Geosci, 3, 695-699.

Lamy, F., C. Rühlemann, D. Hebbeln, and G. Wefer (2002), High- and low-latitude climate control on the position of the southern Peru-Chile current during the Holocene, Paleoceanography, 17, 16, 11-10.

Luan, Y., P. Braconnot, Y. Yu, and W. Zheng (2015), Tropical Pacific mean state and ENSO changes: sensitivity to freshwater flux and remnant ice sheets at 9.5 ka BP, Climate Dynamics, 1-18, doi:10.1007/s00382-015-2467-7.

Merino-Campos, V., R. De Pol-Holz, J. Southon, C. Latorre, and S. Collado-Fabbri (2018), Marine radiocarbon reservoir age along the chilean continental margin, Radiocarbon, 1-16.

Ortlieb, L., G. Vargas, and J.-F. Saliège (2011), Marine radiocarbon reservoir effect along the northern Chile-southern Peru coast (14-24°S) throughout the Holocene, Quaternary Research, 75, 91-103.

Rein, B., A. Lückge, L. Reinhardt, F. Sirocko, A. Wolf, and W.-C. Dullo (2005),

El Niño variability off Peru during the last 20,000 years, Paleoceanography, 20, 10.1029/2004PA001099.

Salvatteci, R., D. Gutiérrez, D. Field, A. Sifeddine, L. Ortlieb, I. Bouloubassi, M. Boussafir, H. Boucher, and F. Cetin (2014), The response of the Peruvian Upwelling Ecosystem to centennial-scale global change during the last two millennia, Clim. Past, 10, 715-731.

Salvatteci, R., D. Gutierrez, A. Sifeddine, L. Ortlieb, E. Druffel, M. Boussafir, and R. Schneider (2016), Centennial to millennial-scale changes in oxygenation and productivity in the Eastern Tropical South Pacific during the last 25,000 years, Quaternary Science Reviews, 131, Part A, 102-117, doi:10.1016/j.quascirev.2015.10.044.

Southon, J. R., A. Oakland Rodman, and D. True (1995), A comparison of marine and terrestrial radiocarbon ages from northern Chile, Radiocarbon, 37, 389-393.

---

## Referee Comment (RC2) · Anonymous Referee #2 · 22 Oct 2018

The authors present a multi-proxy approach based on two sediment cores examining terrestrial or biogenic inputs, sediment oxygen levels and primary productivity to study variations in past oceanographic and climatic changes along the Chilean coast. They suggest that their observations of wetter condition, lower productivity and the higher oxygen levels during the last 2000 years are coupled to the higher El Nino frequency.

The authors present a large range of biogeochemical and microfossil proxies and the results are worth to be published. However, I agree with referee #1 that the discussion of the results is not sufficient and needs substantial alteration, more in-depth interpretation of the own data as well as comparison to relevant literature.

Overall I have several major issues that need to be addressed

1. The Discussion is too short especially in comparison to the methods section. There is actually room for more detailed interpretations, for example the Nitrogen isotopes are not explained or discussed at all. The connections between sentences and paragraphs are often weak or confusing, there is some refinement needed and the reader must be led more through the text, especially the discussion. As the manuscript reads now it appears as you randomly choose some results to discuss one after the other. For example see paragraphs starting in lines 521 and 537. To further strengthen the discussion add more comparisons to local studies such as (Contreras et al., 2007; Díaz-Ochoa et al., 2010; Fukuda et al., 2013; Mohtadi et al., 2008; Ortega et al., 2012).

2. Through the whole manuscript the authors refer to suboxic/anoxic conditions, however, the values given for the water station 16 seems to be well above the suboxic value of

Munsell chart colour really needed?

For example get rid of Line 181 to 183.

Specific remarks:

Figure 1: Please add the surface circulation for the area.

Figure 2: unsharp and colors are hard to distinguish, this needs revision, I suggest to use a color range that is more appropriate to highlight the DO values of the low end of the scale more. Numbers in this plot need to be larger as well.

Figure 5: I think the accumulation rates for TOC should be given here instead of just (%), further please add the core number directly behind a.) and b.) in the figure.

Figure 10: I suggest to also put the Age on the Y-axis here as in all the other figures.

Line 35 – add "The" before Coquimbo

Line 78 to 83: rephrase, you cannot refer to "these boundary current ecosystems" in one sentence and then explain it afterwards.

Line 131-132: maximum Chl a concentrations of ...

Line 209-216: remove this paragraph, the section is already long and you only list the following chapters here.

Line 218: change the comma to a dot.

Line 384- 390. This was a bit confusing as a southern and northern area are introduced, but both cores studied are in the southern area?

Line 720: "Past environmental changes are analogue..." please specify these changes clearly here.

Line 724: "in this regard", it's not clear what you are referring to

Line 726: Studies based on pollen records ... There is a citation missing here!
Line 747-51: rephrase, improve the connection to the sentence before by first saying that you see indications of higher continental inputs due to increased rainfall, than which of your data shows this and which other studies support this observation. I further suggest to split this sentence in two.

Line 759: rephrase "peak drying"

References

Contreras, S., Pantoja, S., Neira, C., Lange, C.B., 2007. Biogeochemistry of surface sediments off Concepción (âĹij36°S), Chile: El Niño vs. non-El Niño conditions. Progress in Oceanography 75, 576–585. doi:10.1016/j.pocean.2007.08.030

De Pol-Holz, R., Ulloa, O., Dezileau, L., Kaiser, J., Lamy, F., Hebbeln, D., 2006. Melting of the Patagonian Ice Sheet and deglacial perturbations of the nitrogen cycle in the eastern South Pacific. Geophys. Res. Lett. 33, L04704. doi:10.1029/2005GL024477

De Pol-Holz, R., Ulloa, O., Lamy, F., Dezileau, L., Sabatier, P., Hebbeln, D., 2007. Late Quaternary variability of sedimentary nitrogen isotopes in the eastern South Pacific Ocean. Paleoceanography 22, PA2207. doi:10.1029/2006PA001308

Díaz-Ochoa, J.A., Pantoja, S., De Lange, G.J., Lange, C.B., Sánchez, G.E., Acuña, V.R., Muñoz, P., Vargas, G., 2010. Oxygenation variability in Mejillones Bay, off northern Chile, during the last two centuries. Biogeosciences Discuss. 7, 4987–5009. doi:10.5194/bgd-7-4987-2010

Fukuda, M., Harada, N., Sato, M., Lange, C.B., 2013. 230 Th-normalized fluxes of biogenic components from the central and southernmost Chilean margin over the past 22,000 years. Geochemical ... 1–17.

Mohtadi, M., Rossel, P., Lange, C.B., Pantoja, S., Böning, P., Repeta, D.J., Grunwald, M., Lamy, F., Hebbeln, D., Brumsack, H.-J., 2008. Deglacial pattern of circulation and marine productivity in the upwelling region off central-south Chile. Earth and Planetary Science Letters 272, 221–230. doi:10.1016/j.epsl.2008.04.043

BGD
Ortega, C., Vargas, G., Rutllant, J.A., Jackson, D., Méndez, C., 2012. Major hydrological regime change along the semiarid western coast of South America during the early Holocene. Quaternary Research 78, 513–527. doi:10.1016/j.yqres.2012.08.002

Verleye, T.J., Martinez, P., Robinson, R.S., Louwye, S., 2013. Changes in the source of nutrients associated with oceanographic dynamics offshore southern Chile ( $41^{\circ}$ S) over the last 25,000years. Quaternary Research 80, 495–501. doi:10.1016/j.yqres.2013.07.002

---

## Author Response (AR1)

Answers to referee 1

*General comments*

*The paper presents a multiproxy analysis of two short coastal sediments cores*
*collected off Coquimbo, Chile, with the aim to document paleoclimate and*
*paleoceanographicvariability during the Holocene. The data presented is original and*
*valuable to understand the millennial dynamics of the South East Pacific coastal*
*upwelling. Authors analyzed a broad range of geochemical and microfossil indicators*
*which should lead to a robust interpretation. However, substantial work is still needed*
*on the manuscript before being published. I have a few methodological concerns with*
*the chronology and with the way metal concentrations are used, that need to be*
*addressed. The text also requires a lot of work. Except for method sections, the text in*
*general lacks clarity, partly because of inappropriate word choices, and partly*
*because of a lack of focus. The introduction needs to be rewritten since it does not*
*present the context, the research motivation, or the objectives of the work. A proper*
*paleoclimate discussion is missing. Almost no comparison with published results was*
*made and none of the relevant literature on the regional paleoceanography or paleo*
*ENSO is cited. This study deserves to be published but the manuscript requires*
*substantial revision. So far the article is essentially focused on sediment chemistry but*
*lacks depth in the paleoceanographic interpretation and discussion which is the*
*objective. I recommend a more active contribution of co-authors in writing the*
*introduction, discussion and conclusions-*

Answer:

The introduction was re-written considering other aspects related to climatic past
variability. The last paragraph highlights the objective of our study.

The paleoclimate section (5.4) was rewritten, considering paleoclimatic conditions
observed by other authors in the Chilean northern margin. We avoided comparisons
with studies conducted over a wide range of time periods which extend beyond the
Holocene. Our cores only show records from the mid-Holocene onwards and therefore,
we focus our discussion on that time range. We made comparisons with studies
conducted near the zone of influence in the northern part of the Southwest Winds so as
to prevent the discussion from being unnecessarily long.  We included studies that work
on our time scale, including: Lamy et al., 1999, 2001, 2010; and Hebbeln et al., 2002.
Some of these studies mention the effects of the ENSO in the area, which is the main
driver of environmental changes therein. Our objective was not to establish periods of
occurrence of these events but to establish changes in productivity and redox conditions,
which are obviously subject to climatic and oceanographic forces, such as El Niño. In
addition, we included some works by Gutiérrez and Salvatecci conducted in southern
Peru, which considers the response of upwelling ecosystems to climatic changes during
the last Holocene.

Answers to detailed comments:

*Detailed comments:*
*- The presentation of results in the abstract is unclear:*

The text was completely modified from lines 33 to 49.

*- The introduction is a lengthy, disorganized list of unfocused information about*
*upwellings in general and sediment proxies. It needs to be entirely rewritten to*
*present the context, the motivation of the research, the scientific questions, the*
*objectives and the scientific strategies chosen to achieve them.*

We modified the introduction completely, from lines 56 to 102.

**- L132-139: this paragraph on pigments seems unnecessary**
We rewrote the section about the study area and we are omitting superfluous
information, we deleted the paragraph between lines 132 and 139.

**- L145: the words "relevance" and "relevant" are repeatedly used in an**
**inappropriate way throughout the manuscript.**

We modified all lines and replaced the word for other more appropriate terms, except in
lines 555 and 706.

*- L167-172: unprecise*
*- L176-178: the fact that two sediments cores were analyzed and their location should*
*be mentioned in the introduction*

We added extra information at the end of the introduction; see lines 118 to 122. We
provided further explanation about our point at the end of the study area section, lines
161-167.

*- Trace metal concentrations:*
*The normalization of Me concentrations using Al does not seem justified to me. The*
*analytic technique used here (ICPMS analyses of dissolved samples) yields*
*quantitative and absolute concentration values thanks to the standards used. As far as*
*I know, uncertainties related to machine variability and matrix effects are not an*
*issue with this technique as it would be with laser ablation technique. In addition, Al*
*does not have a conservative behavior as mentioned: figure 10 shows on the contrary*
*a substantial increase of Al concentration through the Holocene. Normalizing*
*systematically with this element may actually produce biased interpretations.*

Al normalization is extensively used in geochemical studies. The conservative elements
are not affected by chemical or biological processes, but affected by physical; it does
not mean that their concentration does not change. It is important to estimate the
authigenic enrichment of the elements. This process occurs in situ and in some way
depends on the metal fluxes, but the environmental conditions determine the enrichment
of these elements, see Calvert and Pedersen (2007), Tribovillard et al., (2006), Böning
et al. (2004, 2005, 2009) among others. This allows discriminating between enrichment
and terrestrial input; therefore the variability of Al can imply variability in some elements with greater terrestrial impacts than other processes.  Some elements can be used as indicators of terrigenous inputs and their variability can display whether the variability in sedimentary records accounts for contributions from land or for changes in primary productivity or redox conditions.   Therefore, each element must be normalized to Al or Ti, which is also useful to remove the effect of variability produced by changes in grain size. There is some concern about the use of Al and Ti for this purpose. However, caution must be used when using Al or Ti for the interpretation of metal distributions.

The normalization is not related to ICPMs technique. We do not use laser ablation.

*I recommend to use the accumulation rate from the age model and absolute Me concentration to calculate metal fluxes to the sediment.*

The accumulation rate could be a choice but is highly influenced by the age model used. Therefore, a better choice is to use the metal/Al ratio instead of the accumulation rate to establish authigenic enrichment. Accumulation does not depend on the fluxes, it depends on other on site factors that are useful to decipher the redox conditions at bottoms.

*Since Al has mainly a continental origin, ratios with Al is informative for elements whose flux is related to productivity to discuss relative contribution of marine vs terrestrial contributions in the sediment.*

All elements have an earth crust origin. While some elements follow different cycles, like nutrient type elements, they are incorporated into marine organism and deposited on the bottom when primary production settles down. After that, the elements follow other mechanism that allow for their enrichment, depending on their affinity to sulfides, for example. Therefore the normalization with Al is appropriate.

*Finally, the usefulness of the enrichment factors is not obvious. Figure 9 is barely discussed. In addition, I wonder if wetland sediments are really representative of crustal metal concentrations since they also contain organic matter.*

We decided to add a table (table 5) with the most relevant information to indicate that the variations in metals during periods of higher/lower productivity (based on opal accumulations rate) are due to authigenic enrichment, which in turn is a consequence of changes in redox conditions and not variations of continental inputs.

*- Geochronology*
*L248: Calpal2007_HULU calibration curve is an odd choice for radiocarbon calibration.*

We made corrections in the text; we use Clam2.2 program.

*It is also inconsistent with L255 in which Marine13 is mentioned (which is the correct calibration curve to use). There is a couple of issues with the regional radiocarbon reservoir age used for calibration. First, the method to calculate it is not correct. 14C reservoir age should be calculated in the 14C age scale, not in the calendar scale as it was done here. dR is the difference between the marine sample 14C age and the 14C*

*age that corresponds to the absolute age (here obtained from the 210Pb model) using*
*the Marine13 curve. See Southon et al. (1995) for details on the technique. The dR*
*value obtained here is larger than any dR values obtained previously on the Chilean*
*coast Authors should read and use Ortlieb et al., 2011; Carré et al., 2016; and*
*Merino-Campos et al., 2018. The latter reference presents 37 prebomb dR values all*
*along the Chilean coast measured with a reliable technique. Using a value from this*
*publication would be more reliable. The first 2 references show changes in dR values*
*through the Holocene that should also be discussed. Finally, instead of BC/AD, ages*
*should all be presented in the BP scale as it is usual in paleoceanography for*
*Holocene studies.*

We added an explanation in the text. We think upwelling waters are affecting the age of
foraminifers in our cores sites; other records at deeper areas have also used a DR ~ 400
years (De Pol-Holz, 2007). The samples of Carré and Meirno-Campos are submareal
species that live at shallows depths (<30 m), not highly affected by the upwelling. We
resorted to the method used by Sabatier et al., (2010) and we added a table for
informational purposes (table 3). Our estimations considers two pre-bomb data at 5 and
10 cm depth in the sediment cores from Guanaqueros and Tongoy bays; the ages from
210Pbxs correspond 499 and 448 years BP (Reimier et al., 2013) and were compared
with radiocarbon ages from foraminifers at the same depths. In both cases we obtain
similar results, therefore we decided to maintain the original age models, considering
441 years as a local reservoir. These values should correspond to the direct effect of old
upwelled waters in agreement with oceanographic conditions on the sampling stations.

We changed all Cal AD/BC to Cal BP ages.

**Discussion:**
**L505-L514: unclear**
**L521-L536: the discussion about d13C values is unclear, in part because there**
**seem to be a confusion between Total organic carbon(TOC) in the water column**
**and suspended particulate organic Matter (SPM). Is it possible that the difference**
**between d13C values in the water and in the sediment are due to the difference**
**between TOC and SPM? A preferential degradation of 13C enriched particles is**
**mentioned (L528-529): could you support this with a reference?**
**L563-L568: the discussion about K is not very convincing. A reference about the**
**detritic origin of K is needed. Ca could also have a detritic origin so close to the**
**shoreline. Al, Fe are also clear terrestrial input indicators. Why not discuss them**
**together?**
We added the reference and Ca is normally used as an indicator of marine productivity
versus K, which is a major element that has no implications on marine productivity.
Fe is more complicated due to its double origin. In all cases, we attempted to use the
best proxy in order to interpret each process.
**L602-L606: references needed**

**Section 5.3 should be shortened. It is somewhat redundant with other discussion sections and the result section.**
**- Climatic interpretations**

We rewrote this section completely.
**This section lacks in-depth discussion. The results here should be compared to published results to understand how they contribute, support or contradict existing hypothesis about millenial oceanographic variability in Chile.**
**L720-L723: "past changes are analogue with the present meridional displacement of the ITCZ and the SPCH". This should not be taken as a fact. It is only a hypothesis used as an interpretation model.**
**L744-L747: this part is unclear and sounds contradictory (a poleward shift of SWWshould not promote humid conditions in central Chile). In addition, this is a model result.**

**Why not compare with existing paleoenvironmental and paleoceanographic data?**

**There is a series of sediment cores that document past oceanographic conditions in the Peru-Chile upwelling system during the Holocene. This includes Lamy et al (1999, 2001, 2002, 2010), Kim et al. (2002), Hebbeln et al. (2002), Rein et al. (2005), Salvatecci et al. (2014, 2016). On a regional scale, the data presented here confirm a La Niña-like situation in the early to mid-Holocene, which is in agreement with previous datasets including Koutavas et al. (2002), Fontugne et al. (2004), Conroy et al. (2008); Carré et al. (2012), and model experiments such as Brown et al. (2008); Braconnot et al. (2012), Luan et al. (2015). This list is clearly not exhaustive.**

The first version of this manuscript considered the information of studies by Lamy, Hebbeln, Salvatteci. We added others from the list suggested, but focused on the range of time that covered our study and on Chile's central margin. Some studies about southern Peru were also cited. We re-wrote the paleoclimate section and considered the main studies focused from mid-Holocene in the region, identifying the main environmental conditions prevailing during the maximum periods of primary productivity.

**The influence of ENSO variability needs obviously to be discussed. It is here briefly mentioned in the text, appears in the key words, but there is no discussion. Data on past ENSO activity do exist (Koutavas et al., 2006; Cobb et al., 2013; Carré et al., 2014) and they need to be included in the discussion if the role of ENSO in the presented data is to be evaluated.**

We discussed some details about ENSO. Our study is not focused on the ENSO variability, but on changes in primary productivity and redox conditions.

**Figure 2: what about st14? Font on Y scale too small Figure 3: SPM is not the same as TOC Figure 5: it is not clear which curve is grain size and which is susceptibility**

We have no oxygen data for st14 and we made the corrections in the figures.

**Figure 6: Al and Fe are both related to terrestrial input. What information does Fe/Al provide?**

It could show enrichment of Fe by oxidation.

**Figure 9: This figure is not commented in the text. EF calculation does not**

**seem useful.**

We changed it for a more informative and brief table.

Answers to referee 2
*The authors present a large range of biogeochemical and microfossil proxies and the*
*results are worth to be published. However, I agree with referee #1 that the discussion*
*of the results is not sufficient and needs substantial alteration, more in-depth*
*interpretation of the own data as well as comparison to relevant literature.*
Initially, we used several references for the study area. To favor comparable results and
since most works extend further back from the Holocene, we avoided using research
that went beyond the period of time during which we did our work.
We also tried to focus on the area of study. Much work has been carried out far north or
far south, with different responses to atmospheric and oceanographic forcing. These
results are not comparable since our objective was to identify changes in productivity
and changes in redox conditions in the past. While our findings complement the main
results of other studies in the Chilean continental margin, they also provide information
on the possible effects on atmospheric-oceanographic changes in one of the most
important upwelling areas of the Chilean continental margin.
We modified the introduction, the last part of the section about the study area and the
last part of the discussion regarding paleoclimatic interpretations, and we used several
of the references you suggested.
. We believe this improves the discussion of our findings.
**The Discussion is too short especially in comparison to the methods section. There**
**is actually room for more detailed interpretations, for example the Nitrogen**
**isotopes**
**are not explained or discussed at all. The connections between sentences and**
**paragraphs are often weak or confusing, there is some refinement needed and the**
**reader must be led more through the text, especially the discussion. As the**
**manuscript reads now it appears as you randomly choose some results to discuss**
**one after the other. For example see paragraphs starting in lines 521 and 537. To**
**further strengthen the discussion add more comparisons to local studies such as**
**(Contreras et al., 2007; Díaz-Ochoa et al., 2010; Fukuda et al., 2013; Mohtadi et**
**al., 2008; Ortega et al., 2012).**
The discussion in point 5.1 refers to the biogenic versus the terrigenous contributions.
First, the organic component has been discussed when we talk about TOC and stable
isotopes; then the inorganic, when we comment on the susceptibility and magnetic and
metals.
A paragraph has been included regarding the implications of the changes in the 15N
distribution and we have added citations, such as DePol-Holz. Contreras has not been
considered because his work is focused on superficial sediments and temporal variation
of 15N. The study by Mohtadi et al 2008 does not highlight any results for the mid-
Holocene; their core's data dates back 6 Ka and our study could be comparable only in
some parts of their charts. It could be compared with our study only in a few points of
its graphs. The core used off the Coquimbo area corresponds to depths in the slope
under the influence of Intermediate Antarctic Water, as this study is focused on
studying the changes of this water body after the Last Glacial Maximum. We included
Ortega as a work submitted since the manuscript is forthcoming. However, this
manuscript is based on the analysis of a short core of the Tongoy Bay. . In the case of
Díaz-Ochoa, the work focuses on the last 200 years, the implications of which do not match our records. Additionally, the oceanographic dynamics in Mejillones are considerably different from those in Coquimbo.

*Through the whole manuscript the authors refer to suboxic/anoxic conditions, however, the values given for the water station 16 seems to be well above the suboxic value of <0.2 ml/L. For station 1 it's really hard to distinguish if the values may be lower sometimes. I think the value ranges for oxic/suboxic/anoxic need to be given in the introduction. Also, while water values are presented the oxygen levels discussed refer to the sediment which needs to be made much clearer. Just because you have low*
*oxygen in the water column this does not necessarily make the underlying sediments anoxic.*

In the present conditions, bottom waters are normally suboxic. Therefore, in our sedimentary records the enrichment of metals like U, Re and Mo decrease dramatically. When we speak of anoxia, we are referring to periods in the past for which there are no oxygen records, but it is deduced from the distribution of proxies like U, Re and Mo, which point to a very low content of oxygen and even sulfides, suggested by a large enrichment of Cd and Mo. Strictly speaking, all the sediments are anoxic; the penetration of oxygen is only a few mm when the bottom waters are suboxic. In our case, the high deposition of organic material generates seasonal conditions of anoxia on the sediments due to high consumption during its degradation. Then these sediments are under the effect of anoxic conditions which seem to prevail during the mid-Holocene.

**Add a discussion of the nitrogen isotope data. And compare to previous studies, such as (De Pol-Holz et al., 2006; 2007; Verleye et al., 2013).**

We added a paragraph considering the works of De Pol-Holz and others that help establish the effect of the OMZ and upwelling on our site, thereby complementing our interpretations of metal distribution. Lines 537 to 551. No interpretations on nitrate reduction variability could be done since our core covers from the mid-Holocene onwards and because no major changes in 15N are expected during this period.

*General remarks:*
*The figures are often not focused, the labels are too small, and in figure 10 the age should be plotted on the y-axis as in the other figures.*

We corrected the figures, except the figure on moisture pollen which displays better horizontally.

*I would like to see a more comprehensive conclusion, so far it's more a summary. Suggestion: try to reduce information in methods and results section. Is the exact.*

We modified the conclusions.

*Munsell chart colour really needed?*
Is a good guide for establishing the general composition of the sediments

*For example get rid of Line 181 to 183.*
These lines briefly explain how the cores were processed.

**Specific remarks:**

*Figure 1: Please add the surface circulation for the area.*

We added an outline of bay circulation based on studies available; some patterns are under study and yet to be defined.

This is relevant to understand the arguments raised.

*Figure 2: unsharp and colors are hard to distinguish, this needs revision, I suggest to use a color range that is more appropriate to highlight the DO values of the low end of*
*the scale more. Numbers in this plot need to be larger as well.*

We made the corrections and chose the best colors allowed by the Matlab program.

*Figure 5: I think the accumulation rates for TOC should be given here instead of just (%), further please add the core number directly behind a.) and b.) in the figure.*

The organic carbon accumulation was included in the figures 5a and 5b and in the text, but the sedimentation rate does not change much, so to do a calculation with a relatively constant number does not contribute mostly to the results.

*Figure 10: I suggest to also put the Age on the Y-axis here as in all the other figures.*

*Line 35 – add "The" before Coquimbo*
*Line 78 to 83: rephrase, you cannot refer to "these boundary current ecosystems" in one sentence and then explain it afterwards.*
*Line 131-132: maximum Chl a concentrations of ...*
*Line 209-216: remove this paragraph, the section is already long and you only list the following chapters here.*
*Line 218: change the comma to a dot.*
*Line 384- 390. This was a bit confusing as a southern and northern area are introduced,*
*but both cores studied are in the southern area?*
*Line 720: "Past environmental changes are analogue..." please specify these changes clearly here.*
*Line 724: "in this regard", it's not clear what you are referring to*
*Line 726: Studies based on pollen records ... There is a citation missing here!*
*Line 747-51: rephrase, improve the connection to the sentence before by first saying that you see indications of higher continental inputs due to increased rainfall, than which of your data shows this and which other studies support this observation. I further suggest to split this sentence in two.*
*Line 759: rephrase "peak drying"*

Most of these lines were modified. We checked grammar mistakes and we modified several lines and paragraphs in order to answer to the comments by both referees.

This marked-up version include all suggestions. Main changes were highlighted.

**Reconstructing past variations in environmental conditions and paleoproductivity**

**over the last ~8000 years off Central Chile (30° S)**

Práxedes Muñoz[1,2], Lorena Rebolledo[3,4], Laurent Dezileau[5], Antonio Maldonado[2],

Christoph Mayr[6,7], Paola Cárdenas[4,8], Carina B. Lange[4,9,10], Katherine Lalangui[9], Gloria

Sanchez[11], Marco Salamanca[9], Karen Araya[1,5], Ignacio Jara[2], Gabriel Vargas[12], Marcel

Ramos[1,2].

[1]Departamento de Biología Marina, Universidad Católica del Norte, Larrondo 1281,

Coquimbo, Chile.

[2]Centro de Estudios Avanzados en Zonas Áridas (CEAZA), Coquimbo-La Serena,

Chile.

[3]Departamento Científico, Instituto Antártico Chileno, Punta Arenas, Chile.

[4]Centro FONDAP de Investigación Dinámica de Ecosistemas Marinos de Altas

Latitudes (IDEAL), Universidad Austral de Chile, Campus Isla Teja, Valdivia, Chile.

[5]Laboratoire Géosciences Montpellier (GM), Université de Montpellier, 34095

Montpellier Cedex 05, France.

[6]Institut für Geographie, FAU Erlangen-Nürnberg, 91058 Erlangen, Germany.

[7]Department of Earth and Environmental Sciences & GeoBio-Center, LMU Munich,

80333 Munich.

[8]Programa Magister en Oceanografía, Universidad de Concepción, casilla 160C,

Concepción, Chile.

[9]Departamento de Oceanografía, Facultad de Ciencias Naturales y Oceanográficas,

Universidad de Concepción, Casilla 160C, Concepción, Chile.

[10]Centro de Investigación Oceanográfica COPAS Sur-Austral, Universidad de

Concepción, Casilla 160C, Concepción, Chile.

[11]Universidad de Magallanes, Punta Arenas, Chile.

[12]Departamento de Geología, Universidad de Chile, Santiago, Chile.

*Correspondence*: Práxedes Muñoz (praxedes@ucn.cl)

**Abstract**

The Coquimbo (30°S) region, in the North-central Chilean Coast, is characterized by relative dry summers and a short rainfall period during winter months. The wet-winter climate results from the interactions between the Southern Westerly Winds and the South Pacific Anticyclone (SPA). Interdecadal climate trends are mostly associated with El Niño-Southern Oscillation (ENSO), which produces high variability in precipitation. With the aim of establishing past variations of the main oceanographic and climatic features in the Central Chilean coast, we analyze recent sedimentary records of a transitional semi-arid ecosystem susceptible to environmental forcing conditions. Sediment cores were retrieved in two bays, Guanaqueros and Tongoy (29−30°S), for geochemical analyses including: sensitive redox trace elements, biogenic opal, total organic carbon (TOC), diatoms, stable isotopes of organic carbon and nitrogen. Three main periods of increased productivity were established: (1) > cal BP 6500, (2) cal BP 2100 – cal BP 4600 and (3) during recent time (CE 2015) – cal BP ~260. The first period was conspicuously high during the main dry phase concomitant with high fluxes of organic compounds to the bottom and suboxic-anoxic conditions in the sediments. This period reached a maximum at cal BP ~6500, at the time of the maximum Holocene transgression reported for the zone (~ cal BP 6380), followed by a continuous increase in moisture levels, low primary productivity and a more oxygenated environment towards the present, being remarkably stronger in the last 2000 years. We suggest that this might be associated with greater El Niño frequencies or similar conditions that increase precipitation, concomitantly with the introduction of oxygenated waters to coastal zones by the propagation of equatorial origin waves.

Keywords: paleoproductivity, paleoredox, trace metals, diatoms, opal, organic carbon, Coquimbo, SE-Pacific

**1. Introduction**

The northern-central Chilean continental margin (18−30°S) has distinct zones of intense upwelling highly influenced by topographic features (Figueroa and Moffat, 2000). As a result, high primary production (0.5-9.3 g C m$^{-2}$ d$^{-1}$) are developed off Iquique (21°S), Antofagasta (23°S) and Coquimbo (30°S) (González et al., 1998; Daneri et al., 2000, Thomas et al., 2001). This productivity takes place close to the coast above the narrow continental shelf, allowing the development of important fisheries and accounting for up to 40% of total annual catches (Escribano et al., 2004 and references therein).

This high productivity maintains a zone of low dissolved oxygen content along the Chilean margin, reinforcing the oxygen minimum zone (OMZ) that develops along the North and South Pacific Ocean, where their intensity, thickness, and temporal stability vary as a function of latitude (Helly and Levin, 2004, Ulloa et al., 2012). To the north (e.g. 21°S) and off Peru, the OMZ occurs permanently, can extend into the euphotic zone and, in the case of northern Chile and southern Peru, shows no significant interface with the benthic environment due to the presence of a narrow continental shelf (Helly and Levin, 2004).

Past changes in the productivity and oxygenation of bottom waters at different timescales have been evidenced in the SE Pacific through sedimentary records that cover from the Last Glacial Maximum (cal BP 22,000 −18,000) to the present. Different climate-ocean drivers have been proposed to account for these changes. For instance, latitudinal movements of the Southern Westerlies Winds (SWW) and the Antarctic Circumpolar Current (ACC) have been suggested as potential mechanisms (Hebbeln et al., 2002; Lamy et al., 2001; 2002; 2010). In addition, changes in the intensity and position of the Southeast Pacific Subtropical Anticyclone (SPSA) from seasonal, to interdecadal timescale have effects on wind stress and water mass circulation (Ancapichún and Garcés-Vargas, 2015), and therefore past variability in the SPSA has been used to explain changes in paleoceanographic features of the  SE Pacific such as the intensity of upwelling, and circulation patterns responsible for the nutrient supply (Marchant et al., 1999; Hebbeln et al., 2002; Dezileau et al., 2004; Romero et al., 2006; Mohtadi et al., 2008; Gutiérrez et al., 2009; Saavedra-Pellitero et al., 2011; Muñoz et al., 2012). Past climate-upwelling fluctuations at millennial timescales has also been linked to the austral insolation, which influence Antarctic sea ice extent and the Hadley cell, this latter an important forcing to the latitudinal cycle of the ITCZ (Intertropical

Convergence Zone; Kaiser et al., 2008 and reference there in). This variability produces humid and arid conditions along the SE Pacific where the intensity of wind has a key role for the upwelling and hence productivity. On top of all this, an important driver of modern ocean-atmosphere conditions in the South East Pacific is the El Nino/Southern Oscillation (ENSO), which has a major impact on modern marine productivity (Escribano et al., 2002). Paleo-ENSO reconstructions indicate attenuated ENSO events before the mid-Holocene (last 5000 years) and increasing from this period towards the present (Marchant et al., 1999; Koutavas et al., 2006; Vargas et al., 2006), consistent with paleoceanographic and paleoclimate interpretations (Rodbell et al., 1999; Rein et al., 2005). Heavy rainfall episodes in the south East Pacific normally occur during strong El Niño conditions (Montecinos and Aceituno, 2003), increasing the river flux and producing flood debris (Garreaud and Rutllant, 1996). These episodes have been recorded in sedimentary records off northern Chile and southern Peru, establishing a teleconnection which has operated since the mid-Holocene, and identifying the modern manifestation of El Niño starting at ~5300 − 5500 cal BP (Vargas et al., 2006).

The effect of climate variations on primary productivity and biogeochemical cycles could have different responses. For instance, the increase in land-sea thermal contrast in North-Central Chile enhances upwelling and with it, exported production (Vargas et al., 2007). Other evidence, however, suggest that the intrusion of warmer oligotrophic water reduce primary productivity, as observed during the 97-98 ENSO event (Iriarte and Gozález, 2004). Furthermore, in South central Chile (36°S) the oxygenation of bottoms was clearly detected during the 97-98 El Niño event, changing the geochemical conditions of surface sediments and macrofauna composition. These disturbances may extend considerably to the south, with implications persistent for many years and impact the sedimentary records of several proxies (Sellanes et al., 2007; Gutiérrez et al., 2006). Our work focuses on the past variations of the environmental conditions and marine productivity in sedimentary records from a transitional semi-arid ecosystem of Central Chilean coast (30°S), an area highly susceptible to oceanographic and climatic forcing. The study area (Fig. 1) provides an adequate platform to observe environmental variability at different time scales. We were able to identify wet/dry intervals, periods with high/low primary production, and changes in redox conditions at bottoms through inorganic (trace metals) and organic proxies.

[revised manuscript text omitted]
 BP 3300 − 4000 at core BGGC5, not observed at core of Tongoy, instead it showed an increase around cal BP 6500 −7800 . Manganese did not show any clear trend.

A second element group (metal/Al ratios), including Cd, Ni and P (related to primary productivity and organic fluxes), showed a similar pattern than Mo/Al towards the bottom of core BGGC5, i.e. the highest values around cal BP 6500 and a constant reduction towards the present. A third group, consisting of Ba, P and Ca, exhibited a less clear pattern. The Cd/Al and Ni/Al ratios in core BTGC8 showed only slightly decreasing values, and the maximum values were very low compared to the BGGC5

core. The same pattern is observed for other elements. Metal/Al ratios for Ba, Ca and P

were lower and presented a long-term decreasing pattern towards the present.

An exception to the previously described patterns was Cu/Al, which peaked at cal BP

~3600 − 3700 and showed a conspicuous increase in the past ~150 years. This was also observed at core BTGC8, but with lower concentrations than at core BGGC5.

**5. Discussion**

**5.1. Sedimentary composition of the cores: terrestrial *versus* biogenic inputs**

The sediments in the southern zones of the bays constitute a sink of fine particles transported from northern areas and the shelf (Fig. 5a, 5b), responding to the water circulation in the Guanaqueros and Coquimbo Bays described as bipolar, i.e. two counter-rotating gyres which are counterclockwise to the north and clockwise to the south (Valle-Levinson and Moraga, 2006). This is the result of the wind and a coastline shape delimited by two prominent points to the north and south. In the case of Tongoy Bay (the southernmost bay of the system), circulation shows a different pattern due to its northern direction compared to Guanaqueros Bay, which opens to the west. The cyclonic recirculation in Tongoy Bay seems to be part of a gyre larger than the Bay's circulation (Moraga et al., 2011). This could explain differences in sediment particle distribution and composition between the bays. At Tongoy Bay, there are less organic carbon accumulation ($< 3$ g m$^{-2}$ yr$^{-1}$), siliceous microfossils and pollen (Figs. 5b, 6 and 7). Similarly, in Guanaqueros Bay TOC contents are only slightly higher ($> 2$ %), especially between cal BP 3700 and 4000 and before cal BP 6500 ($\sim 4$ %) but higher accumulation rates around 7 and 16 g m$^{-2}$yr$^{-1}$, respectively (Fig. 5a). However, sediments there contain enough microfossils to establish differences in primary productivity periods and also provide a pollen record evidencing the prevailing environmental conditions.

The stable isotopes measured in the study area were in the range of marine sedimentary particles for southern oceans at low and mid latitudes ($\delta^{13}$C; -20 ‰ – -24 ‰; Williams 1970; Rau et al., 1989; Ogrinc et. al. 2005), and slightly lower than the TOC composition at the water column (-18 ‰, Fig. 3). This suggests that the organic particles that settle on the bottom are a more refractory material (C/N: 9−11), remineralized during particle transportation and sedimentation. This results in lighter isotopic compositions, especially at core BTGC8. Furthermore, the $\delta^{15}$N and $\delta^{13}$C of settled particles are more negative at surface sediments due to a preferential degradation of molecules rich in $^{13}$C and $^{15}$N, resulting in more negative values and higher C/N ratios at sediments than in suspended particles (Fig. 3, 5a, 5b). However, this is also due to the stronger diagenetic reactions observed near the bottom layer (Nakanishi and Minagawa, 2003). Thus, these sediments are composed by winnowed particles transported by water circulating over the shelf, and the isotopic variations should not establish clearly the contribution of terrestrial inputs.

Otherwise, the isotopic composition of upwelled $NO_3$ (De Pol-Holz et al., 2007) could influence the variability of $\delta^{15}N$. Values for $\delta^{15}N$ at northern and central Chile are in the range of those measured at BGGC5 core (~11‰; Hebbeln et al., 2000, De Pol-Holz et al., 2007), resulting by the isotopic fractionation of $NO_3$ during nitrate reduction within OMZ leaving a remnant $NO_3$ enriched in $^{15}N$ (Sigman et al., 2009; Ganeshram et al., 2000 and references therein). In this case, the BGGC5 core sediments represent the effect of the nutrient supply by the upwelling and the influence of the OMZ over the shelf resulting in $\delta^{15}N$ of 9 − 10‰. At BTGC8 sediment core, lower values (<8‰) at greater depths within the core should represent the mixing with light terrestrial organic material (Sweeney and Kaplan, 1980), due to the nearest position of a permanent small wetland at southern site of Tongoy Bay. Pachingo wetland material showed $\delta^{15}N$ of 1 − 8‰ (Muñoz et al., data will be published elsewhere) in the range of sedimentary environments influenced by terrestrial runoff (Sigman et al., 2009). In the same sense, at most of the cases, lower TOC is correspondingly with lighter $\delta^{15}N$ values, and also with the highest C/N ratios suggesting the mixing with continental material (Fig. 5b).

[revised manuscript text omitted]
. In this regard, the Holocene period features a series of wet and dry phases resulting from millennial-scale SWW changes (Hebbeln et al., 2002; Lamy et al., 1999; Maldonado and Villagrán, 2002). In particular, pollen records from the southern coastal areas of Coquimbo (32°S) indicate that wet conditions predominated before cal BP 8700, which brought the expansion of swamp forests areas along the coast (Maldonado and Rozas, 2008; Maldonado and Villagrán, 2006). This scenario occurred concomitantly with diminished rainfalls, regional aridity and strong southerly winds consistently with La Niña-like conditions prevailing during the Early Holocene along the arid and semiarid coasts of Chile (Vargas et al., 2006; Ortega et al., 2012), that would have driven increased coastal humidity associated to coastal fogs favored also by a relatively low sea level position with respecto to the present (Ortega et al., 2012). This wet period was followed by a long-lasting arid phase between cal BP 8700 and 5700. Regional aridity matches the relative dry conditions detected in the first portion of our pollen reconstruction from core BGGC5 in the Guanaqueros Bay, which is represented by relative low values of the Pollen Moisture Index in Fig. 9. Similarly, a general increase in regional precipitation after cal BP ~6000, observed in pollen records from the northern margin of SWW (Jenny et al., 2003; Maldonado and Villagrán, 2006) is broadly correlated with the observed long-term trend towards increased precipitation observed in the Pollen Moisture after cal BP ~6500 − 6700. This is also in agreement with Al and Pb, usually considered to be indicators of continental particles that enter to marine waters by fluvial or aerial transport (Calvert and Pedersen, 2007; Govin et al., 2012; Ohnemus and Lam, 2015; Saito et al., 1992; Xu et al., 2015). In our cores, these elements showed trend similar to the pollen record, i.e., a gradual rise in time, suggesting increased humid conditions during recent periods (Fig.9).

In addition, our records indicate long-term increases in grain size and K/Ca ratios and Fe over the last ~8000 years. These increases point to a higher continental inputs most probably caused by increasing rainfall events, which are an important source of sands and K in the northern Chilean margin at the present. At a regional scale, a trend towards increasing precipitations is also consistent with the occurrence of alluvial episodes since 8600 cal. BP, after a period of an almost quiescence of this phenomenon in the coastal region located just to the south of Tongoy bay (Ortega et al., 2012). Increments of Fe have been documented to provide a boost in primary productivity analyzed in sedimentary records (Dezileau et al., 2004). In our cores, a short-term increase in Fe concentrations is observed between cal BP ~4000 −3300 at the Guanaqueros core, whereas persistent high values are recorded in the Tongoy core between cal BP 6500 − 7800. These two increases coincide with periods with relatively high primary productivity based on the diatoms and opal distribution (Figs. 6, 8b). This correlation supports the role of Fe as promoter of coastal productivity in the past. However, we note that maximum productivity observed at cal BP ~6500 seems at odd with the overall dry environmental conditions evidence by the pollen data. An explanation for this discrepancy is that dry conditions were more likely associated with increases in SPSH activity in the region and consequently with higher upwelling (Frugone-Álvarez et al., 2017). The subsequent weakening in paleoproductivity after cal BP 6500 can be explained by a reduction in upwelling due to reduced SPSH activity and by the intrusion of less nutrient-enriched upwelled waters over the shelf, influenced by remote equatorial waves, as it is observed today. It is important also to considerate a possible influence of a sea level located in a lower position with respect to its present day position before 7000-6000 cal BP (Lambeck et al., 2014), that would have influenced productivity variations and also its recording at the sea bottom in these bays. The oldest transgressive deposits at Coquimbo Bay dated from BP 6380 and a gradual progradation of the coast from BP 2500 until present (Ota and Paskoff, 1993), changes the dynamic of the depositional environments due to a greater continental influence, observable in our cores.

The synchronism between highest productivity and dry conditions prior to ~cal BP 6500 highlights the role of the SPSA as an important driver of paleoproductivity changes in the coast of semi-arid Chile during the early portion of the Holocene Period. On the other hand, the pollen and trace element record show both a coherent pattern of increasing humidity and continental discharge over the last 7000 years. The driver of this long-term paleoclimate trend seems to be associated with past shifts in the position of the SWW. In particular, an equatorial displacement of the SWW during mid and later part of the Holocene period has been suggested by reconstructions from terrestrial and marine proxies (Veit, 1996; Lamy et al., 1999; Lamy et al. 2010).

Studies of coastal upwelling from the Central Peruvian and south Central Chilean coasts ($12 - 36$ °S) show that present-day wet/dry variability associated with El Niño Southern Oscillation exert an important influence on the bottom ocean oxygenation (Escribano et al., 2004; Gutiérrez et al., 2008; Sellanes et al., 2007). In this regard, OMZs is expected to be less intense during warm El Niño phases and vice versa. This link has been observed by recent studies, as warm events in the Tropical Pacific tend to be associated with low productivity and weak OMZ in the Peruvian coast (Salvatteci et al., 2014). An increase in the frequency of ENSO-like warm events could partly explain the reduction in productivity recorded after cal BP 6700 in our records, concomitantly with the coastal progradation (Ota and Paskoff, 1993). In this case, warm events in the eastern Pacific could have reduced the ocean productivity and organic fluxes from primary productivity and overall dropping oxygen consumption during organic matter diagenesis. In the light of these mechanisms, our results suggest more El Niño-like conditions during the latter part of the Holocene, an inference that is consistent with the available evidence for an increase in the frequency of El Niño events over the last 4000–5000 years (Conroy et al., 2008; Moy et al., 2002). We further note that present-day El Niño years are generally connected with increased westerly flow over central Chile including the semi-arid region (Montecinos and Aceituno, 2003), and therefore more frequent El Niño states during the latter part of the Holocene are also consistent with a long-term increase in precipitation revealed by the pollen and trace element data.

**6. Conclusions**

Our result indicates that the ocean circulation at our study sites seems to affect both places differently, leaving more variable grain compositions and higher TOC contents in the Guanaqueros Bay (core BGGC5) than in the Tongoy Bay (core BTGC8). This difference should be interpreted as an increase in the time of particle transportation resulting in grain size selection (more leptokurtic at core BTGC8), especially after cal BP 2000. Furthermore, in both bays, constantly decreasing TOC contents were observed after cal BP ~4000 to the present, probably due to higher oxygenation of the bay bottom in time.

Differences in redox conditions in our records could be reconstructed in detail, showing a clear decreasing trend in oxygen bottoms before the beginning of recent time (cal BP

[revised manuscript text omitted]

Moraga-Opazo, J., Valle-Levinson, A., Ramos, M. and Pizarro-Koch, M.: Upwelling-

Triggered near-geostrophic recirculation in an equatorward facing embayment, Cont.

Shelf Res., 31, 1991–1999, 2011.

Mortlock, R. A. and Froelich, P. N.: A simple method for the rapid determination of biogenic opal in pelagic marine sediments, Deep Sea Res. Part A, Oceanogr. Res. Pap.,

36(9), 1415–1426, doi:10.1016/0198-0149(89)90092-7, 1989.

Morse, J.W. and Luther, G.W.: Chemical influences on trace metal–sulfide interactions in anoxic sediments. Geochim Cosmochim Ac., 63, 3373–3378, 1999.

Moy, C.M., Seltzer, G.O., Rodbell, D.T. and Anderson, D.M.: Variability of El

Niño/Southern Oscillation activity at millennial timescales during the Holocene epoch.

Nature, 420(6912), p.162, 2002.

Muñoz, P., Dezileau, L., Dezileau, L., Lange, C.B., Cardenas, L., Sellanes, J.,

Salamanca, M.A., Maldonado, A.: Evaluation of sediment trace metal records as paleoproductivity and paleoxygenation proxies in the upwelling center off Concepción,

Chile (36°S)., Prog. Oceanogr., 92–95, 66–80, 2012.

Nakanishi, T. and Minagawa, M.: Stable carbon and nitrogen isotopic compositions of sinking particles in the northeast Japan Sea, Geochem. J., 37(2), 261–275, doi:https://doi.org/10.2343/geochemj.37.261, 2003.

Nameroff, T., Balistrieri, L. and Murray, W.: Suboxic trace metals geochemistry in the eastern tropical North Pacific, Geochim Cosmochim Ac., 66(7), 1139–1158, 2002.

Ogrinc, N., Fontolan, G., Faganeli, J. and Covelli, S.: Carbon and nitrogen isotope compositions of organic matter in coastal marine sediments (the Gulf of Trieste, N

Adriatic Sea): indicators of sources and preservation, Mar. Chem., 95, 163-181, 2005.

Ohnemus, D. C. and Lam, P. J.: Cycling of lithogenic marine particles in the US

GEOTRACES North Atlantic transect, Deep. Res. Part II Top. Stud. Oceanogr., 116,

283–302, doi:10.1016/j.dsr2.2014.11.019, 2015.

Ortega, C., Vargas, G., Rutllant, J.A., Jackson, D., Méndez, C.: Major hydrological regime change along the semiarid western coast of South America during the early

Holocene, Quaternary Res., 78, 513-527, http://dx.doi.org/10.1016/j.yqres.2012.08.002,

2012.

Ota, Y. and Paskoff, R.: Holocene deposits on the coast of north-central Chile:

radiocarbon ages and implications for coastal changes. Rev. Geol. Chile, 20, 25−32,

1993.

Paytan, A.: Ocean paleoproductivity, Encyclopedia of Paleoclimatology and Ancient

Environments, Encyclopedia of Earth Science Series, Gornitz, V. (Ed.), Kluwer

Academic Publishers. 2008.

Peacock, C.L. and Sherman, D.M.: Copper(II) sorption onto goethite, hematite and lepidocrocite: a surface complexation model based on ab initio molecular geometries and EXAFS spectroscopy. Geochim. Cosmochim. Ac., 68, 2623–2637, 2004.

Pizarro, O., Shaffer, G., Dewitte, B. and Ramos, M.: Dynamics of seasonal and interannual variability of the Peru-Chile Undercurrent, Geophys. Res. Lett., 29(12), 28–

31, doi:10.1029/2002GL014790, 2002.

Quintana, J.M. and Aceituno, P.: Changes in the rainfall regime along the extratropical west coast of South America (Chile): 30-43º S, Atmosfera, 25(1), 1 – 22, 2012.

Ramos, M., Pizarro,O., Bravo, L. and Dewitte, B.: Seasonal variability of the permanent thermocline off northern Chile, Geophys. Res. Lett., 33, L09608, doi:10.1029/2006GL025882, 2006.

Ramos, M., Dewitte, B., Pizarro, O. and Garric, G.: Vertical propagation of extratropical Rossby waves during the 1997–1998 El Niño off the west coast of South

America in a medium-resolution OGCM simulation, J. Geophys. Res., 113, C08041, doi:10.1029/2007JC004681, 2008.

Rau, H. G., Takahashi, T. and Des Marais, D. J.: Latitudinal variations in plankton

$\delta 13C$: implications for $CO_2$ and productivity in past oceans, Nature, 341, 516–518,

1989.

Reimer, P. J., Bard, E., Bayliss, A., Beck, J. W., Blackwell, P. G., Ramsey, C. B., Buck,

C. E., Cheng, H., Edwards, R. L., Friedrich, M., Grootes, P. M., Guilderson, T. P.,

Haflidason, H., Hajdas, I., Hatté, C., Heaton, T. J., Hoffmann, D. L., Hogg, A. G.,

Hughen, K. A., Kaiser, K. F., Kromer, B., Manning, S. W., Niu, M., Reimer, R. W.,

Richards, D. A., Scott, E. M., Southon, J. R., Staff, R. A., Turney, C. S. M. and van der

Plicht, J.: IntCal13 and Marine13 Radiocarbon Age Calibration Curves 0–50,000 Years cal BP, Radiocarbon, 55(4), 1869–1887, doi:10.2458/azu_js_rc.55.16947, 2013.

Rein, B., Lückge, A., Reinhardt, L., Sirocko, F., Wolf, A., Dullo, W-C.: El Niño variability off Peru during the last 20,000 years, Paleoceanogr., PA4003, doi:10.1029/2004PA001099, 2005

Rodbell, D.T., Seltzer, G.O., Anderson, D.M., Abbott, M.B, Enfield, D.B, Newman JH:

An approximately 15,000-year record of El Nino-driven alluviation in southwestern

Ecuador, Science, 283, 516 – 520, 1999.

Romero, O., Kim, J-H, Hebbeln, D.: Paleoproductivity evolution off central Chile from the Last Glacial Maximum to the Early Holocene, Quat. Res., 65, 519 – 525, 2006.

Saavedra-Pellitero,M., Flores, J. A., Lamy, F., Sierro, F. J., Cortina, A.:

Coccolithophore estimates of paleotemperature and paleoproductivity changes in the southeast Pacific over the past ∼27 kyr.

[revised manuscript text omitted]

---

## Referee Report (RR1)

Second Review of Muñoz et al., bg-2018-396

Summary
The revised version of the Manuscript "Reconstructing past variations in environmental conditions and paleoproductivity over the last ~8000 years off Central Chile (30° S) presents an improvement introducing and discussing their data in context of ocean-atmosphere interactions and more importantly comparison with and reference to previous data from the region. However, I still have two major points of criticism, (1) the introduction is still not well written enough and is missing information about proxies being applied to support previous observations and conclusions and more essentially the motivation of the Authors to select the study sites and the proxies ultimately utilized, (2) Although ta paleodiscussion was now added, the discussion is still mainly discusses each result point by point, appears unfocused and needs re-structuring.

Overall there is still a main question or motivation missing throughout the manuscript, there need to be some sentences added why the authors selected the study are and what they hope to improve in the paleoceanographic knowledge about the SE Pacific. I am not convinced by the paragraph (line 120-126) where the Authors introduce their work, there is little connection to what was written in the introduction before.

The Authors improved the introduction by adding more detailed information about the ocean-atmosphere dynamics relevant in the study area. Unfortunately, references to previous work is still too vague, for example just referring to "sedimentary records of several proxies". I think the authors deleted important information on how changes in the ocean-atmosphere dynamics are reflected within sediment records from the previous version. And thus, an introduction about what proxies are feasible to use for the authors research question is basically completely missing. Following on that, there is no information provided on what the others selection of proxies applied was based. Suggestions from my side how to improve the structure of the introduction can be found under the line-to line comments.

Furthermore, the structure is still a bit strange with specific information about the area, then explaining general observations from the SE Pacific This should be reversed, going from the big picture to the study area.

The Discussion of the new data presented by the Authors based on climatic changes and the comparison with previous studies significantly improved in section 5.4 climatic interpretations. However, I think the structure of the discussion needs still improvement. At the moment the different proxies are discussed successively, but this structure results sometimes in non-chronological description of the significant periods highlighted in the manuscript. I suggest to re-structure the discussion in first the modern conditions and afterwards the 3 time intervals (> 6 kyrs, 2.1 to 4.6 kyrs and recent to 260 yrs BP) and finish with section 5.4 climatic interpretations presented in the current manuscript. As the definition of these time intervals is also one of the mayor findings of the study, this structure would improve their significance to the reader.

The text needs still a lot of improvement. Paragraphs are often not properly connected to guide the reader and several grammatical errors are distributed throughout the whole text. Furthermore, the use of "decrease" and "increase" is often inappropriate, as there are no values given for comparison, for example the authors conclude in line 858 that nutrient-type elements are **reduced** at present and **higher** at cal BP 6500.

On the whole manuscript is too long and especially methods descriptions are too detailed. When applying commonly used methods it is sufficient too shortly describe the procedure and refer to the original publication. Detailed explanations are only needed if analysis vary from normal procedures. I suggested some superfluous information under the detailed comments to shorten the manuscript.

The figures were all improved following previous reviewer's suggestions, however Figure 9 presenting the Pollen record is still the only figure were age is given on the x-axis instead of the y-axis. The Authors didn't give a reason for not changing this, it would help comparing the data.

Line by line comments:

Line 34: change "in" to "at" and I don't the commas are needed here.

Line 46-47: rephrase "The first period was conspicuously high..." it is not the period that is high but the productivity during this period, change to something like "The productivity during the first period was conspicuously high..."

Line 49: rephrase "this period reached a maximum at ..." what maximum was reached, needs to be spelled out

Line 52: again rephrase ", being remarkably stronger in the last 2000 years" are you referring to oxygen levels?

Line 64: change "are developed" to "develops"

Line 69: rephrase, second sentence in a row starting with "this high productivity..."

Line 71: Is "where their intensity, ..." supposed to refer to the OMZ? Then please use the singular i.e. "where it's ... "

Line 91: change to "have also been linked"

Line 92: change to "influences"

Line 93: change "this latter an important forcing" to "which acts as an important forcing"

Line 94 onwards: this connection is confusing, you refer to "this variability" producing humid and arid conditions, you seem to refer to changes in the processes (i.e. sea ice

extent, Hadley cell and latitudinal position of the ITCZ. However, you only mention fluctuations in upwelling in the previous sentence. I suggest changing the beginning of the sentence into "Changes/Variability in the austral insolation and the related processes/mechanisms produce..."

Line 96: change "on top of all this" into "An additional important driver...".

Line 120 to 126: This is a summary of what you did, I rather expect here a paragraph about why you selected your core positions on the basis of the introduction you give

Line 176 to 179: delete, these are common procedures and you are not referring to this information in the following.

Line 184 to 211: remove text. All this information is repeated in the following sections, or if not can be added to the appropriate section for each proxy.

Line 241: remove "ages were calculated using 5568 (yrs) as the half-life of radiocarbon" superfluous.

Line 260: I think you can shorten 3.4 Trace Metal analysis. I am not familiar with the method myself but the descriptions appear extremely detailed and could be shorten, as it is commonly applied.

Line 293: section 3.5 TOC and stable isotopes can be significantly shortened, it is sufficient to explain how much material was weighed into tin and silver capsules respectively and where in what machine analysis have been performed, and of course the external reproducibility of measurements must be given. It is not needed to explain the entire procedure of the measurements. In short rephrase the sentence in line 296 to 298 and remove the text until line 306.

Line 482: add "are" behind Trace metals

Line 511: "This is the result of the wind" please elaborate a bit more in what way this is caused by the winds

Line 532 and 533: change "more negative" to "lighter"

Line 539 and thereafter: change $NO_3$ to $NO_3^-$ or use "nitrate"

Line 630: not sure what the Authors are referring to as "it"

Line 716: remove "notorious"

Line 777: change "trend" to "trends"

Line 840: Change "Our result indicates" to "our results indicate"

Line 843: change to "we interpret this difference as..."

Line. 848: add how these redox conditions have been reconstructed.

Line 856: change "where" to "were"

Line 857: get rid of "the presence of"

---

## Author Response (AR2)

**Reviewer #1**

**The manuscript has been greatly improved. All the methodological concerns that I had have been clarified. Some work is still needed, however, before the manuscript can be published.**
**Some general comments:**
**- The text is too long and should be reduced by 20-30%. The results and the discussion in particular should be cleaned of all the textbook comments and context generalities and be focused on the data.**
**The data interpretation is not satisfying. It is generally very confusing and some important issues are not discussed.**
**- The climate interpretation takes shortcuts more based on widely repeated scenarios than their data.**
**- *The differences between the two cores are described. However, the authors do not acknowledge or discuss the fact that these differences, which underline the importance of local processes, greatly limit the geographical extent of the interpretations that can be made from those records. The authors draw large conclusions from the opal and diatom records in one core, which are precisely the proxies without equivalent in the other core.***

The extension of the manuscript was reduced substantially. The introduction began with a review of the most important climatic and oceanographic agents in the region and how they are affected by the decadal-inter-decadal, inter-annual and seasonal variations. Then, the main oceanographic characteristics of the area are explained; for example the upwelling and minimum oxygen zone, followed by an explanation of how the variability of the key parameters accounts for their functioning: i.e. primary productivity and oxygen content can be observed in sedimentary records. We also explain the objectives of the study and then focus on the local characteristics of the study area, for which we propose a work hypothesis based on its special features. In the section on method, only the principle of each method was considered and superfluous details that can be found in the cited papers were avoided. The results were reduced to main findings avoiding superfluous descriptions. The discussion was based on the most important aspects that describe the geochemistry of the area, their functioning, and how the sedimentary records show the most important changes in terms of primary productivity, oxygenation of the bottoms and the relationship of these parameters to climatic changes that have already been described for the region. In this sense, we have tried to better clarify the objectives of this work. However, our results are not aimed at establishing a climate model and rather show important aspects of the variability of this region's main oceanographic features in the past 8,000 years. Take for example the case of minimum oxygen variability –as it has been affected over time– which has implications for the climate change scenario as past climate changes impacted the content of oxygen in bottom waters thereby changing the dynamics of biogeochemical processes in an area under the influence of one of the most important upwelling centers in north-central Chile. This is a partially permanent upwelling system developed in an area known for its climatic transition (30°S), between the northern super-arid zones and more humid climates heading south. .

Notably, there is no information about oxygen condition variability at bottoms in the Chilean margin, which is closely related to the oxygen minimum zone strength, except for some previous work off Concepción (36°S) covering the past 2000 years. Our cores are showing the variability of oxygenation in bottoms for the past 8000 years in a shelf area highly influenced by oceanic circulation due to a very narrow continental shelf.

The regional representativeness of our results lies in the fact that we are working in a zone with semi-permanent upwelling which has an impact on a larger area of Chile's north-central area, which in turn is impacted by the proximity of the oxygen minimum zone extending along the continental margin. The entrance of oxygen deficient water is relatively variable, causing periods of significant anoxia within the bays. The way in which this process is triggered is still a matter of study. Therefore, oceanographic processes observed in relatively shallow waters are representative of processes at a regional scale and their sediments may well account for regional oceanographic and climatic changes.

The difference observed between cores is not contradictory; represent different aspects of the area, which as a whole explain climatic and oceanographic changes, thus making the sedimentary records of both cores complementary. These differences are justified by the higher amount of continental material received by core BTGC8 in comparison with core BGGC5. Circulation within the bays facilitates the deposition of organic matter derived from primary productivity at Guanaqueros Bay. This is due to bay circulation which seems to involve a large area of a "bay system", from Lengua de Vaca Point to the south, towards Punta Choros to the north. Water circulation in the southern part of Tongoy Bay, close to core BTGC8, facilitates the movement of particles outside the bay, which could return from the north reaching the southern part of Ganaqueros Bay, close to the position of core BGGC5. However, the extension of the clockwise circulation has not been defined yet and studies to support this assumption are still pending. This was underlined in the study area section and addressed in the first section of the discussion. Furthermore, the effect of terrigenous material in the Tongoy area due to the discharge of several creeks during major flooding events has been previously observed in sediment cores retrieved close to our sampling site, showing evidence of increasing El Niño events from cal BP 3500 (Ortega et al., 2019.) The continental material is diluting other organic and inorganic proxy records reducing the concentrations, i.e. diatoms valves and nutrient-type elements. Even with this effect, the trace metal records in core BTGC8 show trends similar to those observed in core BGGC5. Therefore, although core BTGC8 was inappropriate for diatom records, it helped to better understand the continental input. Therefore, both cores were used in the interpretations.

The geographical extent of our interpretations is based on the similarities found in the climatic trends found by Maldonado et al. (2006) in coastal areas of southern sites, close to our sampling area (31°S). In addition, we cited the works of Ortega et al. (2012, 2019) and others that suggest an increase in the wet conditions from the mid-Holocene (6000 BP) to recent eras.  Such work do not mention a weakened ENSO during 6000-4000 BP reported for south Peru, suggesting dry periods for the SE Pacific. This was probably caused by the intensification of fog in the Coquimbo Region during La Niña events (~30-31°S), which sustain relevant vegetation along the coast. Otherwise and at an inter-annual scale, the increasing variability of El Niño in the last 100-200 years is also observed in our cores, evidenced in increased grain size, high magnetic susceptibility, and higher Fe and Pb concentrations that point to growing continental runoff. This is also observed at a millennial scale; wet conditions can be deduced by the regular rise of such metals, and higher K/Ca ratios and pollen moisture index, all of which help to establish more humid conditions towards the present.

Concomitantly, oxygen conditions at bottoms changed dramatically in the past 2000 years, most probably due to an increase in ENSO variability, reducing the suboxic conditions of bottoms. This hypothesis is based on the observed changes in the benthic communities during El Niño events during the past 20 years, mostly reported in northern and central Chile, and Peru. The OMZ extension and its impact on the bays in northern Chile is still a matter of study. Some observations are reported by Gallardo et al. (2017), but they fail to explain the main variability of the presence of oxygen deficient water over the shelf. Observations suggest that low oxygen conditions were preceded by a period of southerly winds which were favorable to coastal upwelling and the rise of subsurface waters. In a longer time scale, we assume that the winds are still the main drivers for the intrusion of oxygen deficient waters into the bays.

*- The records of productivity and oxygenation are expected be related as it is observed in sediment cores from Peru (Gutierrez et al., 2009; Salvatecci et al., 2014). However, the opal flux curve shows a sharp decrease at 6500 BP that is not reproduced in U, Re or Mo records. In the recent period (last ~200years), oxygenation seems to increase as well as productivity, which seems contradictory.*

Low oxygen conditions are not only related to the onsite regeneration of organic matter. Such low oxygen condition at the bottom occurs during the intrusion of oxygen depleted waters belonging to OMZ into our margin, mostly associated with ESSW. Therefore, the variability in primary productivity and their remineralization has a seasonality effect on local oxygen bottom water that has not been reported at a millennial time scale. In this case, our records are showing the variability of this oceanographic feature, revealing two relevant changes at cal BP 6700 and 2000.

***The authors should discuss the relationship between oxygen and productivity proxies in their dataset and discuss why productivity and oxygenation which are both related to the upwelling intensity show here different behaviors.***
***- The discussion about the upwelling intensity is only based on the productivity proxy while oxygenation proxies are left aside.***

Upwelling intensity is more related to wind stress, while oxygen content depends on onsite consumption and the presence of water belonging to the OMZ; the first one operates at a local scale while the second at regional scale. Therefore, better indicators of upwelling are diatoms and opal; we later analyze both parameters in terms of their records in sediment cores. We attempted to discuss all the parameters from their geochemical point of view to better explain how the interpretations of the proxies can help define environmental conditions (points 5.1, 5.2, 5.3) and concluded with an overall picture of the environmental conditions in the past 8000 years in our region, using all proxies. We based our assumptions in the previous descriptions of environmental variability for the Holocene in the region, based on marine and coastal lagoon sedimentary records (see references cited).

***- The logic of the discussion does not respect the time line of events in the literature nor in their own records. A supposed increase in ENSO is mentioned to explain both the drop in productivity at 6500BP and the increased oxygenation at 2000 BP. And the increasing humidity through the Holocene. Finally, ENSO is invoked to explain everything without citing the articles which properly reconstruct ENSO using monthly resolved marine records.***

We rewrote the discussion, more specifically point 5.4 (climatic interpretations). Here we included new references, and we focused our discussion on local studies with a time scale relatively similar to ours, based on the studies by Maldonado and Villagran 2006, Maldonado and Rozas 2008; Jenny et al. 2003; and Ortega 2012; 2019, among others. The environmental variability in part of the Holocene was based on the analyses carried out by Lamy et al., 2014; Kaiser et al., 2008; Hebbeln et al., 2002, who interpret the climatic variability in the region according to the movement of SW winds, which is consistent with observations of current seasonal variability as indicated by Montecinos and Aceituno, 2003 and Quintana and Aceituno, 2012, among other prior works that I have not cited. However, the latter summarize very well the current climatology of the area. These studies suggest that the main modeler for climate variability is El Niño-La Niña (ENSO). This inter-annual (El Niño) and inter-decadal (ENSO) changes shape the area's climate. We have tried to explain the variations of our sedimentary records based on this climatic variability, highlighting aspects such as changes in the oxygenation of the bottoms and changes in primary productivity.

*Detailed comments:*
*Abstract:*
*L37: ENSO corresponds to inter-annual variability. Interdecadal variability would be related to the PDO.*
*L49: a "period" does not have a maximum. I think I understand what the authors mean but it needs to be rephrased.*
*L51: I suppose "moisture levels" refers to the climate on land, but it should be clarified.*
*L52: what is being stronger? Please clarify*
*L53: "associated with greater El Niño frequencies". See comments about the discussion.*
*What about the 2 other high productivity periods?*
*L124-126: please mention the time period recorded by the sediment cores.*
*L245: Add " 14C" before " age value".*
*Introduction:*
*Half of the introduction is dedicated to ENSO but the bibliography about paleo-ENSO is incomplete and does not reflect correctly the current knowledge.*
*L99 – L108: "Paleo-ENSO reconstructions indicate attenuated ENSO events before the mid-Holocene (last 5000 years) and increasing from this period towards the present (Marchant et al., 1999; Koutavas et al., 2006; Vargas et al., 2006), consistent with paleoceanographic and paleoclimate interpretations (Rodbell et al., 1999; Rein et 102 al., 2005)."*
*This part incorrectly reflects the literature. The onset of ENSO 5000 years ago is an old hypothesis that had support from low resolution continental rainfall-derived indicators, but that has been contradicted by high resolution marine proxies adapted to ENSO reconstruction. Marchant et al., 1999, show an increased variability at 3ka, not 5ka. Koutavas et al., 2012 showed a minimum in ENSO activity from 6-3 ka. This was confirmed by mollusk shells (Carré et al., 2014), corals across the Pacific (Emile-Geay et al., 2016) and sediments off Peru (Rein, 2007).*

***The Lake Pallcacocha record (Rodbell et al., 1999; Moy et al., 2002) has been shown to reflect glacier activity related to tempêrature and the south American monsoon (hence the precession signal) instead of ENSO (Rodbell et al., 2008).***

We took into account all the detailed comments and changes were made in the text in agreement with such comments.

We changed the introduction and added more information about climatic patterns at SEP and their evolution over time, as well as interpretations by several authors. These references indicate that ENSO is the main oscillation that affects climatic variability while others suggest probable causes for intense and weak El Niño events, including an orbital forcing that impacts the thermal gradient by means of changes in the insolation. All authors agree on the changes that take place at different times scales due to the ITCZ displacement (contraction/expansion), as well as on the changes in the intensity of the winds that impact upwelling strength by atmospheric variability (PDO, SAM). The area's main oceanographic features are impacted by atmospheric variables that operate at different time scales. Some authors conclude that the intense upwelling and high productivity during the early Holocene (>7000 cl BP) were similar to those that take place during La Niña conditions, which is consistent with our findings. Different results for subsequently periods were found, however. After 7000 cal BP, we observed an increase in humid conditions, which points to high ENSO variability. We were unable to identify dry periods between 6000 and 4000 cal. BP, nor weak events at about 3000 cal BP as proposed by several authors. However, there is general agreement in an overall ENSO variability increase towards the present. Our results are mostly in agreement with these interpretations and we show a major oxygenation of the bottoms in the past 2000 years –which was not previously reported. We are establishing a link between such oxygenation and an intensification or high frequency of equatorial waves that are introducing oxygen into central Chile's margin, as observed today. This has important implications for the development of the OMZ and its function as a source of nutrients for the photic zone.

Our motivation was to establish the main environmental conditions during the time scale supported by our cores. The interpretations were based mainly on geochemical proxies observed in sedimentary records. The changes observed were related to the general climatic patterns reported for the SEP, which have implications for the area's oceanographic features. We focused the discussion on the environmental changes due to climatic and atmospheric fluctuations based on scenarios described for the area. We have described how the main oceanographic features have changed in the past 8000 years suggesting the environmental characteristics that must prevail for the observed oceanographic conditions to develop. Such findings are supported by pollen records with which we were able to identify the area's climatic variability.

We believe that our findings deserve to be published. We corrected the main errors found by the referees concerning dates and periods established in the literature for ENSO variability and other minor grammar-related indications. We improve figures 2 (we add st14), 5, 6 and 9 according referees' suggestion.

**5.3. Anoxia is mentioned when metal concentrations show more reducing conditions. Anoxia corresponds to the complete absence of oxygen. Since the authors cannot reconstruct oxygen levels quantitatively, they should not use the term "anoxia" and describe changes in terms of more or less oxygenation.**

Re and Mo enrichment can identify anoxia (sulfate reducing conditions) during sedimentation. Therefore, an explanation was added to help visualize this concept. A specific value of oxygen cannot be determined, but when Re and U exhibit an enrichment they establish sub-oxic conditions (>0.2 – 2 mL/L). When Mo enrichment is high, sulfidic conditions are possible.

**5.4. In the first version, the pollen record was barely mentioned. In the revised version, it represents the largest part of the climate discussion.**

In this regard, in the second version we tried to elaborate on the climatic interpretations and establish their relationship with oceanographic proxies. We think this was relevant to understand the global scenario and how climate conditions favor or disfavor upwelling and primary productivity, as well as the impact on the OMZ. Therefore, this chapter was re-written in this version, and we added other comments to provide an overview of the significance of our results.

 **L824: How an increase in ENSO would explain lower productivity without producing higher oxygenation? In addition, ENSO has been shown to be decreasing at 6700 BP, to enter its minimum activity between 6000 and 3000 BP (Koutavas et al., 2012; Carré et al., 2014).**

As explained before, this version includes more accurate explanations about our findings, but our data shows no evidence about a reduced variability between cal BP 6000-4000, as accurately reported by Koutavas and Carré. However, these studies were carried out in Peru and most probably, this ENSO variability has different impacts on the Chileanmargin. Our data matches other studies in the area.

**Second Review of Muñoz et al., bg-2018-396**

*Summary*
*The revised version of the Manuscript "Reconstructing past variations in environmental conditions and paleoproductivity over the last ~8000 years off Central Chile (30° S) presents an improvement introducing and discussing their data in context of ocean-atmosphere interactions and more importantly comparison with and reference to previous data from the region. However, I still have two major points of criticism, (1) the introduction is still not well written enough and is missing information about proxies being applied to support previous observations and conclusions and more essentially the motivation of the Authors to select the study sites and the proxies ultimately utilized, (2) Although ta paleodiscussion was now added, the discussion is still mainly discusses each result point by point, appears unfocused and needs re-structuring. Overall there is still a main question or motivation missing throughout the manuscript, there need to be some sentences added why the authors selected the study are and what they hope to improve in the paleoceanographic knowledge about the SE Pacific. I am not convinced by the paragraph (line 120-126) where the Authors introduce their work, there is little connection to what was written in the introduction before. The Authors improved the introduction by adding more detailed information about the ocean-atmosphere dynamics relevant in the study area. Unfortunately, references to previous work is still too vague, for example just referring to "sedimentary records of several proxies". I think the authors deleted important information on how changes in the ocean-atmosphere dynamics are reflected within sediment records from the previous version. And thus, an introduction about what proxies are feasible to use for the authors research question is basically completely missing. Following on that, there is no information provided on what the others selection of proxies applied was based. Suggestions from my side how to improve the structure of the introduction can be found under the line-to line comments.*
*Furthermore, the structure is still a bit strange with specific information about the area, then explaining general observations from the SE Pacific This should be reversed, going from the big picture to the study area.*
*The Discussion of the new data presented by the Authors based on climatic changes and the comparison with previous studies significantly improved in section 5.4 climatic interpretations. However, I think the structure of the discussion needs still improvement. At the moment the different proxies are discussed successively, but this structure results sometimes in non-chronological description of the significant periods highlighted in the manuscript. I suggest to re-structure the discussion in first the modern conditions and afterwards the 3 time intervals (> 6 kyrs, 2.1 to 4.6 kyrs and recent to 260 yrs BP) and finish with section 5.4 climatic interpretations presented in the current manuscript. As the definition of these time intervals is also one of the mayor findings of the study, this structure would improve their significance to the reader.*

The introduction was rewritten; we included the missing information about the proxies that was in the first version. This certainly helps to understand why we used them as indicators of the main oceanographic parameters. The relevance of the main oceanographic features of the study area was highlighted. The focus of our work was to establish the main oceanographic changes observed in the past 8000 years related to the main climatic variability reported for the zone during this time period. Based on the characteristic of the study area and its relevance in the northern Chilean margin, our records reveal the past variability of the main oceanographic and climatic conditions in the region. We changed the structure of the introduction starting with the main climatic drivers in the area and continuing with the main oceanographic features and detailed observations of several aspects in the study area.

The discussion was also rewritten, specifically point 5.4. The remaining sections were partly modified in an attempt to establish the validness of the proxies used and establish better relationships with the information provided by the indicators. First, we established the relevance of terrigenous inputs in the area, following with the variability of the organic and oxygenation records and concluding with climatic interpretations based on the changes observed in sedimentary records.

We decided not to follow the suggested structure of separating the discussion in time periods because the data shows a continuous increment in wet conditions over time. Only oxygen shows major changes in specific periods of times, and productivity follows a similar behavior, except at around ~1700-4500 cal BP which does not, however, point to inconsistencies. The slight increase in productivity during this period took place during suboxic conditions which remained until 2000 cal BP, as explained in the text. In addition, we further explained that oxygen conditions depend mainly on water circulation. Productivity has a lower impact on oxygen consumption, even today, increasing the relevance of the intrusion of water with low oxygen content. On this regard, we see no inconsistencies with regional interpretations as was suggested by the referees according to the studies performed in the Peruvian margin.

*The text needs still a lot of improvement. Paragraphs are often not properly connected to guide the reader and several grammatical errors are distributed throughout the whole text. Furthermore, the use of "decrease" and "increase" is often inappropriate, as there are no values given for comparison, for example the authors conclude in line 858 that nutrient-type elements are reduced at present and higher at cal BP 6500. On the whole manuscript is too long and especially methods descriptions are too detailed. When applying commonly used methods it is sufficient too shortly describe the procedure and refer to the original publication. Detailed explanations are only needed if analysis vary from normal procedures. I suggested some superfluous information under the detailed comments to shorten the manuscript.*
*The figures were all improved following previous reviewer's suggestions, however Figure 9 presenting the Pollen record is still the only figure were age is given on the x-axis instead of the y-axis. The Authors didn't give a reason for not changing this, it would help comparing the data.*

The new manuscript version is shorter. We reduced the methods and result sections by half, and we focused our attention on the discussion. Grammar was polished and we modified Fig 9 according to the reviewer's suggestion. Finally, we modified all lines that had detailed comments by the reviewer.

*Line by line comments:*
*Line 34: change "in" to "at" and I don't the commas are needed here.*
*Line 46-47: rephrase "The first period was conspicuously high..." it is not the period that is high but the productivity during this period, change to something like "The productivity during the first period was conspicuously high..."*
*Line 49: rephrase "this period reached a maximum at ..." what maximum was reached, needs to be spelled out*
*Line 52: again rephrase ", being remarkably stronger in the last 2000 years" are you referring to oxygen levels?*
*Line 64: change "are developed" to "develops"*
*Line 69: rephrase, second sentence in a row starting with "this high productivity..."*
*Line 71: Is "where their intensity, ..." supposed to refer to the OMZ? Then please use the singular i.e. "where it's ... "*
*Line 91: change to "have also been linked"*
*Line 92: change to "influences"*
*Line 93: change "this latter an important forcing" to "which acts as an important forcing"*
*Line 94 onwards: this connection is confusing, you refer to "this variability" producing humid and arid conditions, you seem to refer to changes in the processes (i.e. sea ice extent, Hadley cell and latitudinal position of the ITCZ. However, you only mention fluctuations in upwelling in the previous sentence. I suggest changing the beginning of the sentence into "Changes/Variability in the austral insolation and the related processes/mechanisms produce..."*
*Line 96: change "on top of all this" into "An additional important driver...".*

*Line 120 to 126: This is a summary of what you did, I rather expect here a paragraph about why you selected your core positions on the basis of the introduction you give*

*Line 176 to 179: delete, these are common procedures and you are not referring to this information in the following.*

*Line 184 to 211: remove text. All this information is repeated in the following sections, or if not can be added to the appropriate section for each proxy.*

*Line 241: remove "ages were calculated using 5568 (yrs) as the half-life of radiocarbon" superfluous.*

*Line 260: I think you can shorten 3.4 Trace Metal analysis. I am not familiar with the method myself but the descriptions appear extremely detailed and could be shorten, as it is commonly applied.*

*Line 293: section 3.5 TOC and stable isotopes can be significantly shortened, it is sufficient to explain how much material was weighed into tin and silver capsules respectively and where in what machine analysis have been performed, and of course the external reproducibility of measurements must be given. It is not needed to explain the entire procedure of the measurements. In short rephrase the sentence in line 296 to 298 and remove the text until line 306.*

*Line 482: add "are" behind Trace metals*

*Line 511: "This is the result of the wind" please elaborate a bit more in what way this is caused by the winds*

*Line 532 and 533: change "more negative" to "lighter"*

*Line 539 and thereafter: change NO3 to NO3- or use "nitrate"*

*Line 630: not sure what the Authors are referring to as "it"*

*Line 716: remove "notorious"*

*Line 777: change "trend" to "trends"*

*Line 840: Change "Our result indicates" to "our results indicate"*

*Line 843: change to "we interpret this difference as..."*

*Line. 848: add how these redox conditions have been reconstructed.*

*Line 856: change "where" to "were"*

*Line 857: get rid of "the presence of"*

[revised manuscript text omitted]

**Comentario [P75]:** Figure was modified according referees's suggestion

---

## Author Response (AR3)

Dear Editor;

We have carefully reviewed the previous comments to the second revision to find out which arguments had not been dealt with. We addressed all of them except for the one related to making changes to the organization of the discussion. In this sense, the resulting manuscript was very redundant and there were extensive explanations that in no way helped to improve it. Therefore, this version has been restructured according to the previous suggestions. We arranged the discussion by periods and all the sections were shortened. The manuscript now contains 22 pages, without references, tables and figures.

The referee indicated that the discussion jumps from discussing terrigenous inputs to primary productivity in the next section, without explaining why these are important points in this area. We tried to improve these issues in the discussion, but in the understanding that while the relationship between these aspects is a known process, there is a need to explain the sedimentary records. We also made a sound description of the study area, highlighting the relevance of the continental inputs, the effects of the El Niño events, and the productivity of the area. In addition, the introduction describes the main climatic and oceanographic characteristics of the area and how they could influence sedimentary records, to finalize with the objectives. Thus, we really focused our efforts on the discussion trying to avoid large explanations based on text book knowledge.

Is important to mention that we found that our records make a quite good description of the climatic trends previously illustrated for the Holocene in the region. The climatic interpretations provide a point of view that helps understand the prevalence of the environmental conditions in the last 8000 years. These clearly point to a change to more frequent El Niño events that will have an impact not only on climate and primary productivity, but also on bottom oxygenations.

We look forward to a positive assessment by the reviewers and for the manuscript to be accepted for publication.

Sincerely yours,

Práxedes Muñoz

On behalf of all authors

Referee answer

*I think the Authors improved the content of the manuscript in comparison to the previous version. Unfortunately, I am not satisfied with the changes performed, especially as the authors avoided to apply a lot of suggestions/did not apply them thoroughly or even reported to have incorporated them but actually haven't.*

Answer: I don't fully agree with this comment. All changes suggested were incorporated. This can be corroborated by comparing the original manuscript with the one with track changes, but the reorganization of the discussion in a timeline was not properly done. Otherwise, we added new references; the introduction was changed completely; we reduced the extension of the methods and added several comments that extended the discussion and the length of the manuscript. This resulted in redundant explanations of our main results which could be explained by the same environmental processes. It was too ambitious to provide a deep analysis of each proxy and still maintain the focus of the study.

*Although the discussion gained important content the authors did not clean the text of textbook knowledge, more precisely it is not advisable to explain all proxies you used, or compare your data to, in detail within the discussion. This should either be moved into the introductions/methods or be removed completely. Furthermore, both reviewers found the discussion confusing and unstructured, just adding more text does not help the readability of the text, the discussion must be I shortened and restructured in some way. If the authors do not follow the suggestion of the reviewers they could have changed the structure differently. The content of paragraphs and sections are still not connected, the discussion jumps from discussing terrigenous inputs to primary productivity in the next section without explaining why these are important points in this area or have any train of thought.*

Answer: In this version we reorganized the discussion, highlighting three main periods when we observed important oceanographic changes based on productivity proxies and redox sensitive elements to interpret the main prevailing oceanographic conditions. Later, in section 5.3, we explain these variations according to climatic interpretations based mainly on pollen records that are consistent with the information reported for this region. In addition, we reduced the abstract, the methodology, the results and the discussion. To reduce the length, we avoided detailed explanations of each proxy focusing the discussion on the main interpretations of the proxies. Some paragraphs were reduced and moved to the introduction and the study site sections; others were totally removed. Some points about discrepancies between cores and proxies were discussed in previous version but now they were explained in more detail, for example in lines 485-492, 572-578, 681-692.

*I do not advise publication of this manuscript before it was not shortened to less than 25 pages, 20 would be better and should be sufficient to discuss the main points of the presented data.*

Answer: The paper was reduced from 30 pages (970 lines) to almost 22 pages (728 lines), without references and figures.

[revised manuscript text omitted]

**Comentario [P8]:** We combine the discussion of proxies for oxygenation and productivity. This paragraph start with a short explanation of the profiles.

**Comentario [P9]:** We analyze the proxies (organics and inorganics) according time (from the past to the present), considering the main periods observed.

nutrient-type elements (Ni, Sr, Ca, Cd) into the present is consistent with the rise in
oxygen in bottoms. Hence, the slight rise of Ba from cal BP 4000 to the present (Fig.
8a) is a response to this less anoxic environment leading to negative correlation with
TOC (-0.59; Table 5) due to Ba remobilization in anoxic conditions before
cal BP 6500. On the other hand, P distribution showed a trend similar to that of TOC
and other elements related to organic fluxes into the bottom (Ni, Cd), although with a
lower correlation (~0.6). This is consistent with the distributions observed for U, Re,
and Mo at core BGGC5, which indicate that anoxic or suboxic conditions were
developed from cal BP 8200 to ~ cal BP 1700, but were stronger before cal BP 6500
(Fig. 8a, 8b). After this period and into the present, a remarkable reduction in their
concentration suggests a more oxygenated bottom environment, concurrent with lower
organic fluxes to the sediments. The Re profile shows the influence of suboxic waters
not necessarily associated with higher organic matter fluxes to the bottom. Since this
element is not scavenged by organic particles, its variability is directly related to
oxygen changes (Calvert and Pedersen, 2007, and references therein). Additionally, it
is strongly enriched above crustal abundance under suboxic conditions (Colodner et
al., 1993; Crusius et al 1996), being >10 times at core BGGC5 (Table 4) before cal BP
1700. In the same manner, U shows a similar pattern and while organic deposition has
an impact on its distribution (Zheng et al., 2002), it is also related to changes in bottom
oxygen conditions.

Otherwise, the accumulation of P depends on the deposition rate of organic P (dead
plankton, bones and fish scales) on the bottom, and is actively remineralized during
aerobic or anaerobic bacterial activity. P and TOC showed a declining trend towards
the present, suggesting reducing flux of organic matter over time, which was also
observed for Ni and Cd distributions. Alternatively, reducing fluxes of organic proxies
could be explained by the higher remineralization of organic material settled on the
bottom due to higher oxygen availability, as shown by U, Mo and Re distributions
(Figs. 8a, 8b). To better approach this issue, establishing the variability of primary
productivity over time and the environmental factors that facilitate its development is
required.

Productivity reconstructions were based on qualitative diatom and sponge spicules
relative abundances, quantitative diatom counts (valves g$^{-1}$), and biogenic opal content
only in core BGGC5, since core BTGC8 registered low valve counts (< 1 % in relative
diatom abundance). However, in both cores diatom assemblages were represented mainly by *Ch.* resting spores, which are used as upwelling indicators (Abrantes 1988,

Vargas et al., 2004). The downcore siliceous productivity based on opal distribution (Fig. 6) distinguished three main time intervals of higher productivity, which coincide with the ages highlighted by the distribution of the sedimentary proxies seen previously: (1) > cal BP 6500, (2) cal BP 1700 – cal BP 4500 and (3) recent times (CE

2015) – cal BP ~130. The opal accumulation rate in the first interval was remarkably high, amounting to ~27 ± 13 g m$^2$ yr$^{-1}$ (range: 9 – 53 g m$^{-2}$yr$^{-1}$, Table 4), when

*Chaetoceros* spores were predominant, indicating an upwelling intensification. During the first period, all metal proxies showed primary productivity increases before cal BP

6500, as indicated by opal accumulation within the sediments. Here, Cd and U

accumulations in the sediments resulted in high Cd/U ratios, even at core BTGC8 (> 2;

Fig. 6), pointing to very low oxygen conditions (Cd/U ratios could vary between 0.2

and 2 from suboxic to anoxic environment; Nameroff et al., 2002). In addition, during this period the presence of sulfidic conditions is suggested by Cd and Ni enrichments (>140 and 3, respectively, Table 4), since its buildup within the sediments is highly controlled by sulfide concentrations (Chaillou et al., 2002; Nameroff et al., 2002;

Sundby et al., 2004), though Mo and Re where not especially high during this period (~17 and ~19, respectively, Table 4), all of which is suggesting an intensification of the upwelling during this time interval.

In the second interval, opal accumulation decreased to ~ 11 ± 4 g m$^2$ yr$^{-1}$ (range: 2 – 21

g m$^{-2}$yr$^{-1}$, peaking at cal BP 3500–4000; Table 4, Fig. 6a), which is partially consistent with nutrient–type element distributions (Fig. 8a). Fe clearly shows higher values around cal BP 3500 (Fig. 8a), helping to boost primary productivity at this time, with a small diatom increase, measured as valves per gram and abundance (%) (Fig. 6a).

Other elements showed less prominent accumulations (Ni, Cd, Ba, Ca and P), pointing to lower organic matter deposition into the sediments during this period (Fig. 8a).

However, low oxygen conditions within the sediments are maintained, which could be more related to the manifestation of the oxygen minimum zone close to the coast, favoring Mo and Re accumulation until cal BP 1700 (Fig. 8a). Lower Cd/U ratios (~ 1;

Fig. 6) were estimated, suggesting higher variations in primary productivity but with moderate changes in oxygen conditions at the bottoms. After cal BP 1700, there is an evident and remarkable reduction in organic fluxes to the sediments and a drastic change to a less reduced environment towards the present, suggesting a more oxygenated bottom environment concurrent with a reduction in primary productivity,

**Comentario [P10]:** We define the main productivity periods that are related with the other proxies distribution mentioned before.

**Comentario [P11]:** We highlighted the main observations in the first period.

**Comentario [P12]:** We continued with the observations done in the second period

**Comentario [P13]:** And observations in the third period, including the last ~ 130 years 
[revised manuscript text omitted]

---

## Author Response (AR4)

Dear Editor;

We have carefully reviewed the comments of the referees from the last revision and we have addressed all of them. As suggested by the reviewers we have complemented the discussion with new references in order to put the data in a regional context. We have improved the age model, and re-organized and re-wrote several paragraphs.

We reduced the introduction and descriptions of the study area and we included a paragraph to discuss Cu concentrations in recent times, according to the instructions of Referee # 1. In addition, we have improved the introduction and discussion section to respond to the comments of both reviewers. Misspelling errors have been also thoroughly checked.

It is important to mention that our records describe very well the observed climatic trends for the region in the Holocene, and our interpretations help to understand the effect of climatic variability on the oxygenation and productivity in the last 8000 years.

We hope that the reviewers positively evaluate the several improvements incorporated in this revised version of the manuscript.

Sincerely yours,

Práxedes Muñoz

On behalf of all authors

**Referee answer**

**Referee#1**

The paper has been greatly improved. I recommend to publish it with the following minor revisions:

- The abstract should be corrected for english. Several inacurate choices of words.

Answer. The revised version of the manuscript has been checked by the Elsevier Language Editing service.

- the introduction is still too long. I suggest deleting all the unfocused generalities from line 73 to line 111.

Answer. The introduction was modified and reduced (was cut in 13 lines). We re-wrote several paragraphs avoiding generalities but considering all the necessary aspects to establish the main objective of the project that was to identify the past variability in the main climatic forcing and oceanographic conditions that affected the area in the last 8000 years through the study of sedimentary records.

- the description of the study area is also long and includes unnecessary generalities. Delete from line 182 to line 190, at least.

Answer. The study area was reduced, 10 lines were eliminated removing repeated ideas that were mentioned in previous paragraphs in the text.

- There is a confusion with the calculated reservoir age: it is presented as a local deviation (DR) in the method (line 241), and as a reservoir age (R) in the results (line 318-322). The values in Carré et al., 2016 and Merino-Campos et al., 2018 are DR values. Check what the age of 411 years really corresponds to (DR or R) and correct the text accordingly (and the age model if necessary).

Answer. The text was corrected, the explanations for calculations were revised and also we changed the modeling method considering the hiatus observed by referee 2.

- line 585-586: I am surprised that the author do not relate the large Cu increase (and to a lesser extent Fe and Mn) in the last century to mining activities. It is said that it indicates an increase of productivity but this is not consistent with other productivity indicators, nor, as they mention, with the increase in oxygenation.

Answer. As suggested the increase of Cu in the last century is now discussed (lines 632 -646).

- the discussion about climatic implications (section 5.3.) is mostly focused on ENSO, which is somewhat speculative since the record does not have an interannual resolution, while in the other hand changes in the mean state, which is indeed what sediments most clearly record on the first order, is not sufficiently discussed. The most robust result here is the long-term trend from dry, disoxigenated, productive conditions in the early Holocene, towards more humid, more oxigenated and less productive conditions at present. This is a strong support for the scenario proposing that early-midHolocene was characterizaed by a La Nina like mean conditions in response to insolation.

Answer. The discussion is separated in three main sections, first we discussed the main sedimentary composition, and then the temporal variability of the main productivity and oxygenation records, following with the main climatic implications. The PDO and ENSO are the main climatic drivers at centennial and decadal scale at eastern Pacific, we base our discussion in the variability imposed by these forcings, comparing with other studies in the area. We separate the records in periods according to their distribution and we identify similarities with other reports in the region, which based their observations on the variability of these processes. It is recognized that ENSO related variability has played a main role in the climatic expressions, but phases of PDO have also to be taken into account. The movement of the ITCZ depend on both, from which the humid and dry conditions have been predicted, all of them have been reported as the key drivers for the climatic variability from the mid-Holocene until today. We discussed our findings based on this information and highlight the variability of the OMZ effect over the shelf, which is an important characteristic at the East Pacific margin and has a significant impact on main biogeochemical cycles.

The La Niña-like conditions is mentioned in the discussion, but strangely, the authors cite De Pol-Holz et al., 2006 and Kaiser et al., 2008, who found warmer and more oxigenated conditions between 10 and 5ka further offshore in that area, in contradiction with the results presented here. The authors should discuss this discrepancy and present their results in the context of the early to mid Holocene La Niña-like hypothesis which was shown by Koutavas et al., Science, 2002, and by Carré et al., Quat. Int., 2012. This latter paper already discussed some coastal vs offshore discrepancies observed in Chile in this period.

Answer. De-Pol Holz indicated a decreasing trend of the denitrification rate from the early to mid-Holocene, but is not comparable with estimations at previous periods (> 15 kyr BP). The reduced ventilation during the Holocene was attributed to a decreased wind forcing and latitudinal SST gradient, but the higher denitrification rates indicated a more intense OMZ, compared to the estimations in the LGM. However, our sedimentary cores were taken at shallower depths, close to the coast, recording the direct effect of the upwelled waters and the surface variability of the oxygen minimum zone. Besides, the works of the cited authors do not refer directly to increasing oxygen concentrations, but to the OMZ response to the glacier and Patagonian dynamics. In the case of Koutavas et al., they estimated the surface temperatures deduced from the oxygen isotopic composition of foraminifers; they showed that mid-Holocene was the beginning of higher climatic variability. Our records match well with several descriptions for this period (~6500-8000) indicated for several authors, included De-Pol Holz and Koutavas, indicating that this period is the beginning of higher climatic variability towards present.

Several authors indicate clearly the predominance of La Niña at mid-Holocene, and in the case of the SST, the necessary conditions to develop La Niña condition is a higher zonal and meridional SST gradient. At tropics the surface temperature is higher around 21-22°C, in comparison with temperatures at southern sites (18°) in this period. However is relevant to understand that our cores are under the effect of the main upwelling center and the in-shore upwelling waters are cooler (~13°C) than of shore, as mentioned by Carré et al., 2012. Therefore is not a contradiction to establish northward displacement of the ITCZ at mid-Holocene, which has been discussed by Koutavas et al., 2006 and others.

- Line 616-617: the La Niña-like conditions may be associated to reduced ENSO and a shifted ITCZ, but we cannot say it is caused by them.

Answer. This line was corrected

**Referee#2**

**The manuscript by Muñoz et al. presents a multiproxy approach analysis of two short sediment cores retrieved off central Chile to reconstruct paleoceanographic and paleoclimatic variability during the last 8000 years. While several paleoceanographic studies have been conducted in southern Chile and off Peru, paleoceanographic records from central Chile covering the Holocene are extremely scarce. Thus, this study presents valuable data to understand millennial-scale changes in the Eastern South Pacific that can potentially attract the paleoceanographic community. I agree with most of the comments and observations made by the two anonymous reviewers on the first manuscript. After reading the revised version of the manuscript I think that the manuscript still needs substantial work before it is considered for publication in Biogeosciences.**

**My main criticism, also mentioned by both anonymous reviewers, is that the manuscript lacks a proper paleoceanographic interpretation combining relevant paleorecords from the Peruvian Upwelling system and from southern Chile. This could help to disentangle high latitude versus tropical forcing driving the observed changes in the data. During the last decade multiple studies focused on the Peruvian upwelling system were published that could help to gain insights on the mechanisms driving millennial-scale changes off Central Chile. Surprisingly these studies were not included in the revised version of the manuscript.**

**Answer.** We have now complemented the discussion with new references in order to give a major scope of the data in a regional context, considering some works developed in south-central Chile and Peru. However this is limited to the mid and late-Holocene, and some information for the most recent time (two last centuries). Our cores record only ~8000 years and the sedimentation rates were relatively low, avoiding to obtain a higher resolution in recent times. Notwithstanding, our records establish relevant episodes of environmental changes that have been also documented at southern and far northern areas of our study site. In addition, this record is also in agreement with climatic descriptions for the south-central Chile and northern regions up to Peru. We consider that this version now includes a thorough overview of climate variability in the region, as requested by the reviewer, for the time interval recorded in our cores.

**The second main problem that I see with the manuscript is that the Discussion lacks focus, probably because of the lack of a clear scientific question and the multiple types of proxies presented by Muñoz et al. I suggest including a clear scientific question in the Introduction to better guide the reader throughout the manuscript.**

**Answer.** The aim of this project was to establish past variability of in the main oceanographic and climatic features in the study area a semiarid zone in the north-central Chile. The climatic fluctuations produce humid and arid conditions along the SE Pacific where the intensity of the wind remains the key factor for the strength of the upwelling and, therefore, for the supply of nutrients to the photic zone, being required for the development of the primary productivity. In our area, a main relevant upwelling zone has an impact over a narrow shelf and coastal zone along the year. In the same context, the oxygen conditions at bottoms and within the sediments are highly dependent on the oxycline, that normally is related with the intensity of upwelling and the oxygen consumption during the organic matter diagenesis, and the position of the oxygen minimum zone (OMZ). These characteristics have been described in the Introduction and Study Area sections, indicating the work done to establish the relationship between proxies measured in the sedimentary records with the main climatic variability described for the region. Our data indicates the greater relevance of the OMZ in the shelf over the oxygen consumption derived from the degradation of primary productivity, and that the latter had a variability that was not consistent with the variation in oxygen. Therefore, the oxygenation of the bottoms has been related with the variability of the oxycline position, which is affected today during warm events (El Niño), leading to the biggest changes in the biogeochemical cycles by the deepening of the oxycline and lower productivity. This is consistent with the main climatic shifts reported for the region from the mid-Holocene. Our work has showed the relevance of climatic variability at larger scales, both temporal and spatial, in the main environmental conditions in our region.

**Some other issues that need to be addresses are:**

**1) I am completely surprised that coastal sediment cores of just ~110 cm long can contain records of ~8000 years, especially in areas associated with high marine productivity. After examining the age models and Fig. 5, it is evident that the sediment cores present large discontinuities or sections with extremely low sedimentation rates like during the mid-Holocene in core BGGC5. This observation merits an in-depth discussion in the manuscript given that Holocene records from the Peruvian Upwelling systems are well known for discontinuities and erosion events (e.g. Erdem et al. 2016). Specifically, what is the role of winnowing on the proxies presented in this manuscript?**

**Answer.** The study area is considered as moderated in productivity and in consequence, the rate of organic carbon is lower than for other upwelling areas (e.g., off Concepción 36°S), however the upwelling here is perennial and the sedimentation rate is comparable to other shelf areas in northern and southern Chile (Muñoz et al., 2004), thus with these low accumulation rates we can confidently say that our cores contain the material accumulated in the last ~8000 years. Our sediments did not showed evidence of slumps, but the very low sedimentation rate in two sections of the core should be interpreted as a discontinuity for erosion, as was suggested by the reviewer. We analyzed the cores taking into account these discontinuities and thus we used other age modeling.

**2) The authors show records of Mo, U and Re for both cores to infer changes in paleo-oxygenation. The use of Mo/U systematics is a powerful tool to reconstruct sub-oxic and anoxic**

**conditions (Algeo and Tribovillard, 2006; Scholz et al., 2011; Salvatteci et al. 2016;). I recommend the authors to use this strategy to infer paleo-oxygenation (Section 5.3).**

**Answer.** Figure 9 shows the U/Mo ratios, and we also include the Re/Mo. In addition, enrichment factors of several elements were also showed, indicating the relevance of authigenic processes related with the oxygen conditions at bottoms and the presence of sulfides. The Cd and Mo enrichment account for it. All were showed in the context of more or less reduced conditions at bottoms, productivity and climatic variability. We also compared our findings with the data reported for south central Chile and Peru at shallower zones over the shelf.

**3) The results of some paleoceanographic studies need to be compared and contrasted with the results presented by Muñoz et al. to gain a mechanistic understanding of the processes driving the temporal trends in the data. For example, there are very few records of Mo, U and Re in the Eastern South Pacific (e.g. Muratli et al., 2010; Scholz et al., 2014; Salvatteci et al., 2016). A comparison and discussion with these records seems mandatory to better understand the mechanisms driving oxygen changes off central Chile. Other relevant studies to be consulted are: Doering et al. (2016), Mollier-Vogel et al. (2018), Salvatteci et al. (2016); Salvatteci et al. (2019).**

**Answer.** The major variability observed for atmospheric conditions from the mid-Holocene onwards, reflected the main oceanographic features described for the eastern Pacific. We used pollen records to help us obtaining better approximations of climatic variability in the region, and our records fix very well with continental records in Laguna Aculeo and other data from Los Vilos, identifying dry/humid conditions in the same periods than our marine records, and in agreement to the millennial variability observed for all the region, from the north to central Chile. We used marine information than describes productivity variations and changes in the reduced conditions in the seafloor, the later related with the effect of the OMZ over the shelf. According with new references, some works have established that the ventilation of the OMZ occurs during the deglaciation, and during climatic shifts that operated off Peru at mid-Holocene, however, the variability of its position over shallower zones has not been established yet. We think that our work contributes to establish the variability of the oxycline over the shelf related with the deepening of the OMZ.

**4) The authors should carefully revise the manuscript and the references for mistakes. There are several typos in the text (e.g. the use of BC and BP; misspelling names in the references, etc).**

**Answer.** We checked all typing errors

[revised manuscript text omitted]

**Comentario [A43]:** This figure was modified according the new model used to reconstruct the geochronology

[Figure]

Figure 5. Characterization of sediment cores retrieved from (a) Guanaqueros Bay (BGGC5) and (b) Tongoy Bay (BTGC8), where the color (Munsell chart scale)

represents the depth, dry bulk density, mean grain size, granulometry (% sand, silt, and clay), statistical parameters (skewness, kurtosis), organic components (TOC, C/N ratio, stable isotopes $\delta^{15}$N and $\delta^{13}$C ) and chemical composition (K/Ca, Ca/Fe).

a)

[Figure]

b)

[Figure]

Figure 6. Diatom and sponge spicule relative abundances, total diatom counts (valves g$^-$

$^1$) and opal (%), opal accumulation (g m$^{-2}$ y$^{-1}$), and downcore variations in *Ch.* RS

percentages as proxies of upwelling intensity in the BGGC5 and BTGC8 cores (Guanaqueros and Tongoy Bay, respectively). The medium dashed line represents the average of *Ch. resting* spores for the respective core.

[Figure]

Figure 7. Pollen record in BGGC5 core.

[Figure]

Figure 8. Downcore trace element variations in: (a) Guanaqueros Bay (BGGC5) and (b)

Tongoy Bay (BTGC8), off Coquimbo (30 °S).

a)

[Figure]

b)

[Figure]

Figure 9. Opal accumulation and authigenic enrichment factor (EF) of trace elements calculated for Guanaqueros Bay (BGGC5 core). Lithogenic background was estimated from the surface sediments of Pachingo wetland cores (see text). Pollen moisture index defined as the normalized ratio between Euphorbiaceae (wet coastal shrub land) and

Chenopodiaceae (arid scrubland). Positive (negative) values for this index indicate the relative expansion (reduction) of coastal vegetation under wetter (drier) conditions. Pb and Al distribution at BGGC5 core, representatives of terrigenous input to the bay.

**Comentario [A45]:** This figure was modified.

---

## Author Response (AR5)

Dear Editor;

We have carefully reviewed the referee's comments and we addressed all of them. We think that we have pointed out all the issues questioned by the referee and offer an adequate discussion, brief, and kept the length of the discussion. All changes were highlighted in the text. The lines with changes and the new lines are indicated below.

We hope that this version is appropriate for publication in your prestigious journal.

Sincerely yours,
Práxedes Muñoz
On behalf of all authors

Referee suggestion:

*Referee: The paragraphs in the Discussion section are extremely long, I suggest dividing the long paragraphs into two or three to facilitate the readability of the manuscript.*

Answer: We separated the long paragraphs into shorter ones.

*Referee: Lines 410-412: Move "We use Al as a normalizing parameter for the enrichment/depletion of elements due to its conservative behavior. The elements are presented as metal/Al ratios" the Methods section*

Answer: We moved these lines to the Methods section. Lines 251-258.

*Referee: Line 438: add "in core BTGC8" after "were lower"*

Answer: We added this sentence in line 447.

Referee: Lines 440-446: Move the first 3 sentences to the Methods section.

Answer: These lines were moved to the Methods section. Lines 249-251

*Referee: Lines 652: Yes, it seems to be contradictory, but a similar trend (trend towards higher productivity associated with decreasing reducing conditions) was observed off Peru (Salvatteci et al., 2014; Cardich et al., 2019). These authors based their observations on d15N, redox sensitive metals and benthic foraminifera assemblages. I suggest citing these papers and discuss whether the mechanisms proposed by these authors can explain or not the observed pattern off Guanaqueros Bay (BGGC5) and Tongoy Bay (BTGC8).*

Answer: We considered the referee's suggestion and we added other references to discuss the main mechanisms proposed for this oxygenation at coastal areas. Lines 662-677

*Referee: Lines 693-696. The westerlies were located in a more poleward position during the mid-Holocene (Lamy et al. 2001). Mollier-Vogel et al. (2019) erroneously indicate, citing Lamy et al. (2001), that the during the mid-Holocene the Westerlies were situated in a more northward position (see Salvatteci et al., 2019, for further discussion on this topic). This needs*

*to be corrected in the manuscript and the original citation must be also included (Lamy et al., 2001).*

Answer: We modified the paragraph changing the reference indicated by the referee. Lines 724-734

*Referee: Lines 725-730. The authors need to briefly compare and contrast their results with the reconstructed shifts of the storm tracks of the westerlies (Lamy et al., 2001).*

Answer: We discussed the southern Westerlies shifts in several paragraph; we compared our records in a general view for all SE Pacific first, and then reviewed the records for north-central Chile, establishing the main environmental conditions reported for the region. Lines 735-746, lines 778-780.

*Referee: Lines 763-767. Mollier et al. (2019) show evidence for decreased denitrification during the Mid-Holocene based on multiple d15N records along the coast. They do not show evidence for changes in sediment redox conditions. Rephrase this sentence or provide a better reference.*

Answer: A decreased denitrification implies a change in the redox condition and an oxygenation of the bottoms.  Therefore, the reference is appropriate. We added a sentence for further clarity. Lines 811-812

[revised manuscript text omitted]

**Comentario [A7]:** A separate point was introduced on this line.
The following line was modified for the reading continuity.

The slight rise in Ba in the last ~115 years (Fig. 8a) is a response to a less anoxic
environment, owing to better preservation within the sediments in less anoxic
environments with moderate productivity (Torres et al., 1996; Dymon et al., 1992) as
is the case with our study site (Gross Primary Productivity = 0.35 to 2.9 g C m$^{-1}$d$^{-1}$;
Daneri et al., 2000). This leads to a negative correlation with TOC (−0.59; Table 4),
owing to the remobilization of Ba under anoxic conditions before cal BP 6600.
Meanwhile, the P distribution showed a trend similar to that of TOC and the other
elements related to the organic fluxes to the bottom (Ni, Cd), although with a lower
correlation (~0.6). This is consistent with the distributions observed for U, Re, and Mo
at core BGGC5, which indicate that anoxic or suboxic conditions were developed
from cal BP 7990 to 1800 but were stronger before cal BP 6600 (Figs. 8a, 8b). After
this period and to the present, a remarkable reduction in their concentration suggests a
more oxygenated bottom environment, concurrent with lower organic fluxes to the
sediments. The Re profile shows the influence of suboxic waters not necessarily
associated with higher organic matter fluxes to the bottom. Since this element is not
scavenged by organic particles, its variability is directly related to oxygen changes
(Calvert and Pedersen, 2007, and references therein).

[revised manuscript text omitted]

**Comentario [A12]:** We modify this paragraph, we are citing and discussing the references suggested by the referee and we add other papers to improve the discussion the oxygen variability in coastal zones, with a major impacts on redox-sensitive elements in the surface sediments.

**5.3. Main climatic implications**

[revised manuscript text omitted]

**Comentario [A17]:** A separated point was introduced here, and the next line was modified.

**Comentario [A18]:** We add some lines in order to discuss the main findings of several papers including Lamy et al 2001.

[revised manuscript text omitted]

---

## Author Response (AR6)

Dear Editor;

We have carefully reviewed the referee's comments and we addressed all of them. We think that we have pointed out all the issues questioned by the referee and offer an adequate discussion, brief, and kept the length of the discussion. All changes were highlighted in the text. The lines with changes and the new lines are indicated below.

We hope that this version is appropriate for publication in your prestigious journal.

Sincerely yours,
Práxedes Muñoz
On behalf of all authors

Referee suggestion:

*Referee: The paragraphs in the Discussion section are extremely long, I suggest dividing the long paragraphs into two or three to facilitate the readability of the manuscript.*

Answer: We separated the long paragraphs into shorter ones.

*Referee: Lines 410-412: Move "We use Al as a normalizing parameter for the enrichment/depletion of elements due to its conservative behavior. The elements are presented as metal/Al ratios" the Methods section*

Answer: We moved these lines to the Methods section. Lines 251-258.

*Referee: Line 438: add "in core BTGC8" after "were lower"*

Answer: We added this sentence in line 447.

Referee: Lines 440-446: Move the first 3 sentences to the Methods section.

Answer: These lines were moved to the Methods section. Lines 249-251

*Referee: Lines 652: Yes, it seems to be contradictory, but a similar trend (trend towards higher productivity associated with decreasing reducing conditions) was observed off Peru (Salvatteci et al., 2014; Cardich et al., 2019). These authors based their observations on d15N, redox sensitive metals and benthic foraminifera assemblages. I suggest citing these papers and discuss whether the mechanisms proposed by these authors can explain or not the observed pattern off Guanaqueros Bay (BGGC5) and Tongoy Bay (BTGC8).*

Answer: We considered the referee's suggestion and we added other references to discuss the main mechanisms proposed for this oxygenation at coastal areas. Lines 662-677

*Referee: Lines 693-696. The westerlies were located in a more poleward position during the mid-Holocene (Lamy et al. 2001). Mollier-Vogel et al. (2019) erroneously indicate, citing Lamy et al. (2001), that the during the mid-Holocene the Westerlies were situated in a more northward position (see Salvatteci et al., 2019, for further discussion on this topic). This needs*

*to be corrected in the manuscript and the original citation must be also included (Lamy et al., 2001).*

Answer: We modified the paragraph changing the reference indicated by the referee. Lines 724-734

*Referee: Lines 725-730. The authors need to briefly compare and contrast their results with the reconstructed shifts of the storm tracks of the westerlies (Lamy et al., 2001).*

Answer: We discussed the southern Westerlies shifts in several paragraph; we compared our records in a general view for all SE Pacific first, and then reviewed the records for north-central Chile, establishing the main environmental conditions reported for the region. Lines 735-746, lines 778-780.

*Referee: Lines 763-767. Mollier et al. (2019) show evidence for decreased denitrification during the Mid-Holocene based on multiple d15N records along the coast. They do not show evidence for changes in sediment redox conditions. Rephrase this sentence or provide a better reference.*

Answer: A decreased denitrification implies a change in the redox condition and an oxygenation of the bottoms.  Therefore, the reference is appropriate. We added a sentence for further clarity. Lines 811-812

[revised manuscript text omitted]

**Comentario [A2]:** We modify the redaction of these lines due to the Al normalization was explained in methods section.

pattern, similar to the Fe/Al distribution, with a maximum value at cal BP 3500−3800

and a conspicuous upward trend over the past ~80 years. A third group, consisting of

Ba and Sr, exhibited a similar pattern but smoother, showing the maximum values before cal BP 6600. At BTGC8 core, a less clear pattern was demonstrated. Ca, Ni,

Cd, and P ratios at core BTGC8 showed only slightly decreasing values and very low peak values compared with core BGGC5; however, Ni/Al showed increasing concentrations over the past 80 years, which was not observed at core BGGC5.

Metal/Al ratios of Ba and Sr showed no substantial variation in time. In general, all the elemental concentrations were lower in core BTGC8 than in core BGGC5 and presented similar long-term reduction patterns toward the present, except for Cu, Ni, and Fe.

The authigenic enrichment expressed as EF values, suggest a large enrichment of nutrient-type elements in a period prior to cal BP 6600, following the trend of the

Me/Al ratios, except for Ba and Fe, which did not show authigenic enrichment. The EFs exhibited a sharp decrease in enrichment in recent times after cal BP 90 (Fig. 9).

**5. Discussion**

**5.1. Sedimentary composition of the cores: terrestrial *versus* biogenic inputs**

The sediments in the southern zones of the bays are a sink of fine particles transported from the north and the shelf (Figs. 5a, 5b), and respond to water circulation in the

Guanaqueros and Coquimbo Bays (Fig. 1) with two counter-rotating gyres moving counterclockwise to the north and clockwise to the south (Valle-Levinson and

Moraga, 2006) (Fig. 1). The differences established by the sediment composition of the bays show that the sediments of Guanaqueros Bay better represent the organic carbon flux to the bottom, with higher accumulation rates (mean value: 16 g $m^{-2} y^{-1}$)

and higher amounts of siliceous microfossils. Furthermore, is it a better zone than

Tongoy for pollen identification (Figs. 5a, 6 and 7). Both areas have sediments composed by winnowed particles and relatively refractory material (C/N: 9–11), which has a slightly lower isotopic composition than the TOC composition in the column water (−18‰, Fig. 2) and is transported by water circulating over the shelf.

The isotopic variations in $\delta^{13}C$ and $\delta^{15}N$ did not clearly establish differences between the sediments of the two bays; however, minor differences in $\delta^{15}N$ would indicate a greater influence of the upwelling nutrient supply and OMZ on the shelf, resulting in a

**Comentario [A3]:** These words were added according reviewer suggestion.

**Comentario [A4]:** Several lines were moved to the methods section (reviewer suggestion) therefore we must to modify this line.

[revised manuscript text omitted]

**Comentario [A18]:** We add some lines in order to discuss the main findings of several papers including Lamy et al 2001.

[revised manuscript text omitted]